# Transcription stress at telomeres leads to cytosolic DNA release and paracrine senescence

Athanasios Siametis [1,2,7], Kalliopi Stratigi [1,7], Despoina Giamaki[1,2,3,6], Georgia Chatzinikolaou[1], Alexia Akalestou-Clocher[1,2], Evi Goulielmaki[1], Brian Luke[3], Björn Schumacher [4,5] & George A. Garinis [1,2] ✉

Transcription stress has been linked to DNA damage -driven aging, yet the underlying mechanism remains unclear. Here, we demonstrate that $Tcea1^{-/-}$ cells, which harbor a TFIIS defect in transcription elongation, exhibit RNAPII stalling at oxidative DNA damage sites, impaired transcription, accumulation of R-loops, telomere uncapping, chromatin bridges, and genome instability, ultimately resulting in cellular senescence. We found that R-loops at telomeres causally contribute to the release of telomeric DNA fragments in the cytoplasm of $Tcea1^{-/-}$ cells and primary cells derived from naturally aged animals triggering a viral-like immune response. TFIIS-defective cells release extracellular vesicles laden with telomeric DNA fragments that target neighboring cells, which consequently undergo cellular senescence. Thus, transcription stress elicits paracrine signals leading to cellular senescence, promoting aging.

Transcription stress arises from various challenges encountered by the transcription machinery, such as bulky DNA lesions, DNA breaks, RNA polymerase pausing, and collisions with DNA repair factors[1]. The impact of transcription stress on cellular function and genome stability is profound, affecting the overall health of cells and organisms. Notably, defects in transcription-coupled repair, which lead to stalled RNA polymerases at DNA damage, have been associated with accelerated aging syndromes, such as Cockayne syndrome, suggesting a causal contribution of transcription-blocking DNA lesions to normal aging[2–5]. To cope with transcription roadblocks, cells have evolved mechanisms involving DNA repair pathways, stress response pathways, and specialized transcription factors[2,6–8]. Among these factors, Transcription Factor IIS (TFIIS) plays a critical role in resuming transcription after RNA polymerase II (RNAPII) stalls or pauses during elongation[9–12]. TFIIS stimulates a ribonucleolytic activity within RNAPII, enabling the cleavage of the displaced transcript during backtracking and facilitating

the restart of transcription[13]. Additionally, TFIIS promotes the bypass of 8-oxoguanine DNA lesions, preventing cell death[14,15] and helps RNAPII traverse nucleosomes during transcription elongation[16,17].

The impact of obstacles blocking RNAPII on transcription elongation is well-documented[1,2,18]. However, the mechanism by which RNAPII stalling can lead to physiological outcomes that threaten cell viability remains unclear. In this study, we investigated TFIIS-defective cells ($Tcea1^{-/-}$) and have established a previously unknown role of persistent RNAPII stalling in the loss of telomere integrity. We reveal that compromised transcription elongation, either due to defective TFIIS in $Tcea1^{-/-}$ or in naturally aged cells, leads to increased R-loops formation and the release of DNA fragments into the cytosol. We further determined that the DNA fragments accumulate in the cytoplasm and are packaged in extracellular vesicles that are released into recipient cells where they induce paracrine senescence. These findings provide novel insights into the mechanisms underlying cellular

[1]Institute of Molecular Biology and Biotechnology (IMBB), Foundation for Research and Technology-Hellas, Heraklion, Crete, Greece. [2]Department of Biology, University of Crete, Heraklion, Crete, Greece. [3]Institute of Molecular Biology (IMB), Mainz, Germany; Institute of Developmental Biology and Neurobiology (IDN), Johannes Gutenberg-Universität, Mainz, Germany. [4]Institute for Genome Stability in Aging and Disease, Medical Faculty, University of Cologne, Cologne, Germany. [5]Cologne Excellence Cluster for Cellular Stress Responses in Aging-Associated Diseases (CECAD) and Center for Molecular Medicine (CMMC), University of Cologne, Cologne, Germany. [6]Present address: Institute of Animal Pathology, Vetsuisse Faculty, University of Bern, 3012 Bern, Switzerland. [7]These authors contributed equally: Athanasios Siametis, Kalliopi Stratigi. ✉e-mail: garinis@imbb.forth.gr

senescence by unraveling how transcription-blocking DNA lesions are functionally linked to telomere integrity and innate immune DNA sensing as pivotal regulators of cellular senescence.

## Results

### Loss of TFIIS in mouse embryonic fibroblasts results in early cellular senescence

Complete abrogation of TFIIS causes embryonic lethality in mice[19]. To investigate the role of TFIIS in mammalian physiology, we, therefore, generated mice that carry loxP sites flanking exon 3 of the *Tcea1* gene (*Tcea1*[+/fl]) (see "Methods" and Supplementary Fig. 1A-B). Homozygous *Tcea1*[fl/fl] mice were then intercrossed with animals ubiquitously expressing the CMV-Cre transgene during early embryogenesis (Supplementary Fig. 1C). PCR amplification on genomic DNA from E12.5 *Tcea1*[-/-] mouse embryonic fibroblasts (MEFs) (Fig. 1A and Supplementary Fig. 1D) and Western blot analysis (Fig. 1B) confirmed the excision efficiency of *Tcea1* alleles and the lack of detectable TFIIS protein levels in *Tcea1*[-/-] MEFs, respectively. Consistent with previous data, complete abrogation of the *Tcea1* gene in mice leads to embryonic lethality[19]. Specifically, E12.5 *Tcea1*[-/-] embryos were smaller than *Tcea1*[+/+] or *Tcea1*[+/-] littermates, had fetal liver hypoplasia, and displayed tail curling (Fig. 1A). *Tcea1*[-/-] MEFs grew to a lower density and presented with a senescent-like "fried-egg" morphology and a larger, flattened cytoplasm (Supplementary Fig. 2A-B). In agreement, we find a higher number of senescence-associated β-galactosidase-positive (SA-β-gal[+]) MEFs in *Tcea1*[-/-] embryos compared to wild type (wt) controls (Fig. 1C). Additionally, there were higher mRNA levels for the senescence-associated cyclin-dependent kinase 4 and 6 inhibitor *p16*[INK4a] (*p16*), the transcription factor *Trp53* (p53) and the tumor suppressor *Retinoblastoma* (*Rb*) genes compared to wt control cells (Fig. 1D). Consistent with their senescent phenotype[20], BrdU-Propidium Iodide (PI) staining showed that *Tcea1*[-/-] MEFs accumulate in the G2/M phase of the cell cycle (Fig. 1E). We also transfected wt MEFs with a mutant form of TFIIS lacking the C-terminal domain III (TFIIS[ΔIII]), responsible for the activation of the RNAPII RNA cleavage function (Supplementary Fig. 2C)[13]. This overexpression recapitulates the G2/M accumulation of cells, suggesting that the *Tcea1*[-/-] phenotype involves the transcription-related function of TFIIS. To trace dividing cell generations, we next used the carboxyfluorescein succinimidyl ester (CFSE) proliferation assay, in combination with flow cytometry analysis, and monitored the dye dilution. Our findings showed that *Tcea1*[-/-] MEFs have a limited proliferative capacity and undergo fewer cell divisions within 48 h compared to wt controls (Fig. 1F and Supplementary Fig. 2D). Q-PCR analysis showed no increase in the expression of apoptosis-associated genes in *Tcea1*[-/-] MEFs (Supplementary Fig. 2E) and western blot analysis revealed comparable cleaved caspase 3 levels in *Tcea1*[-/-] and wt MEFs, indicating a lack of apoptosis induction in *Tcea1*[-/-] cells (Supplementary Fig. 2F). Accordingly, Annexin V/Propidium Iodide staining using flow cytometry showed no significant differences between the wt and *Tcea1*[-/-] cell populations (Supplementary Fig. 2G). RNA sequencing (RNA-Seq) profiling in *Tcea1*[-/-] and wt MEFs revealed 2811 differentially expressed genes [meta−false discovery rate (FDR) ≤ 0.01, fold change ≥ ±1.5; 1376 up-regulated genes; 1435 down-regulated genes]. Gene ontology classification revealed those biological processes with a significantly disproportionate number of genes relative to those expected in the murine genome (FDR ≤ 0.05). Among others, we found that type I and type II inflammatory response (GO:0006954, GO:0002828), cellular secretion (GO:1903530) and a response to hydrogen peroxide (GO:0042542) were enriched in the set of *Tcea1*[-/-] up-regulated genes. Down-regulated genes are associated with the regulation of transcription by RNAPII (GO:0006366), apoptosis (GO:0042981) or cell proliferation (GO:0008284) (Fig. 1G). Importantly, the overexpression of murine TFIIS in *Tcea1*[-/-] MEFs (Supplementary Fig. 2H) alleviated their senescent phenotype, as seen by the reduction of the *p16*[INK4a]

(p16) mRNA levels (Supplementary Fig. 2I) and the lower number of senescence-associated β-galactosidase-positive (SA-β-gal[+]) cells (Supplementary Fig. 2J), compared to non-transfected, TFIIS-deficient cells. Taken together, deletion of the *Tcea1* gene in MEFs leads to cellular senescence, accumulation of cells in G2/M phase, impaired proliferative capacity and absence of apoptosis.

### *Tcea1* deletion triggers the formation of R-loops and leads to genomic instability in MEFs

Senescent cells associate with higher levels of reactive oxygen species (ROS), generated mainly by dysfunctional mitochondria[21,22]. Our finding that abrogation of TFIIS leads to cellular senescence, prompted us to examine the levels of oxidative DNA lesions in MEFs. Immunofluorescence staining with an antibody raised against 8-oxo-7,8-dihydroguanine (8-oxoG) revealed significantly higher levels of 8-oxoG lesions in *Tcea1*[-/-] MEFs compared to wt control cells (Fig. 2A). The higher levels of oxidative DNA damage in *Tcea1*[-/-] MEFs were in line with the increased mitochondrial abundance associated with senescence[21,22] and, consistently, were reduced when the cells were treated with the antioxidant N-acetyl cysteine (NAC) (Fig. 2A and Supplementary Fig. 3A). As a positive control, $H_2O_2$ treatment increased the 8-oxoG levels in wt cells. Additionally, there was an increase in mitochondrial superoxide production, demonstrated by the accumulation of the mitochondrial marker TOM20, a subunit of the mitochondrial Translocase of the Outer Membrane (TOM) complex (Supplementary Fig. 3B) and the oxidation of the MitoSOX Red dye (Supplementary Fig. 3C). Consistently, the high 8-oxoG levels decreased when *Tcea1*[-/-] cells were treated with MitoTEMPO, an antioxidant compound known to scavenge mitochondrial superoxide (Supplementary Fig. 3D). TFIIS promotes transcription elongation by assisting RNAPII in bypassing 8-oxoG lesions, which prompted us to examine the impact of oxidative DNA damage on transcription arrest. In *Tcea1*[-/-] cells, we observed a notable increase in the serine 2-phosphorylated form of RNAPII (pS2-PolII), indicating enhanced levels of the elongating form of RNAPII, as evidenced by western blot analysis (Fig. 2B). Notably, the use of the antioxidant MitoTEMPO reduced the pS2-PolII levels in *Tcea1*[-/-] cells (Supplementary Fig. 3E). Moreover, chromatin immunoprecipitation (ChIP) for pS2-PolII, followed by dot blot analysis using an 8-oxoG-specific antibody, revealed that in *Tcea1*[-/-] MEFs, more elongating RNAPII was associated with 8-oxoG DNA lesions in TFIIS-deficient cells compared to wt controls (Fig. 2C). The pS2-PolII ChIP pulldown efficiency was found to be close to 80% both in wt and *Tcea1*[-/-] MEFs (Supplementary Fig. 4A). In support, a substantially higher percentage of untreated *Tcea1*[-/-] MEFs exhibited nuclear colocalization of pS2-PolII with 8-oxoG foci in comparison to wt cells. Treatment with hydrogen peroxide ($H_2O_2$) in both wt and *Tcea1*[-/-,] MEFs further increased this colocalization (Supplementary Fig. 4B). Any residual cytoplasmic 8-oxoG signal (after RNase A treatment or pre-extraction) observed in pS2-PolII/8-oxoG could be attributed to mitochondrial 8-oxoG levels, as portrayed by the co-staining of 8-oxoG with MitoTracker (Supplementary Fig. 4C). Together with the accumulation of pS2-PolII on 8-oxoG DNA lesions, we noticed a decrease in bromouridine (BrU) incorporation into nascent transcripts in *Tcea1*[-/-] MEFs (Fig. 2D), as observed upon treatment with the transcription elongation inhibitor 5,6-dichloro-1-beta-D-ribofuranosylbenzimidazole (DRB), which also decreased BrU incorporation in wt cells (Fig. 2D and Supplementary Fig. 4D). Additionally, treatment with the antioxidant MitoTEMPO (Supplementary Fig. 5A), or overexpression of TFIIS (Supplementary Fig. 5B), ameliorated the transcription impairment seen in *Tcea1*[-/-] MEFs. The BrU incorporation decrease observed in *Tcea1*[-/-] MEFs indicates impaired transcription elongation likely due to the absence of TFIIS-dependent transcript cleavage, which exacerbates transcription stress.

The accumulation of stalled RNAPII on actively transcribed genes leads to the gradual buildup of R-loops in cells[23]. Consistently, dot

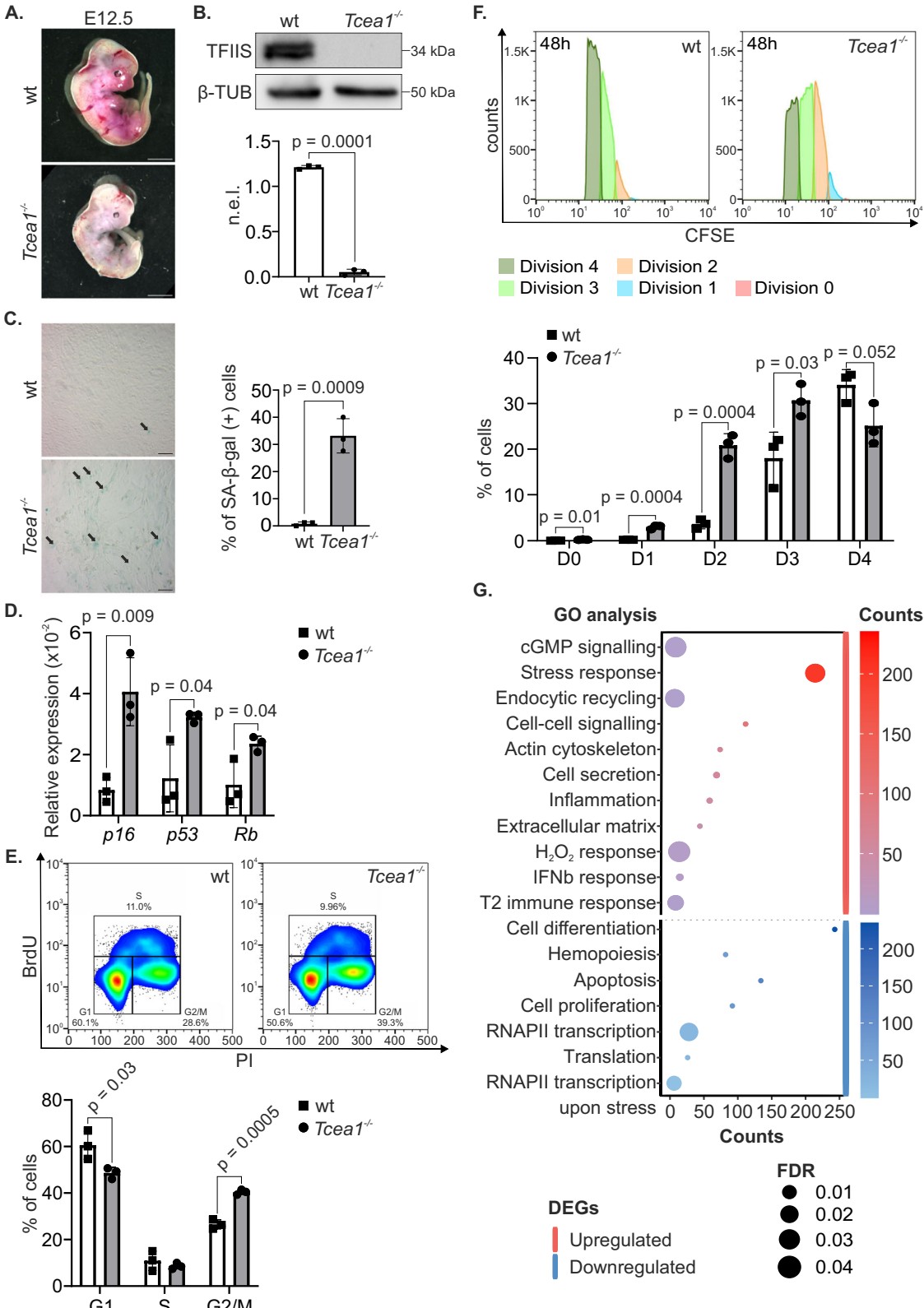

blotting of isolated genomic DNA and immunofluorescence assays using the RNA-DNA hybrid-specific S9.6 antibody, revealed an increase of R-loops in *Tcea1*[-/-] MEFs compared to wt control cells (Fig. 2E and Supplementary Fig. 5C). Treatment of MEFs with RNase H, which specifically degrades the RNA strand of an RNA-DNA duplex, effectively eliminated RNA-DNA hybrids, resulting in a significant reduction of R-loops in these cells (Fig. 2E and Supplementary Fig. 5C). The same

result was also observed with the use of MitoTEMPO (Supplementary Fig. 5C). R-loops expose long stretches of ssDNA leading to the spontaneous formation of DSBs, threatening genome integrity[24–26]. Consistently, immunofluorescence staining with antibodies raised against the phosphorylated form of H2AX (γH2AX), a general DNA damage marker, and 53BP1, a marker specifically for DNA double strand breaks (DSBs), showed a higher percentage of *Tcea1*[-/-] MEFs exhibiting at least

**Fig. 1 | *Tcea1*−/− MEFs exhibit a senescent phenotype. A** Representative images of E12.5 wt and *Tcea1*−/− embryos. **B** TFIIS protein levels in whole-cell extracts from wt and *Tcea1*−/− MEFs. β-TUBULIN (β-TUB) was used to normalize protein expression levels (n.e.l., n = 3 biologically independent replicates). **C** SA-β-gal assay in wt and *Tcea1*−/− MEFs (n = 3). Scale bar is set at 20 μM. **D** *p16, p53* and *Rb* mRNA levels in wt and *Tcea1*−/− MEFs (n = 3). **E** Cell cycle profiling and representative images of FACS analysis of wt and *Tcea1*−/− MEFs. The graph shows the percentage of wt and *Tcea1*−/− MEFs in each cell cycle phase (n = 3). **F** Carboxyfluorescein Diacetate Succinimidyl Ester (CFSE) proliferation assay and representative images of FACS analysis of wt and *Tcea1*−/− MEFs. The graph represents the percentage of cells that divided 0 to 4 times (D0-D4), during a 48 h time-period, according to CFSE fluorescence intensity analysis (n = 3). **G** Bubble plot of the Panther pathway enrichment analysis. The significantly enriched Gene Ontology (GO) terms of differentially expressed genes (DEGs) in *Tcea1*−/− compared to wt MEFs [meta−false discovery rate (FDR) ≤ 0.05, fold change ≥ ±1.3, n = 4] are indicated. The color scale indicates the number of genes in each corresponding pathway (up-regulated: red scale, down-regulated: blue scale) and the dot size indicates the FDR threshold. Data analysis was performed using two-tailed Student's *t* test. All data are presented as mean values ± SEM. Unless otherwise indicated, n = biologically independent experiments and scale bars are set at 5μm. Source data are provided as a Source Data file.

three foci positive for both γH2AX and 53BP1 (γH2AX+53BP1+) in comparison to wt controls (Fig. 2F). Etoposide-treated wt MEFs were used as a positive control (Supplementary Fig. 4E). Importantly, transfection of recombinant RNase H in *Tcea1*−/− MEFs substantially lowered the percentage of cells with γH2AX+53BP1+ foci, compared to untreated corresponding controls. In support, a BrdU pulse of wt and *Tcea1*−/− MEFs, followed by immunofluorescence against anti-BrdU, under non-denaturing conditions, revealed a larger amount of exposed ssDNA stretches in TFIIS-deficient cells compared to wt controls (Fig. 2G). Thus, abrogation of TFIIS results in the accumulation of 8-oxoG lesions, stalled RNAPII complexes, reduced ongoing mRNA synthesis and an increase in R-loop formation, ultimately leading to DNA breaks.

**Transcription stress impairs cell cycle progression due to telomere attrition and fusions**

To address whether *Tcea1*−/− MEFs might have compromised overall genome integrity, we performed a series of immunofluorescence assays. DAPI staining showed misshapen nuclei and chromatin bridges along with micronuclei in the cytoplasm of *Tcea1*−/− MEFs (Fig. 3A). Chromatin bridges arise when defects during mitosis prevent the complete segregation of chromosomes into their respective daughter cells[27]. To investigate this, we monitored the mitotic progression of *Tcea1*−/− MEFs. We first synchronized cells in the G1 phase using a transient thymidine block in cell growth media. Next, we treated the cells with nocodazole to block them in the G2/M phase, followed by a mitotic shake-off, to time all phases of the cell cycle progression. This approach revealed that the chromatin bridges arise during early anaphase and persist through telophase (Fig. 3B and Supplementary Fig. 6A). Comparison of the different mitotic phases showed that *Tcea1*−/− cells accumulate in anaphase, with fewer cells entering the subsequent phases and only a small percentage of TFIIS-deficient MEFs exiting mitosis, compared to wt MEFs (Fig. 3B, C and Supplementary Fig. 6A). Further analysis revealed that out of all *Tcea1*−/− cells progressing through mitosis, ~18% of cells exhibit anaphase bridges (Fig. 3D) and ~19% of cells present lagging/broken chromosomes (Fig. 3E). These findings indicate cell division abnormalities and explain the interphase chromatin bridges and micronuclei seen in *Tcea1*−/− MEFs. Chromatin bridges are generated during anaphase when fused chromosomes or sister chromatids are improperly segregated.

We, therefore, investigated the occurrence of fusion events by preparing metaphase spreads from colcemid-synchronized wt and *Tcea1*−/− MEFs. Fluorescence in situ hybridization (FISH) experiments, with a PNA telomere (TelC) probe, indicated that approximately 50% of the assessed metaphases displayed at least one telomere fusion event (Fig. 3F), while TelC-FISH also detected telomeric sequences spanning chromatin bridges in interphase cells (Fig. 3G). Telomere fusions occur when the chromosomal ends are left unprotected following telomere loss, which can arise due to double-strand breaks (DSBs), errors during DNA replication or as telomeres erode during aging[28]. QPCR revealed that the mRNA levels of telomerase reverse transcriptase (*Tert*) and telomerase RNA (*Terc*) genes remain unchanged in TFIIS-deficient cells (Supplementary Fig. 6B, C). Likewise, the levels of the telomeric long noncoding RNA TERRA known to be involved in telomerase recruitment and telomere maintenance, apart from the PAR locus[29] (Supplementary Fig. 6D), were found to remain unaffected in *Tcea1*−/− cells, as evidenced by qPCR analysis targeting specific subtelomeric sequences of chromosome 18 (Supplementary Fig. 6E), chromosome 2 (Supplementary Fig. 6F) and total TERRA RNA levels with cDNA generated with a TERRA-specific RT primer (Supplementary Fig. 6G), or random hexamers (Supplementary Fig. 6H), (TTACCC)7-Cy5.5 RNA FISH (Supplementary Fig. 6I-J) and total RNA dot blot experiments (Supplementary Fig. 6K). Nevertheless, Southern blotting for the detection and quantification of the telomere length of wt and *Tcea1*−/− MEFs and quantitative (Q-) FISH experiments revealed a moderate but detectable reduction of average telomere length in *Tcea1*−/− cells (Fig. 3H and Supplementary Fig. 6L), which was increased when we treated the cells with the antioxidant NAC (Fig. 3H). In agreement, we find a discernible reduction of telomere DNA in a yeast *dst1Δ* mutant, the *Tcea1* homolog in *Saccharomyces cerevisiae* (Supplementary Fig. 7A), while yeast TERRA levels are not affected (Supplementary Fig. 7B). As previously shown[30], telomere fusions arise due to the function of DNA Ligase IV, as part of the Non-Homologous End Joining (NHEJ) pathway. We thus reasoned that NHEJ inhibition would surpass the limitation of average quantification of telomere length and reveal the shorter telomeres in *Tcea1*−/− compared to wt cells. Indeed, specific inhibition of DNA Ligase IV, with the use of the selective inhibitor SCR130, revealed γH2Ax-stained telomeres (Fig. 3I and Supplementary Fig. 6M) and led to *Tcea1*−/− metaphase chromosomes without fusions, but with discernible chromosome ends with short or completely missing telomeres (Fig. 3J). We further quantified the relative size of the telomeres at each chromatid end with qFISH, which corroborated the average telomere length measurements in interphase cells, and revealed a clear difference in single telomere sizes between wt and TFIIS-deficient cells (Fig. 3K). Taken together, our findings indicate that a defect in TFIIS results in shorter telomeres that become fused, leading to anaphase bridges and mitotic aberrations.

**A defect in TFIIS associates with telomere uncapping and DDR activation**

Telomere uncapping can occur as a consequence of defects in telomere structure, DNA breaks or telomere shortening, which in turn activate the DNA damage response (DDR)[31]. To test whether *Tcea1*−/− telomeres activate DDR, we examined the presence of Telomere Dysfunction-Induced foci (TIFs) in *Tcea1*−/− MEFs using TelC-FISH experiments, along with immunostaining for γH2AX and 53BP1. Our results showed that approximately 30% of *Tcea1*−/− MEFs exhibited dysfunctional telomeres (Fig. 4A). Furthermore, we observed significantly higher levels of the phosphorylated ataxia telangiectasia-mutated protein (pATM), a central mediator of the DDR, in *Tcea1*−/− cells compared to wt controls (Fig. 4B). The shelterin complex is a group of six proteins, namely TRF1, TRF2, POT1, RAP1, TIN2, and TPP1 that bind to and protect telomeres by regulating telomerase activity, preventing telomeres from being recognized as DNA breaks, and ensuring the timely recruitment of DNA repair proteins[32]. Although the mRNA and protein levels of TRF1, TRF2 or TIN2 were unaltered in

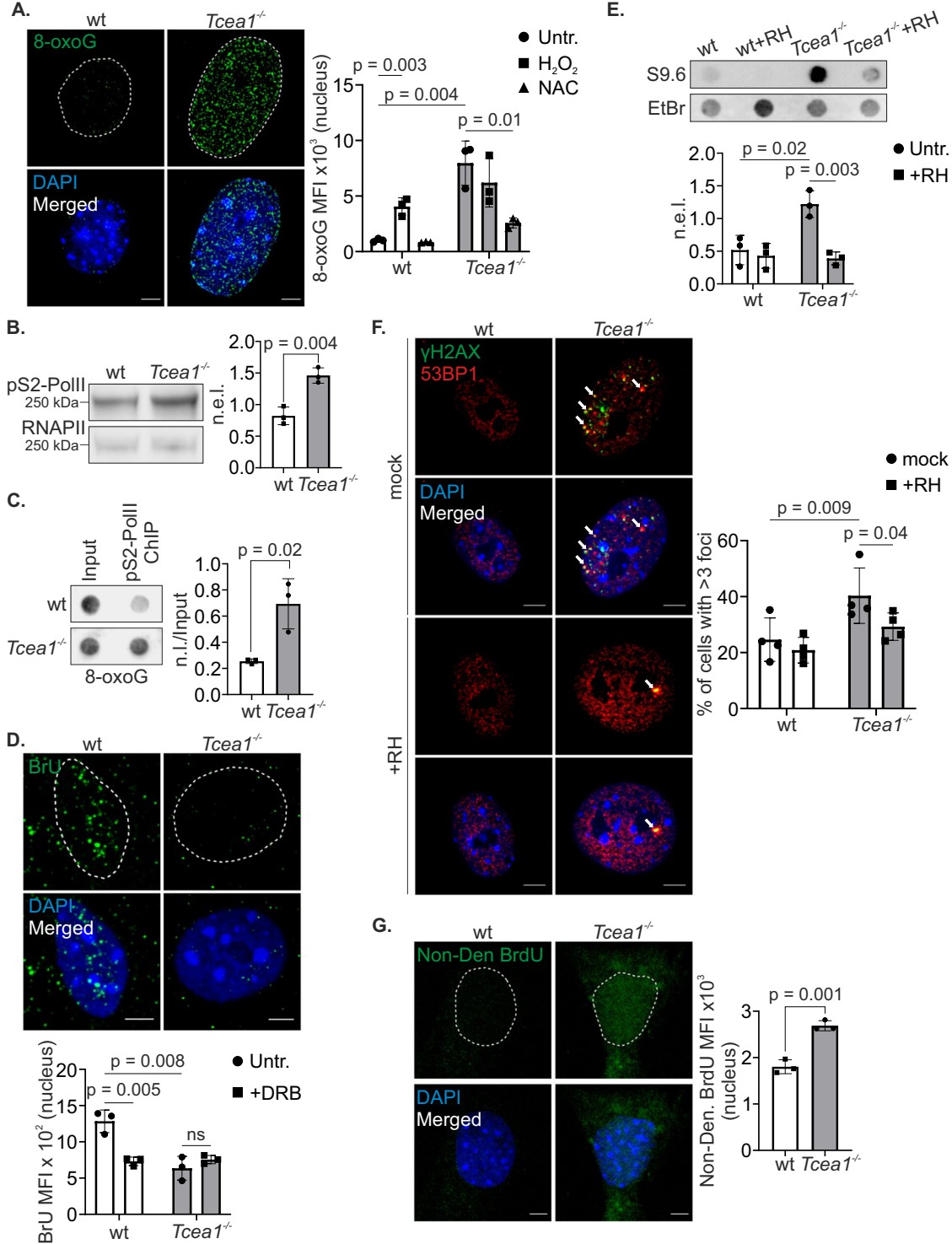

**Fig. 2 | Transcription stress in *Tcea1*⁻/⁻ MEFs leads to genomic instability.**
**A** Immunostaining of 8-Oxoguanine (8-oxoG) in untreated wt and *Tcea1*⁻/⁻ MEFs
(n = 3). The graph depicts the 8-oxoG MFI per cell nucleus for untreated, H₂O₂-
treated and NAC-treated cells. **B** Serine 2 (pS2)-phosphorylated RNAPII (pS2-PolII)
in whole-cell extracts from wt and *Tcea1*⁻/⁻ MEFs. The graph depicts the total
RNAPII-normalized protein expression levels (n.e.l., n = 3). **C** Dot blot against
8-oxoG in pS2-PolII ChIP samples from wt and *Tcea1*⁻/⁻ MEFs. The graph represents
pS2-PolII ChIP samples normalized over input samples (n.l./Input, n = 3). **D** BrU
incorporation in untreated wt and *Tcea1*⁻/⁻ MEFs. The graph shows the BrU MFI per
cell nucleus for untreated and DRB-treated cells (n = 3). **E** Dot blot against s9.6 in
untreated and Rnase H-treated genomic DNA from wt and *Tcea1*⁻/⁻ MEFs. The graph

represents input samples normalized over Ethidium Bromide (EtBr) labeling, as a
loading control (n = 3). **F** Immunofluorescence detection of γH2AX and 53BP1 co-
localized foci in *Tcea1*⁻/⁻ MEFs, in the presence or absence of transfected recom-
binant RNase H. The graph represents the percentage of cells with >3 co-localized
γH2AX/53BP1 foci per cell nucleus (n = 4). **G** BrdU immunostaining under non-
denaturing (Non-Den) conditions in wt and *Tcea1*⁻/⁻ MEFs. The graph depicts the
BrdU MFI per cell nucleus (n = 3). Data analysis was performed using two-tailed
Student's *t* test. All data are presented as mean values ± SEM. Unless otherwise
indicated, n = biologically independent experiments and scale bars are set at 5 μm.
Source data are provided as a Source Data file.

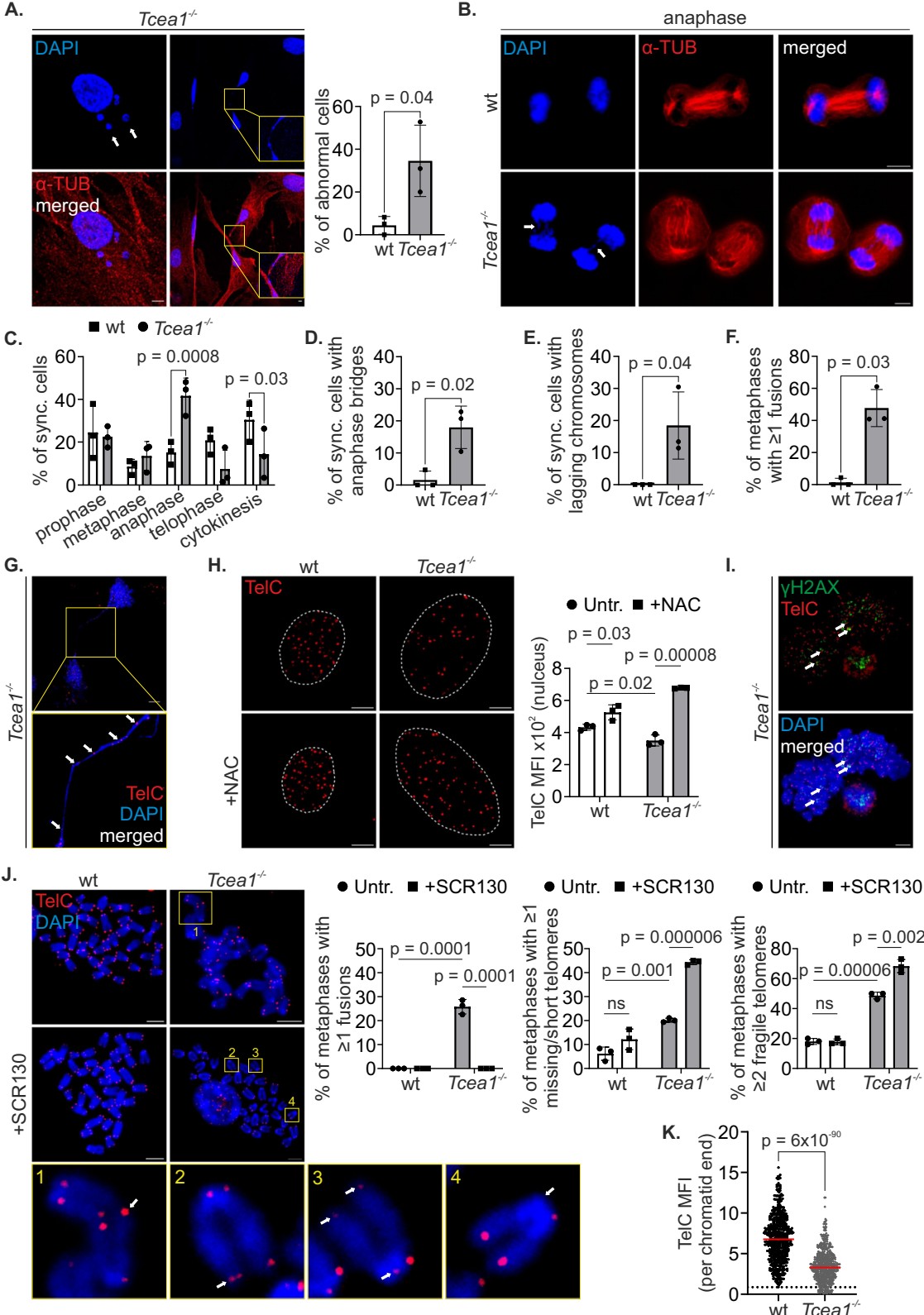

*Tcea1*[-/-] compared to wt MEFs (Fig. 4C, D and Supplementary Fig. 7C–F), chromatin immunoprecipitation (ChIP) assays showed that TRF1, TRF2 and TIN2 ChIP signals are significantly reduced on telomere DNA (Fig. 4E and Supplementary Fig. 7G, H), justifying the observed fragility[33] (Fig. 3J, as indicated) and DDR activation on *Tcea1*[-/-] telomeres. The same was observed for the telomeric recruitment of the Rap1 protein in the yeast *dst1Δ* mutant (Supplementary Fig. 7I). Next,

we conducted additional experiments to confirm the release of TRF1 from telomeric DNA in *Tcea1*[-/-] telomeres. This was prompted by the *Tcea1*[-/-] transcription defect and the role of TRF1 in controlling telomere silencing and assisting in telomere transcription through its interaction with RNAPII[34]. We utilized TelC-FISH and immunofluorescence staining for TRF1 specifically on *Tcea1*[-/-] telomeres (Fig. 4F). When TRF1 is not bound to telomeres, it undergoes rapid

**Fig. 3 | Impaired cell cycle progression due to telomere end fusions and telomere attrition in *Tcea1*$^{-/-}$ MEFs. A** Immunostaining of α-TUBULIN (α-TUB) in *Tcea1*$^{-/-}$ MEFs (n = 3). Graph depicts the percentage of cells with at least one abnormality (micronuclei – white arrows, chromatin bridge – yellow insert). **B** Immunofluorescence of cell cycle-synchronized wt and *Tcea1*$^{-/-}$ MEFs. Images show cells during the anaphase of the cell cycle (lagging chromosome - left arrow, chromatin bridge - right arrow). **C** Percentage of wt and *Tcea1*$^{-/-}$ MEFs at each cell cycle phase (n = 3). Percentage of wt and *Tcea1*$^{-/-}$ MEFs with (**D**). Anaphase bridges (n = 3). **E** Lagging chromosomes (n = 3). **F** At least one fusion event (n = 3, metaphases >20/genotype). **G** Telomeric sequences on a chromatin bridge of *Tcea1*$^{-/-}$ MEFs (white arrows). **H** Quantitative FISH of untreated and NAC-treated wt and *Tcea1*$^{-/-}$ MEFs for telomeric DNA (n = 3). The graph depicts the mean fluorescence intensity (MFI) per cell nucleus. **I** Immunostaining of γH2AX with FISH for telomeric DNA in *Tcea1*$^{-/-}$ metaphase spreads. White arrows denote γH2AX signal on telomeres. **J** Wt and *Tcea1*$^{-/-}$ metaphase spreads, in the presence or absence of SCR130 inhibitor. Telomeric DNA was detected by FISH (n = 3, metaphases >50/condition). White arrow in yellow frames: 1: chromosomes with telomere fusion. 2: chromosome with fragile telomere. 3: chromosome with short telomere. 4: chromosome with missing telomere. The graphs depict the percentage of metaphases with at least one fusion event (left), missing/short telomere (middle) or fragile telomere (right). **K** Telomeric DNA MFI per chromatid end of wt and *Tcea1*$^{-/-}$ MEFs. Dotted line represents the shortest wt telomere (n = 3). Data analysis was performed using two-tailed Student's *t* test. All data are presented as mean values ± SEM. Unless otherwise indicated, n = biologically independent experiments and scale bars are set at 5 μm. Source data are provided as a Source Data file.

ubiquitination and is targeted for proteasomal degradation[35]. Consistent with this, our observations in *Tcea1*$^{-/-}$ MEFs reveal that TRF1 is released from the chromatin-bound fraction and instead accumulates in the cytoplasm of these cells, as seen both by Western blot after protein extract fractionation (Supplementary Fig. 7J) and by immunofluorescence experiments with and without detergent-mediated removal of soluble proteins (Supplementary Fig. 7K). These findings support the notion that TFIIS deficiency disrupts the proper localization of TRF1, leading to its aberrant cytoplasmic accumulation. Consistently, when *Tcea1*$^{-/-}$ cells were treated with the proteasome inhibitor MG132, we observed an increase in the ubiquitinated form of immunoprecipitated TRF1 compared to corresponding wt controls (Supplementary Fig. 7L–M). Taken together, our findings indicate that in *Tcea1*$^{-/-}$ cells, TRF1 is released from telomeres, leading to its ubiquitination and subsequent translocation to the cytoplasm. As a result, telomeres become uncapped and dysfunctional, leading to the activation of an ATM-mediated DNA damage response on the chromosome ends.

## TRF1 interacts with pS2-PolII and is released from telomeres upon oxidative stress

ROS-induced DNA damage accelerates telomere attrition and aging. Telomere physiology and the abundance of guanines in their repeat sequences render them particularly vulnerable to oxidative damage[36]. Importantly, the presence of a single 8-oxoG lesion on telomeric DNA can disrupt TRF1 binding[37,38]. We, therefore reasoned that the high global 8-oxoG levels observed in *Tcea1*$^{-/-}$ MEFs could potentially impact telomeric sequences leading to the reduced recruitment of TRF1. To test this, we utilized OxiDIP, a sensitive single-stranded DNA immunoprecipitation approach, coupled to qPCR, to detect 8-oxoG lesions on telomeres. As anticipated, telomeres from *Tcea1*$^{-/-}$ and $H_2O_2$-treated wt MEFs accumulated a greater number of 8-oxoG lesions compared to untreated wt cells (Fig. 4G). A GC-rich region was used as a positive control, showing a greater accumulation of 8-oxoG lesions in the *Tcea1*$^{-/-}$ and $H_2O_2$-treated wt MEFs, while an AT-rich region, with no such induction, served as a negative control. Consistently, TelC-FISH combined with immunofluorescence experiments showed an increase in 8-oxoG foci on telomeres in *Tcea1*$^{-/-}$ cells compared to wt controls (Fig. 4H).

Next, we examined whether treatment of MEFs with $H_2O_2$ impacts the recruitment of TRF1 on telomeres. Immuno-FISH experiments demonstrated a significant decrease in TRF1 recruitment on the telomeres of $H_2O_2$-treated wt MEFs when compared to the untreated control cells (Fig. 5A). In line with the established role of TFIIS in facilitating RNAPII bypass of 8-oxoG DNA lesions, we observed an increase in TFIIS recruitment on telomeres following treatment with $H_2O_2$ (Fig. 5B). Similarly, the ChIP signals of stalled elongating pS2-PolII increased when TFIIS was abrogated in *Tcea1*$^{-/-}$ telomeres (Fig. 5C). A series of immunoprecipitation assays revealed that TFIIS is in complex with TRF1 and RNAPII in MEFs and these interactions persist in $H_2O_2$-treated wt cells (Fig. 5D and Supplementary Fig. 8A). Likewise, TRF1

remains in complex with pS2-PolII in *Tcea1*$^{-/-}$ MEFs (Fig. 5E). The interaction of TRF1 with the elongating pS2-PolII and TFIIS prompted us to test whether active transcription is somehow involved in TRF1 recruitment on telomeres. To test this, we utilized BrU incorporation coupled to immuno-FISH, which showed that ongoing transcription is significantly impaired in *Tcea1*$^{-/-}$ telomeres (Fig. 5F). Subsequently, we employed $H_2O_2$-treated wt MEFs and subjected them to a recovery period of 16 h after washing away the $H_2O_2$. During this recovery period, we treated the cells with or without the transcription elongation inhibitor 5,6-dichloro-1-beta-D-ribofuranosylbenzimidazole (DRB). Our findings indicate a decrease in TRF1 ChIP signals on the telomeres of $H_2O_2$-treated wt MEFs. However, when the $H_2O_2$ is removed and the cells undergo a recovery period of 16 h, the TRF1 ChIP signals are fully restored (Fig. 5G). Importantly, we observed that when $H_2O_2$-treated cells underwent recovery in the presence of DRB, the TRF1 ChIP signals remained reduced (Fig. 5G), a reduction which was not observed upon DRB treatment alone (Supplementary Fig. 8B). TRF1 ChIP signals were only restored when the reversible transcription inhibitor was removed from the culture media, highlighting the dependence of TRF1 recruitment on active transcription during the recovery period (Fig. 5G). Immunofluorescence experiments for TRF1, in combination with TelC-FISH further confirmed these results (Supplementary Fig. 8C–E). Of note, despite the fact that, as expected, DRB-treated cells show a reduction in *Trf1* mRNA levels (Supplementary Fig. 8F), no differences in TRF1 protein levels were observed by Western blot (Supplementary Fig. 8G). Likewise, when $H_2O_2$-treated MEFs underwent recovery in the presence of the reversible transcription inhibitor Triptolide (TPL) or the non-reversible Actinomycin D (Act.D), the TRF1 signals on telomeres remained reduced (Supplementary Fig. 8H). Unlike with TPL or DRB, the removal of Act.D failed to restore the TRF1 protein levels on telomeres. In contrast, we discovered that the same recovery time was not adequate for the re-loading of TRF1 on telomeres, when utilizing the non-reversible DNA damaging agent Illudin S (Ill.S), which introduces transcription-blocking lesions (Supplementary Fig. 8H). Taken together, our findings indicate that *Tcea1*$^{-/-}$ telomeres exhibit higher levels of 8-oxoG DNA lesions, stalled RNAPII and R-loops. TFIIS is found to be in complex with pS2-PolII and TRF1. Notably, TRF1 is released from telomeres upon oxidative DNA damage and involves active transcription to be recruited on telomeric DNA.

## R-loop-derived cytosolic telomeric fragments associate with an inflammatory response

Next, we sought to investigate whether the global formation of R-loops, resulting from transcription stress, in *Tcea1*$^{-/-}$ MEFs had any influence on telomere integrity. DRIP (DNA-RNA hybrids immunoprecipitation) experiments revealed substantial accumulation of RH-sensitive R-loops in *Tcea1*$^{-/-}$ telomeres (Fig. 6A); the R-loop prone *Rpl13a* gene locus was used as a positive control (Supplementary Fig. 9A). Interestingly, when the cells underwent ~6 additional cell division cycles (passage 2 to passage 5), the number of telomeric RNA-DNA hybrids quantified in the P2 cell population, were diminished in

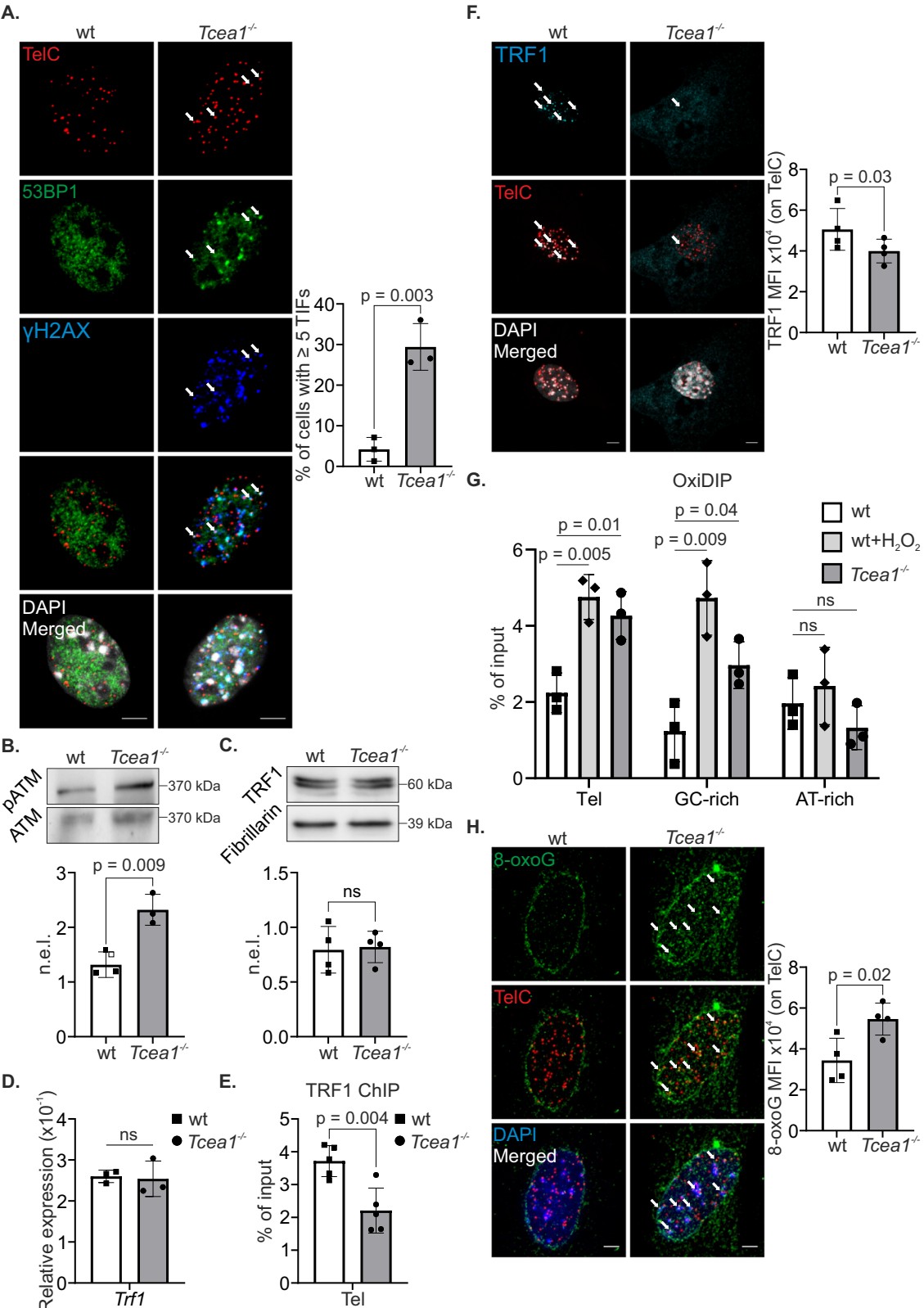

the P5. In order to further assess the involvement of R-loops as a source of the *Tcea1*$^{-/-}$ genome instability, we performed Breaks Labeling, Enrichment on Streptavidin, and Sequencing (BLESS) in order to directly label and isolate DSBs in *Tcea1*$^{-/-}$ and wt cells. In line with the decrease in R-loop levels seen in *Tcea1*$^{-/-}$ cells cultured from P2 to P5, we observed an increase in DSB levels in P5 compared to P2 *Tcea1*$^{-/-}$ MEFs. The *Rpl13a* gene, used as an R-loop-prone positive control, showed similar kinetics in DSB quantification, albeit in a much smaller range, while a non-transcribed intergenic region, used as a negative control, presented no DSB accumulation (Supplementary Fig. 9B). More importantly, BLESS experiments, upon transfection with recombinant RNase H, showed that the resolution of R-loops results in fewer DSBs on telomeres and the R-loop-prone gene, providing additional evidence for the R-loop-induced genomic instability

**Fig. 4 | TRF1 release from *Tcea1⁻/⁻* telomeres is associated with telomere dysfunction induced foci (TIFs). A** Immunofluorescence of 53BP1 and γH2AX with in situ hybridization of telomeric DNA (TelC) in wt and *Tcea1⁻/⁻* MEFs. Arrows denote TIFs on telomeres. The graph depicts the mean percentage of cells presenting ≥5 TIFs (n = 3). **B** pATM protein levels in wt and *Tcea1⁻/⁻* MEFs whole-cell extracts (n = 3). The graph depicts the total ATM-normalized protein expression levels (n.e.l.). **C** TRF1 protein levels in whole-cell extracts from wt and *Tcea1⁻/⁻* MEFs. Fibrillarin was used to normalize protein expression levels (n.e.l., n = 4). **D** *Trf1* mRNA levels in wt and *Tcea1⁻/⁻* MEFs (n = 3). **E** ChIP signals of TRF1 protein (shown as percentage of input after IgG normalization) on telomeres of wt and *Tcea1⁻/⁻* MEFs (n = 5). **F** Immunofluorescence of TRF1 with in situ hybridization of telomeric DNA (TelC) in wt and *Tcea1⁻/⁻* MEFs. The graph depicts the mean fluorescence intensity (MFI) of TelC in *Tcea1⁻/⁻* MEFs and wt controls (n = 4). **G** OxiDIP signals of 8-oxoG (shown as percentage of input after IgM normalization) on telomeres, GC-rich and AT-rich regions of untreated wt, $H_2O_2$-treated wt and untreated *Tcea1⁻/⁻* MEFs (n = 3). **H** Immunofluorescence against 8-oxoG with in situ hybridization of telomeric DNA (TelC) in wt and *Tcea1⁻/⁻* MEFs. White arrows indicate 8-oxoG signal on telomeres (n = 4). The graph depicts the 8-oxoG mean fluorescence intensity (MFI) on telomeric DNA (TelC) in *Tcea1⁻/⁻* and wt MEFs. Data analysis was performed using two-tailed Student's *t* test. All data are presented as mean values ± SEM. Unless otherwise indicated, n = biologically independent experiments and scale bars are set at 5µm. Source data are provided as a Source Data file.

(Supplementary Fig. 9C). Lastly, treatment of *Tcea1⁻/⁻* MEFs with the antioxidant NAC, reduced the observed DSBs accumulation on telomeres and the R-loop-prone *Rpl13a* gene (Supplementary Fig. 9D), suggesting that the 8-oxoG accumulation in TFIIS-deficient cells leads to pS2-PolII stalling (Supplementary Fig. 4B), R-loop accumulation (Supplementary Fig. 5C) and genome instability.

In view of the decrease in R-loop levels along with the telomere attrition seen in *Tcea1⁻/⁻* MEFs, we employed the TelC PNA probe and detected higher levels of DNA telomeric fragments in the cytoplasm of *Tcea1⁻/⁻* MEFs compared to wt controls (Fig. 6B). Further treatment of wt and *Tcea1⁻/⁻* MEFs with the ssDNA-specific S1 nuclease or RNase A, showed that these DNA fragments are mainly single-stranded (Supplementary Fig. 9E). Importantly, when MEFs were transfected with recombinant RNase H, we observed a marked reduction in the abundance of cytosolic telomeric DNA fragments indicating that R-loops are a major source of these fragments (Fig. 6B). Sequencing of these cytoplasmic fragments revealed an enrichment of TTAGGG repeats in *Tcea1⁻/⁻* MEFs, compared to wt controls (Supplementary Fig. 10A). In the lack of a murine telomere-to-telomere genome assembly, we exploited the conservation of telomeric repetitive sequence between mouse and humans and verified the sequenced reads by mapping them on the complete telomere-to-telomere human reference genome (T2T-CHM13), which showed an enrichment of TTAGGG reads on human telomeres in *Tcea1⁻/⁻* compared to wt MEFs (Supplementary Fig. 10B). We went further to verify the *Tcea1⁻/⁻* enrichment on selected mouse TTAGGG-containing chromatin regions; the TERRA-expressing locus on Chromosome 2 (Supplementary Fig. 10C), the PAR locus on the X Chromosome (Supplementary Fig. 10D) and on the Y Chromosome (Supplementary Fig. 10E) and the Chromosome 18 telomere (Supplementary Fig. 10F) showed an increased read coverage (reads/50 bp bin), in *Tcea1⁻/⁻* compared to wt MEFs, suggesting the presence of these DNA sequences in the cytoplasms of TFIIS-deficient cells. The Rpl13a locus, which was found to accumulate R-loops in *Tcea1⁻/⁻* cells (Supplementary Fig. 9A), was also found to be enriched in the cytoplasmic DNA fragments, yet to a smaller extent, when compared to TTAGGG repeats (Supplementary Fig. 11A). An intergenic region from chromosome X is displayed as a negative control (Supplementary Fig. 11B). Similarly, sequences from other R-loop-prone genes were found to be increased in the cytoplasm of *Tcea1⁻/⁻* cells compared to wt controls (Supplementary Fig. 11C-E) indicating the contribution of transcription stress in genome instability.

In view of the R-loop and DNA damage accumulation evidenced in *Tcea1⁻/⁻* MEFs, genome-wide, we examined the contribution of telomere dysfunction in the senescent phenotype of these cells. To this end, we overexpressed either TERT, TFIIS or both TERT and TFIIS, in wt and *Tcea1⁻/⁻* MEFs over the course of 4 passages (Supplementary Fig. 11F, G). We confirmed by qFISH that the average telomere length was increased in TERT⁺, TFIIS⁺ and TERT⁺/TFIIS⁺ *Tcea1⁻/⁻* cells, compared to untransfected *Tcea1⁻/⁻* controls (Supplementary Fig. 11H). Importantly, we observed that TERT, TFIIS or TERT/TFIIS overexpression resulted in a decrease in *p16^{INK4a}* (p16) mRNA levels (Supplementary Fig. 11I) and a significant reduction of senescence-associated β-galactosidase-positive (SA-β-gal⁺) *Tcea1⁻/⁻* cells (Supplementary Fig. 12A), suggesting that telomere integrity is a key contributing factor in the observed *Tcea1⁻/⁻* senescent phenotype. In order to assess if the non-canonical roles of TERT in gene expression impinge on the *Tcea1⁻/⁻* phenotype, we additionally treated cells with G-rich terminal oligonucleotides (GTR)[39,40]. Regardless of the few passages the cells were cultured with the G-rich oligonucleotides, there was a significant increase in average telomere length in *Tcea1⁻/⁻* MEFs (Supplementary Fig. 12B). Consistently, p16 mRNA levels were reduced and the number of SA-β-gal⁺ *Tcea1⁻/⁻* cells was decreased (Supplementary Fig. 12C, D).

We and others have shown that the presence of DNA moieties in the cytoplasm triggers a viral-like response, leading to the expression of type I interferon-related genes[41–45]. In agreement, qPCR experiments showed increased mRNA levels for the *Ifnβ* gene and Interferon Signature Genes (ISGs), such as *Mx1*, *Ifitm1*, *Ifit2*, *Ifi207* and *Irf1*, in *Tcea1⁻/⁻* MEFs compared to wt controls (Fig. 6C and Supplementary Fig. 13A), also supported by the total mRNA sequencing analyses (Fig. 1G). Consistently, incubation of the cells with S1 nuclease-loaded vesicles resulted in a decrease in cytoplasmic TelC-positive fragments (Supplementary Fig. 13B) and a consequent reduction in the type I interferon-related expression levels (Fig. 6C and Supplementary Fig. 13A). Senescence and inflammation are tightly associated with aging and the premature onset of age-related diseases. To evaluate the potential contribution of transcription stress-induced telomere de-protection to the aging process, we next studied 2- and 24-month old naturally aged wt mice. Similar to the decrease in TRF1 levels observed on the telomeres of *Tcea1⁻/⁻* or $H_2O_2$-treated wt MEFs, a series of IF/TelC-FISH experiments revealed a reduction in TRF1 recruitment on the telomeres of primary hepatocytes (Fig. 6D) and pancreatic cells (Supplementary Fig. 13C) derived from naturally aged 24-month-old mice, compared to 2-month-old young animals. No differences in TRF1 protein levels were detected between hepatocytes or pancreatic cells from 2 m and 24 m old mice (Supplementary Fig. 13D-E). Consistently, primary hepatocytes and pancreatic cells from 24-month-old animals also showed increased levels of telomeric fragments in their cytoplasm, in comparison to cells derived from 2-month-old mice (Fig. 6E and Supplementary Fig. 13F). The cytoplasmic TelC signal was diminished when R-loops were removed in cells transfected with recombinant RNase H (Fig. 6E and Supplementary Fig. 13F). In line, qPCR analyses confirmed a pro-inflammatory phenotype in the 24-month-old compared to 2-month-old animals, with increased mRNA levels for the *Ifnβ*, *Mx1*, *Ifitm1*, *Ifit2* and *Irf1* genes (Fig. 6F and Supplementary Fig. 13G, H). Notably, this inflammation induction was alleviated by incubating the cells with vesicle-delivered recombinant ssDNA-specific S1 nuclease, pinpointing its source to the cytoplasmic DNA fragments (Fig. 6F and Supplementary Fig. 13H). Taken together, our findings demonstrate that transcription stress-induced telomeric RNA-DNA hybrids causally contribute to the generation of cytosolic Tel-DNA fragments. Importantly, these fragments are associated with a type I inflammatory response, both in TFIIS-deficient MEFs and primary cells derived from naturally aged mice.

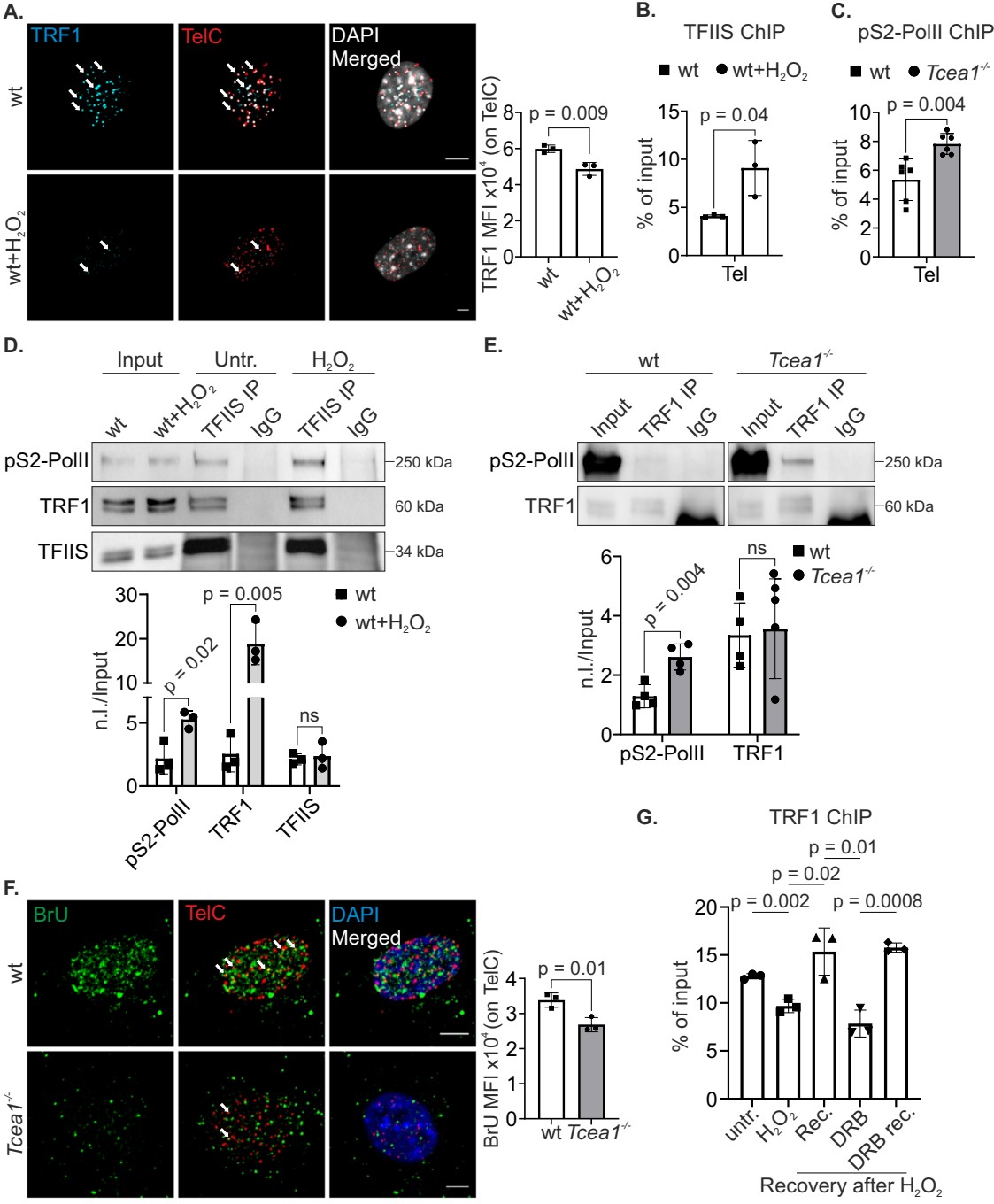

**Fig. 5 | Transcription-associated TRF1 recruitment on telomeres.**
**A** Immunofluorescence of TRF1 with in situ hybridization of telomeric DNA (TelC) in untreated and $H_2O_2$-treated wt MEFs. White arrows indicate cells with reduced TRF1 signal on telomeres. The graph shows the TRF1 MFI on telomeric DNA (TelC) in untreated and $H_2O_2$-treated wt MEFs (n = 3). **B** ChIP signals of TFIIS protein (shown as percentage of input after IgG normalization) on telomeres of untreated and $H_2O_2$-treated wt MEFs (n = 3). **C** ChIP signals of pS2-PolII protein on telomeres of wt and $Tcea1^{-/-}$ MEFs (n = 6). **D** Co-immunoprecipitation experiments using anti-TFIIS in nuclear extracts of untreated and $H_2O_2$-treated wt MEFs, analyzed by western blotting for pS2-PolII and TRF1. The graph represents normalized levels of IP samples over input samples (n.l./Input, n = 3). **E** Co-immunoprecipitation experiments using anti-

TRF1 in nuclear extracts of wt and $Tcea1^{-/-}$ MEFs, analyzed by western blotting for pS2-PolII. The graph represents normalized levels of IP samples over input samples (n.l./Input, n = 3). **F** BrU incorporation in telomeres (Cy3-PNA TelC probe) of wt and $Tcea1^{-/-}$ MEFs (n = 3). The graph shows the BrU MFI per telomere. **G** ChIP signals of TRF1 protein on telomeres of wt MEFs (untr: untreated, $H_2O_2$: treatment with $H_2O_2$, Rec: $H_2O_2$-treated, washed and incubated for 16 h, DRB: $H_2O_2$-treated, washed and incubated with DRB for 16 h, DRB rec: $H_2O_2$-treated, washed, incubated with DRB for 16 h, washed and incubated for 6 h, n = 3). Data analysis was performed using two-tailed Student's $t$ test. All data are presented as mean values ± SEM. Unless otherwise indicated, n = biologically independent experiments and scale bars are set at 5 μm. Source data are provided as a Source Data file.

## $Tcea1^{-/-}$- secreted EVs are loaded with telomeric DNA and trigger bystander senescence

Senescent cells can induce cellular senescence in neighboring normal "bystander" cells in vitro through a mechanism that remains elusive[46]. Extracellular vesicles (EVs) are small, membrane-bound structures that

are released by cells into the extracellular space. They are produced by most cells and are involved in a wide range of physiological and pathological processes[47]. Previous work in our lab showed that upon DNA damage, macrophages release EVs that target recipient cells leading to metabolic reprogramming and inflammation[48]. We,

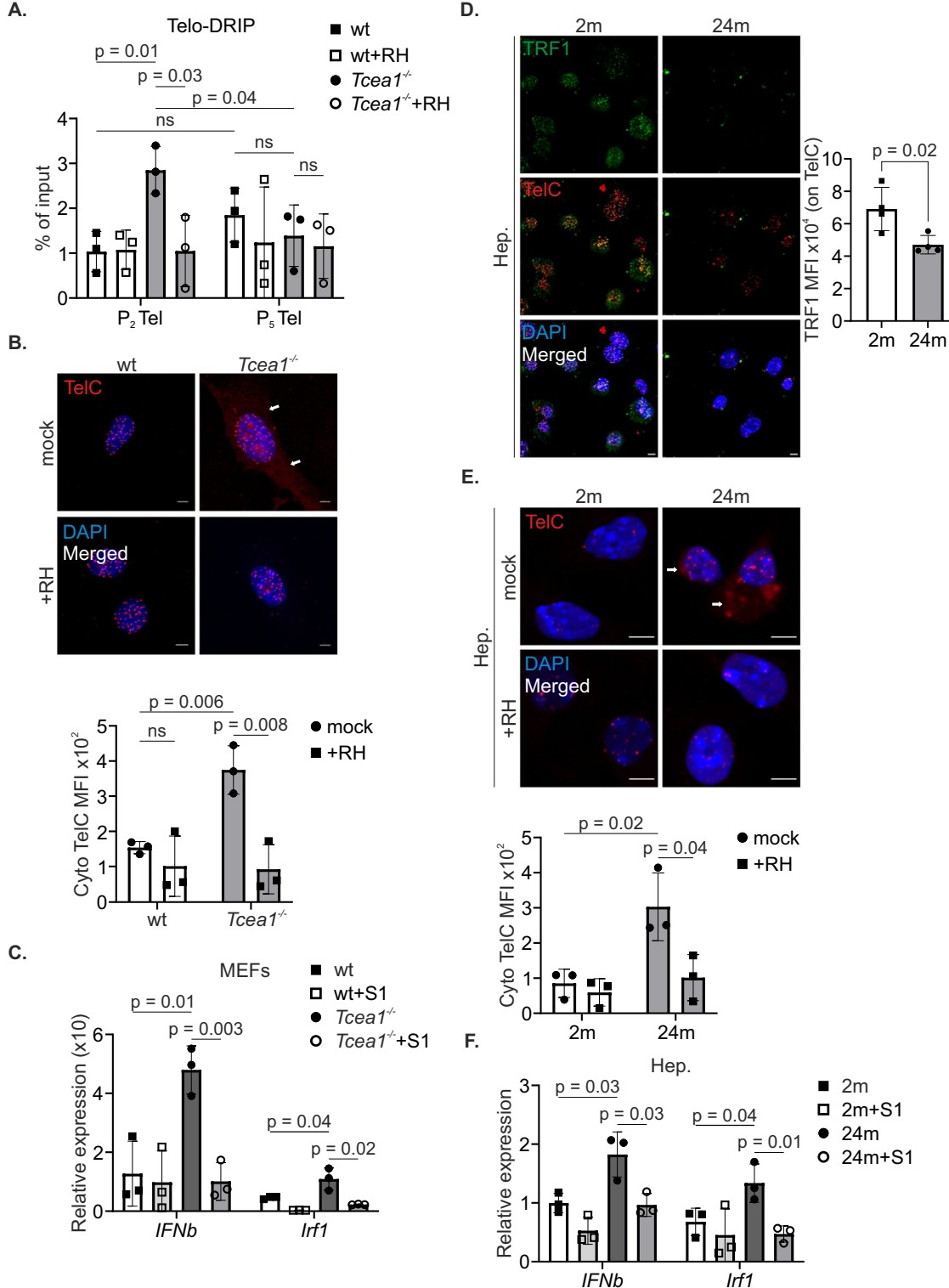

**Fig. 6 | R-loop-derived cytosolic telomeric fragments in *Tcea1*⁻/⁻ MEFs. A** DNA-RNA hybrids immunoprecipitation signals using the s9.6 antibody (shown as percentage of input after IgG normalization) on telomeres of wt and *Tcea1*⁻/⁻ MEFs, cultured for two (P₂) or five (P₅) passages (n = 3). **B** Fluorescence in situ hybridization using a Cy3-PNA TelC probe in wt and *Tcea1*⁻/⁻ MEFs, either untreated or transfected with RNase H (RH). The graph depicts the mean fluorescence intensity (MFI) of TelC in the cytoplasm of cells (n = 3). **C** *Ifnβ* and *Irf1* mRNA levels wt and *Tcea1*⁻/⁻ MEFs, untransfected or incubated with vesicle-delivered S1 nuclease (n = 3). **D** Immunofluorescence of TRF1 with in situ hybridization of telomeric DNA (TelC) in hepatocytes from 2-month- and 24-month-old mice. The graph depicts the mean

fluorescence intensity (MFI) of TRF1 on telomeres of hepatocytes (n = 4). **E** Fluorescence in situ hybridization using a Cy3-PNA TelC probe in primary hepatocytes from 2-month- and 24-month-old mice, either untreated or transfected with RNase H (RH). The graph depicts the mean fluorescence intensity (MFI) of TelC in the cytoplasm of hepatocytes (n = 3). **F** *Ifnβ* and *Irf1* mRNA levels in hepatocytes from 2-month and 24-month-old mice, untransfected or incubated with vesicle-delivered S1 nuclease (n = 3). Data analysis was performed using two-tailed Student's *t* test. All data are presented as mean values ± SEM. Unless otherwise indicated, n = biologically independent experiments and scale bars are set at 5 μm. Source data are provided as a Source Data file.

therefore, sought to investigate whether *Tcea1*[−/−] MEFs secrete EVs loaded with telomeric DNAs that, in turn, exert an effect on recipient cells. To test this, we isolated intact EVs from the culture media of *Tcea1*[−/−] and wt MEFs, labeled them with ExoFlow-ONE dye, spread them on poly-L-lysine-coated coverslips, and applied TelC-FISH to examine the presence of telomeric DNA fragments in their cargo (Supplementary Fig. 14A). Immunostaining of *Tcea1*[−/−] EVs against s9.6 antibody showed that they do not contain RNA-DNA hybrids (Supplementary Fig. 14B). The accumulation of telomeric DNA fragments in *Tcea1*[−/−] EVs was also confirmed by dot blot analyses, which had a higher TelC-FISH signal compared to EVs isolated from an equal number of wt MEFs (Fig. 7A). Of note, *Tcea1*[−/−] MEFs secrete more CD81[+]-EVs, as observed by Western blotting, compared to the same number of wt cells (Supplementary Fig. 14C). Subsequently, EVs derived from an equal number of wt and TFIIS-deficient MEFs (donor cells) were isolated and labeled with ExoFlow-ONE. The labeled EVs were then incubated with recipient wt MEFs for a duration of 10 h. Importantly, we find that a greater number of recipient cells have taken up EVs derived from *Tcea1*[−/−] cells, which carry telomeric DNA fragments, in comparison to recipient MEFs cultured with EVs from wt donor cells (Fig. 7B). Given the inflammatory and senescent phenotype exhibited by the donor *Tcea1*[−/−] MEFs, we subsequently examined the impact of this uptake on the gene expression profile of recipient cells. Our analysis revealed an increase in the mRNA levels of the pro-inflammatory *Ifnβ*, *Mx1*, *Ifitm1*, *Ifit2*, *Irf1*, *Ifi207* and *Irf7* genes in wt cells cultured with *Tcea1*[−/−] EVs, compared to cells incubated with wt EVs (Fig. 7C). Intriguingly, when the EVs were pre-treated with DNase I (Supplementary Fig. 14D), the same inflammatory response was not induced in the recipient cells as observed in cells incubated with untreated *Tcea1*[−/−] EVs (Fig. 7C and Supplementary Fig. 14E). This suggests that the inflammatory effect observed in targeted cells incubated with *Tcea1*[−/−] EVs is dependent on the presence of intact DNA within the EVs. Next, we investigated whether the pro-inflammatory property of the telomeric DNA fragments was dependent on cGAS (cyclic GMP-AMP synthase), a key enzyme involved in the recognition of cytosolic DNA and activation of the innate immune response. Our analysis revealed higher levels of cGAS that colocalized with TelC-DNA fragments in the cytoplasm of cells that had been incubated with *Tcea1*[−/−] EVs, compared to those incubated with wt EVs (Supplementary Fig. 14F). Notably, cGAS protein levels were also elevated in EVs derived from *Tcea1*[−/−] MEFs compared to wt MEFs, as confirmed by Western blotting (Supplementary Fig. 14G). Additionally, qPCR analyses showed an increase in the mRNA levels of the *Trp53* and *Rb* genes (Fig. 7D), while no induction was evident in the mRNA levels of *bcl2*, *bcl_xl*, *Casp8*, *Bax* and *Bad*, in cells incubated with *Tcea1*[−/−] EVs, compared to recipient cells with wt EVs, or cells that were incubated with DNase-treated EVs (Supplementary Fig. 14H).

Consistently, our findings demonstrated a higher number of SA-β-gal+ cells when wt MEFs were incubated with *Tcea1*[−/−] EVs compared to wt EVs or EVs pre-treated with DNase I. This suggests the induction of bystander senescence by *Tcea1*[−/−] EVs (Fig. 7E). Furthermore, cell proliferation assays using BrdU labeling revealed that incubation with *Tcea1*[−/−] EVs attenuated the proliferation rate of wt recipient cells (Fig. 7F). Taken together, our findings indicate that *Tcea1*[−/−] MEFs secrete EVs loaded with telomeric DNA fragments that can elicit an immune response and induce a senescent phenotype in targeted wt cells.

## Discussion

The role of transcription-coupled DNA repair in effectively removing DNA lesions on the actively transcribed strand of genes has been well established[49–51]. However, our understanding of the broader impact of inherent transcription machinery abnormalities on genome integrity is limited beyond conventional DNA repair processes. Overexpression of a mutant form of TFIIS leads to increased R-loop formation and DNA breaks in human cells[12]. Consistently, we find that TFIIS depletion in MEFs results in RNAPII stalling at 8-oxoGs, reduced nascent RNA synthesis, and increased formation of R-loops. These findings and the associated decline of nascent RNA synthesis seen in *Tcea1*[−/−] senescent cells are reminiscent of recent discoveries showing an age-associated transcriptional decline by accumulating DNA damage[18,52].

Unpredictably, we found that compromised transcription profoundly impacts telomere integrity. Firstly, in *Tcea1*[−/−] cells, the gradual buildup of 8-oxoGs triggers the release of TRF1 from telomeric DNA, resulting in uncapping of telomeres, activation of DDR, and chromosomal fusions. Furthermore, TRF1 interacts with TFIIS and the elongating form of RNAPII (pS2). Importantly, active transcription is implicated in TRF1 reloading onto telomeres. Therefore, even though *Tcea1*[−/−] cells are proficient in DNA repair, their TFIIS transcription defect results in the accumulation of oxidative DNA lesions[21,22] in GC-rich telomeres, which can hinder transcription. Overall, this transcription defect initiates a detrimental cycle of genome-wide RNAPII stalling, telomere dysfunction, genome instability, and persistent DDR signaling, further exacerbating the observed phenotype.

R-loops often form in GC-rich regions[53] and are important determinants of both telomere length dynamics and proliferative potential after telomerase inactivation[54]. Moreover, RNAPII stalled at a DNA lesion leads to R-loop formation and activation of the DNA damage checkpoint kinase ATM[23]. In support, TFIIS-defective cells accumulate RNase-H-sensitive R-loops at telomeres, which consequently increases DSB accumulation. The strand displacement in R-loops could potentially contribute to the telomeres' susceptibility to oxidative lesions. DNA fragments are often released in the cytoplasm, initiating a process in which the nuclear genome purges itself from extraneous DNA due to exposure to intrinsic or exogenous DNA damage[55,56]. Interestingly, we find that the presence of R-loops triggers the release of DNA fragments into the cytoplasm of *Tcea1*[−/−] cells, including telomeric repetitive sequences. These findings support previous data showing that R-loops causally contribute to the active release and build-up of single-stranded DNAs in the cytoplasm of cells carrying a defect in the structure-specific endonuclease complex ERCC1-XPF required for lesion excision during nucleotide excision repair[5,57]. Moreover, treatment with recombinant RNase H substantially reduced cytosolic telomeric DNAs, emphasizing the role of R-loops in this process. Besides telomeric DNA fragments, RNase H treatment also reduced the population of RNA-DNA hybrids in the cytoplasm of *Tcea1*[−/−] cells. The latter is in line with recent observations revealing the involvement of cytoplasmic R-loops in innate immune activation[58]. Importantly, primary cells derived from naturally aged mice exhibit a decrease in TRF1 ChIP signals at telomeres, an increase in R-loop formation, and an accumulation of cytoplasmic Tel-DNA fragments highlighting the causal role of transcription stress[18] and cytosolic DNA moieties[59] in the aging process.

We recently demonstrated that persistent DNA damage in circulating macrophages triggers the release of EVs. These EVs target and release their cargo to multiple cell types activating an innate immune response that leads to chronic inflammation[60,61]. Here, we find that cells lacking TFIIS secrete EVs loaded with cytosolic telomeric DNAs that target nearby cells. In turn, the uptake of Tel-DNA EVs induces an immune response akin to a viral infection that specifically leads to bystander senescence in the recipient cells. Thus, RNAPII stalling in a single cell type can trigger transcription stress-associated telomere dysfunction leading to a secretory phenotype that compromises the functionality of neighboring cells and promotes paracrine cellular senescence (Fig. 7G). Thus, the design of EVs carrying specialized nucleases to targeted cells loaded with cytoplasmic DNA moieties may offer a promising strategy to reduce innate immune signaling and delay transcription stress-induced chronic inflammation during aging.

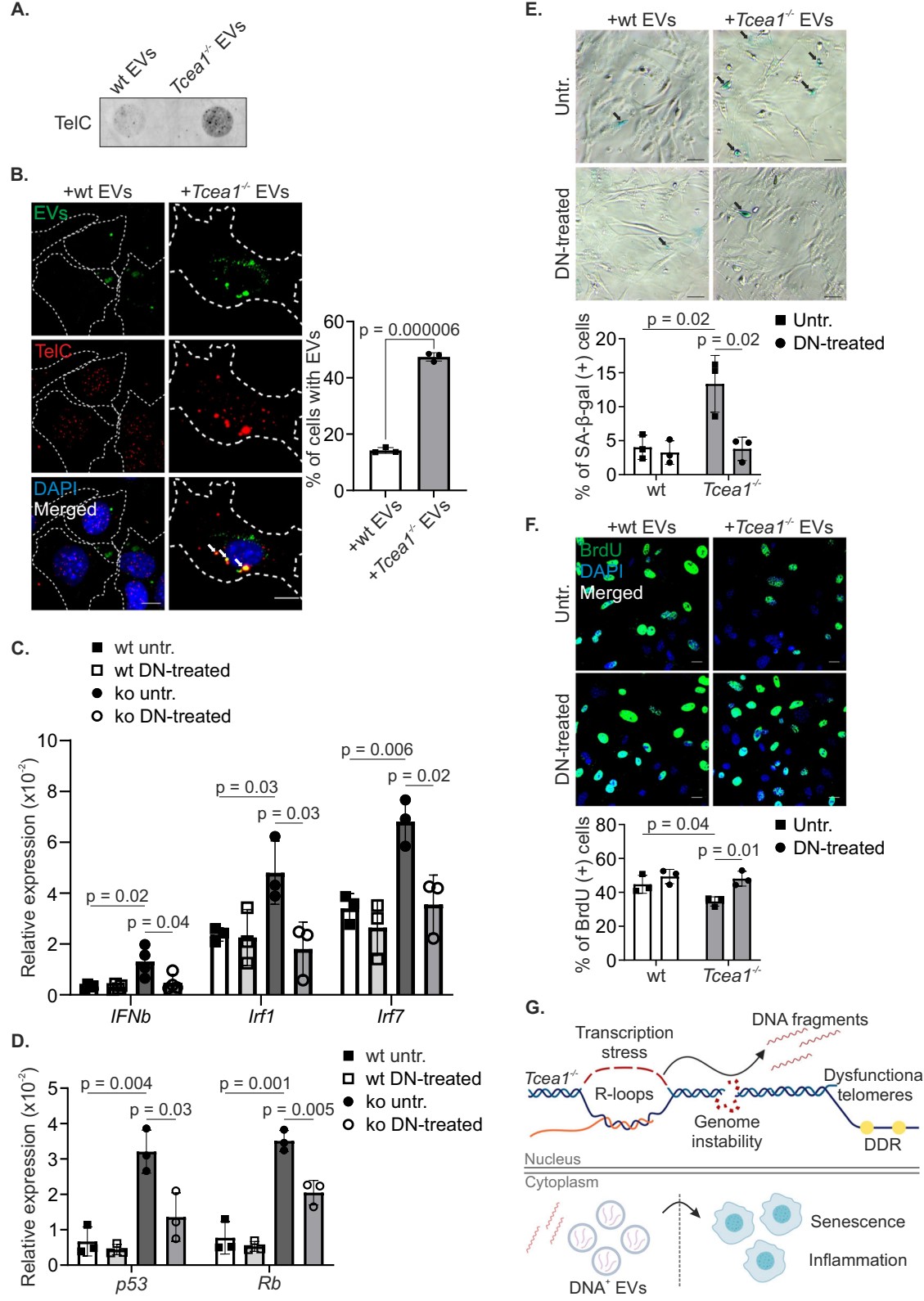

## Methods

### Animal models and primary cells

Animals were kept on a regular diet and housed at the IMBB animal house, which operates in compliance with the "Animal Welfare Act" of the Greek government, using the "Guide for the Care and Use of Laboratory Animals" as its standard. As required by Greek law, formal permission to generate and use genetically modified animals was obtained from the responsible local and national authorities. All animal studies were approved by independent Animal Ethical Committees at FORTH. All animals were housed in appropriate cages with 12 h dark and 12 h light cycle, ambient temperature and humidity. The $Tcea1^{fl/+}$ mice were generated by Cyagen. Briefly, to create a conditional $Tcea1$ knockout mouse model in C57BL/6 background, exon 3 of the $Tcea1$ gene was selected as the targeted region (Supplementary Fig. 1A). In

**Fig. 7 | *Tcea1⁻ᐟ⁻*-secreted, TelC-loaded EVs trigger bystander senescence. A** Dot blot using an Alexa488-PNA TelC probe in EVs isolated from wt and *Tcea1⁻ᐟ⁻* MEFs. **B** Immunofluorescence of ExoFlow-stained EVs with in situ hybridization of telomeric DNA (TelC) in wt and *Tcea1⁻ᐟ⁻* MEFs. White arrows indicate the presence of ExoFlow⁺;TelC⁺ EVs in the cytoplasm of cells. The white dashed line marks the cytoplasm of cells. The graph shows the percentage of wt and *Tcea1⁻ᐟ⁻* cells with ExoFlow⁺;TelC⁺ EVs (n = 3). **C** *Ifnβ*, *Irf1* and *Irf7* mRNA levels in wt MEFs incubated with untreated or DNase I-treated EVs from wt and *Tcea1⁻ᐟ⁻* MEFs (*Ifnβ* n = 4, *Irf1* n = 3, *Irf7* n = 3). **D** *p53* and *Rb* mRNA levels in wt MEFs incubated with untreated or DNase I-treated (DN-treated) EVs from wt and *Tcea1⁻ᐟ⁻* MEFs (n = 3). **E** SA-β-gal assay in wt MEFs incubated with untreated or DNase I-treated EVs from wt and *Tcea1⁻ᐟ⁻* MEFs. The graph shows the percentage of SA-β-gal⁺ cells (n = 3). Scale bar is set at

20 μM. **F** BrdU incorporation in wt MEFs incubated with untreated or DNase I-treated EVs from wt and *Tcea1⁻ᐟ⁻* MEFs. The graph shows the percentage of BrdU⁺ cells (n = 3). **G** Transcription stress in *Tcea1⁻ᐟ⁻* cells, leads to impaired transcription, accumulation of R-loops, telomere uncapping, and genome instability, ultimately resulting in cellular senescence. The formation of R-loops contributes to the release of DNA fragments in the cytoplasm of cells, which, packed in EVs can trigger an immune response in neighboring cells, which consequently undergo cellular senescence. Data analysis was performed using two-tailed Student's *t* test. All data are presented as mean values ± SEM. Unless otherwise indicated, n = biologically independent experiments and scale bars are set at 5 μm. Panel (**G**) created with BioRender.com released under a Creative Commons Attribution-NonCommercial-NoDerivs 4.0 International license. Source data are provided as a Source Data file.

the targeting vector, the Neo cassette was flanked by SDA (self-deletion anchor) sites and DTA was used for negative selection. Mouse genomic fragments containing homology arms (HAs) and conditional knockout (cKO) region were amplified from a BAC clone using a high fidelity Taq DNA polymerase and were sequentially assembled into a targeting vector together with recombination sites and selection markers. Next, C57BL/6 ES cells were used for gene targeting. To generate homozygous targeted mice (*Tcea1ᶠˡ/ᶠˡ*), heterozygous targeted mice were inter-crossed (*Tcea1ᶠˡ/⁺* x *Tcea1ᶠˡ/⁺*) and we assessed each mouse genotype by using the F2 and R2 primers, designed for the targeted allele (Supplementary Data 1 and Supplementary Fig. 1B). Then, a homozygous targeted mouse was bred with a CMV.Cre mouse to generate mice that are heterozygous for the targeted allele and carry the Cre transgene. For genotyping, we used the aforementioned F2-R2 primers and primers designed for the Cre transgene (Supplementary Data 1) and for the targeted allele after Cre-recombination (Supplementary Data 1 and Supplementary Fig. 1B). Last, heterozygous, Cre+ mice were inter-crossed with homozygous mice (Supplementary Fig. 1C) and primary MEFs were isolated from E12.5 mice embryos. All animals bearing the Cre transgene but failed to remove the targeted allele were sacrificed. MEFs were cultured in standard medium containing Dulbecco's Modified Eagle Medium (DMEM) supplemented with 10% Fetal Bovine Serum (FBS), 50 μg/ml streptomycin, 50U/ml penicillin (Sigma) and 2 mM L glutamine (Gibco). Cre recombination was also tested in *Tcea1ᶠˡ/ᶠˡ* mice crossed with Lgr5-EGFP-IRES-creERT2, Albumin.Cre, LysM.Cre and CD4.Cre and was found incomplete. All MEFs used in the experiments are derived from E12.5 wt or *Tcea1⁻ᐟ⁻* embryos at P4, unless otherwise indicated. For primary hepatocytes and pancreatic cells, tissues were digested with collagenase (2.5 mg/ml, C9263, Sigma) at 37 °C for 15 min. Collagenase was neutralized with 10% FBS and red blood cells were lysed with 1.5 M NH₄Cl, 0.1 M KHCO₃, 0.01 M EDTA for 5 min on ice. The yeast strains used are listed in Supplementary Table 1.

## Cell treatments
Cells were rinsed with PBS, treated with H₂O₂ (100 μM, 2 h; H1009, Sigma), triptolide (62 nM, 16 h; T3652, Sigma), DRB (50μM, 16 h; D1916, Sigma), Illudin S (50 ng/ml, 16 h; sc-391575, Santa Cruz), Act.D (250 ng/ml, 16 h; A1410, Sigma) and cultured at 37 °C prior to subsequent experiments. Etoposide (E1383, Sigma) was added in the cell culture media at 25 μM for 1 h. For the protein transfection experiments (Pierce Protein Transfection Reagent, 89850, Thermo Fisher Scientific), 40U of recombinant RNase H (5U/μl; M0297, New England Biolabs) was used according to the manufacturer's instructions. EVs were purified from media of 5–6 × 10⁶ cells using the differential ultracentrifugation protocol⁴⁸. Briefly, culture medium was centrifuged sequentially at 300 g (10 min), 2000g (10 min), and 10,000 × *g* (30 min) to remove dead cells and cell debris. EVs were purified with the final step of ultracentrifugation at 100,000 × *g* for 1.5 h. For functional experiments, the EVs used were purified from cells at a quantity of five times the number of recipient cells. Human

recombinant DNAse I Dornase alfa (Pulmozyme®, Roche) or 10U of S1 nuclease (100 U/μl; Thermo Fisher Scientific) were loaded using 0.2% saponin, for 20 min at RT, for the functional experiments with *Tcea1⁻ᐟ⁻* EVs or for S1 nuclease-loaded NIH-derived vesicle delivery, respectively. Following 30 min incubation at 37 °C, the EVs were centrifuged for 1.5 h at 100,000 × *g* and passed through a 0.2μm filter. Where indicated, EVs were labeled with the ExoFlow-ONE Garnet Far Red EV labeling kit for Flow Cytometry (EXOF200A-1, System Biosciences) for 20 min at 37 °C, according to the manufacturer's instructions. NAC (*N*-Acetyl-L-Cysteine, A9165, Sigma) was added in the cell culture media at 1 mM for 24 h. MitoTEMPO (SML0737, Sigma) was added in the cell culture media at 20 μM for 24 h, Mito-Tracker Red CMXRos (M7512, Invitrogen) at 0,5 μM for 1 h, and the MitoSOX Red mitochondrial superoxide indicator (M36008, Invitrogen) for 15 min at 37 °C, according to the manufacturer's instructions. For cell cycle and proliferation analyses, cells were treated with 30 μM BrdU for 1 h and with 20 μM BrdU for 10 h (B5002, Sigma), respectively. For the ubiquitin immunoprecipitation experiments, the proteasome inhibitor MG132 (M7449, Sigma) was added in the cell culture media at 5 μM for 6 h. For cell cycle synchronization during mitosis, cells were treated with 4 mM thymidine (sc-296542, Santa Cruz) for 24 h, washed with PBS and released for 9 h, and then treated with 20 ng/ml Nocodazole (sc-3518, Santa Cruz) for 4 h, before being collected by mitotic shake-off.

To generate pTFIIS and pTFIISᐞᴵᴵᴵ, the cDNA encoding the whole open reading frame (ORF) of *Tcea1* or the ORF of *Tcea1* lacking the last 128nt (Domain III) were amplified by PCR from cDNA prepared from B6 MEFs using Phusion HF DNA polymerase (NEB, MO530S) and appropriately designed primers. The PCR product was purified, cleaved with EcoR1 and inserted using T4 DNA ligase into a modified pcDNA-Flag plasmid (kind contribution of Dr. Papamatheakis), linearized with EcoRI and dephosphorylated with SAP (Promega, M820A). The construct was verified by sequencing. The primers used were as follows (restriction sites and stop codons are underlined):

pTFIISFor: GAATTCATGTGTCCCTCGGTGTGTACC,
pTFIISRev: GAATTCTCAACAGAACTTCCACCGAT,
pTFIISΔIIIFor: GAATTCATGTGTCCCTCGGTGTGTACC,
pTFIISΔIIIRev: GAATTCTCAGTCAGTCTGGGTCCCACCAG.

Overexpression experiments for TERT (pT3-EF1A-mTert, #162555, Addgene) or TFIIS/TFIISᐞᴵᴵᴵ or TERT-TFIIS, plasmids were performed by transfecting 75 × 10⁴ wt and *Tcea1⁻ᐟ⁻* MEFs with 5 μg total DNA, three times during P2-P4, using the Amaxa™ MEF 1 Nucleofector™ kit (VPD-1004, Lonza), according to the manufacturer's instructions.

Experimental elongation of telomeres was performed by repeated treatment of wt and *Tcea1⁻ᐟ⁻* MEFs with G-rich terminal oligonucleotides (GTR) over three passages (P2-P4). The oligonucleotide (TTAGGG)₄ was used at 30 μM.

## Immunofluorescence, western blot and antibodies
For immunofluorescence experiments, cells were fixed in 4% formaldehyde, permeabilized with 0.5% Triton-X and blocked with 1%

BSA. After incubation with primary antibodies for 1 h at RT, secondary fluorescent antibodies were added and DAPI was used for nuclear counterstaining. All images were acquired on a Leica SP8 confocal laser scanning microscope equipped with a 63X oil objective and captured at 2084 × 2084 pixels and the scale bar was set at 5 µm, unless otherwise indicated. For quantification analyses, the same number of 0.35µm sections were used for z-stacks of different conditions within each experiment, using the Fiji (ImageJ) software. Specifically, z-stacks were prepared by using all sections (0.35µm intervals) of a scan, from slide to coverslip. Then, for total protein fluorescence intensity measurements, we split the channels and a ROI was designed (around the whole cell or nucleus). The Integrated Density (IntDen = mean x area) was calculated by subtracting the IntDen of a "background" ROI (non-fluorescent area) from the IntDen of the fluorescent signal. For cytoplasmic measurements, the nuclear IntDen was subtracted from the whole cell. For TRF1-TelC FISH, a ROI of the size of the telomere was drawn, and TRF1 IntDen was calculated on each telomere; an average IntDen per cell was plotted. Cells with cytoplasms or telomeres overlapping on the z-axis were excluded from quantifications. For 8-oxoG immunostaining cells were prepared as previously described[62]. Briefly, they were incubated in 0.05 N HCl for 5 min on ice, washed in 1xPBS and incubated in RNAse A (100 µg/ml, 1 h, 37 °C, 740397, Macherey-Nagel) in 150 mM NaCl, 15 mM sodium citrate. After 1xPBS washes, cells were dehydrated in 35%, 50% and 75% ethanol for 3 min each, incubated with 0.15 N NaOH in 70% ethanol for 4 min, fixed in 4% formaldehyde in 70% ethanol for 2 min and incubated with Proteinase K (5 µg/ml, 10 min, 37 °C) in TE. For BrU incorporation assay, MEFs were grown on coverslips, washed with ice-cold TBS buffer (10 mM Tris-HCl, 150 mM NaCl, 5 mM MgCl$_2$) and further washed with glycerol buffer (20 mM Tris-HCl, 25% glycerol, 5 mM MgCl$_2$, 0,5 mM EGTA) for 10 min on ice. Washed cells were permeabilized with 0,5% TritonX-100 in glycerol buffer (with 25U/ml RNase inhibitor) on ice for 3 min and immediately incubated at RT for 30 min with nucleic acid synthesis buffer (50 mM Tris-HCl pH7.4, 10 mM MgCl2, 150 mM NaCl, 25% glycerol, 25U/ml RNase inhibitor, protease inhibitors, supplemented with 0,5 mM ATP, CTP, GTP and 0,2 mM BrU (850187, Sigma). After incorporation, cells were fixed with 4% formaldehyde in PBS on ice for 10 min. Incorporation was quantified with a-BrdU antibody. For cell cycle-BrdU staining, cells were fixed in methanol for 10 min and incubated in 2 N HCl for 30 min. For immunofluorescence experiments with pre-extraction, cells were seeded on coverslips, incubated with Cytoskeletal buffer (100 mM NaCl, 300 mM sucrose, 10 mM PIPES, pH6.8, 3 mM MgCl$_2$, 0.5% TritonX-100, 1 mM EGTA) for 5 min on ice, then fixed in 4% FA/1xPBS, for 10 min on ice.

For SDS−polyacrylamide gel electrophoresis (PAGE) analysis, cell pellets were resuspended in NP-40 lysis buffer (10 mM Tris-HCl, pH 7.9, 10 mM NaCl, 3 mM MgCl$_2$, 0.5% NP-40, and protease inhibitors) and incubated for 10 min at 4 °C. After centrifugation, the supernatant was kept as the cytoplasmic fraction, and pellets were resuspended in high-salt extraction buffer (10 mM Hepes-KOH, pH7.9, 380 mM KCl, 3 mM MgCl$_2$, 0.2 mM EDTA, 20% glycerol, and protease inhibitors) and incubated for 60 min at 4 °C. The supernatant after centrifugation was kept as the nuclear fraction and pellets were resuspended in RIPA buffer (1% Sodium Deoxycholate, 50 mM Tris-HCl pH7.2, 0.1% SDS, 1% NP-40 and protease inhibitors), incubated for 20 min at 4 °C and mildly sonicated. The supernatant after centrifugation was kept as the chromatin-bound fraction. For whole-cell extract preparations, cell pellets were resuspended in RIPA buffer and incubated on ice for 20 min. 80 µg of total protein were loaded for each SDS−PAGE analysis and for Co-IP input samples.

Antibodies against γH2AX (05-636, IF: 1:12000), s9.6 (MABE1095, IF: 1:100, Telo-DRIP: 5 µg), 8-oxoG (MAB3560, IF: 1:100, WB: 1:1000, oxi-DIP: 4 µg), goat anti-rabbit IgG-HRP (AP132P, WB: 1:10000), goat anti-mouse IgG-HRP (AP124P, WB: 1:5000) and mouse IgM negative control (MABC008) were from Millipore. Antibodies against γH2AX

(ab22551, WB: 1:1000), fibrillarin (ab5821, WB: 1:2500), TRF1 (ab192629, IF: 1:100), β-tubulin (ab6046, WB: 1:5000), TFIIS (ab185947, WB: 1:1000, IP: 6 µg), TRF1 (ab10579, WB:1:500, IP/ChIP: 6 µg) and pS2-PolII (ab5095, IF: 1:1000, WB: 1:1000, ChIP: 6 µg) were from Abcam. Antibodies against cleaved caspase 3 (9661, IF: 1:50, WB: 1:500), pATM (4526, IF: 1:100), α-tubulin (3873, IF: 1:2000), cGAS (31659, IF: 1:100, WB: 1:1000), CD81 (10037, WB: 1000) and H2A.Z (2718, WB: 1:1000) were from Cell Signaling Technology. Antibodies against 53BP1 (NB100-304, IF: 1:200), ATM (NB100-220, WB: 1:500) and goat anti-mouse IgM 550 (NB120-9167R, IF: 1:200) were from Novus Biologicals. Anti-BrdU antibody (555627, IF: 1:250, FACS: 1 µg/10$^6$ cells) was from BD Pharmingen. Antibodies against TOM20 (sc17764, IF: 1:50), RNAPII (sc-55492, WB: 1:500), Ubiquitin (sc-8017, WB: 1:1000, IP: 6 µg), yeast Rap1 (sc-20167, ChIP: 5 µg) and normal mouse IgG (sc-2025) were form Santa Cruz. Antibodies against TRF1 (67592-1-Ig, WB: 1:500, IP/ChIP: 6 µg), TRF2 (66893-1-Ig, WB:1:500, ChIP: 6 µg) and TIN2 (11368-1-AP, WB:1:500, ChIP: 6 µg) were from Proteintech. Antibody against pATM (200-301-400, WB: 1:300) was from Rockland. IgG from rabbit serum (I5006) was from Sigma. Goat anti-mouse IgG Alexa Fluor 488 (A-11001, IF: 1:2000, FACS: 1:250), goat anti-mouse IgG AlexaFluor 555 (A-21422, IF: 1:2000), donkey anti-rabbit IgG AlexaFluor 488 (A-21206, IF: 1:2000), goat anti-rat IgG AlexaFluor 647 (A-21247, IF: 1:2000), donkey anti-rabbit IgG AlexaFluor 555 (A-31572, IF: 1:2000) and DAPI (62247, IF: 1:20000) were from ThermoFisher Scientific. For the SA-β-gal activity, the Beta-galactosidase (β-gal) assay kit was used (9860, Cell Signaling Technology) according to the manufacturer's instructions. For s9.6 immunostainings, fixed cells were incubated with RNAse T1 (4000U, 01218429, Thermo Scientific) and RNase III (3U, AM2290, Ambion) at 37 °C for 45 min with or without RNase H[63] (20U, 5U/µl, M0297, New England Biolabs). 10U of recombinant mung bean S1 nuclease (100 U/µl; Thermo Fisher Scientific) or RNase A (1 mg/ml; Santa Cruz Biotechnology) were used for the cyto-TelC controls. Scale bars in the figures are depicted as a white line set at 5µm, unless otherwise indicated.

## Cell cycle, Annexin V − Propidium Iodide and proliferation FACS analysis
For cell cycle analysis cells were fixed with 70% ethanol for at least 1 h and incubated with 2 N HCl/0.5% Triton-100 for 30 min, RT. Cells were then resuspended in 0.1 M sodium tetraborate for 2 min, washed with PBS/1% BSA and incubated with Anti-BrdU in 0.5% Tween 20/1% BSA/PBS for 1 h, RT. After washing with PBS/1% BSA, cells were stained with anti-mouse IgG 488 for 30 min, RT and then incubated with RNase A (10 µg/ml) and propidium iodide (20 µg/ml) for 30 min, RT. The FITC Annexin V Apoptosis Detection Kit I (556547, BD Biosciences) was used for Annexin V − Propidium Iodide staining according to the manufacturer's instructions. For proliferation assay cells were stained with CellTrace CFSE Cell Proliferation Kit, for flow cytometry (C34554, Invitrogen) according to the manufacturer's instructions. A FACS Calibur (BD Biosciences) was used, and data were analyzed using the FlowJo software (Tree Star).

## Fluorescence in situ hybridization (FISH)
MEFs were seeded on coverslips and fixed in 4%PFA / 1xPBS for 10 min at 4 °C. After dehydration of the cells in increasing concentration of ethanol, 250 nM of C-rich PNA telomeric probe ((CCCTAA)$_n$ TelC-Cy3, F1002) was added to the coverslip, in hybridization buffer (20 mM Tris, pH7.4, 70% formamide 0.1 µg/ml salmon sperm DNA, 2xSSC). The cells were denatured at 80 °C for 50 min and then left at RT for 2 h. Then, cells were washed in pre-warmed 2xSSC / 0.1%Tween and nuclei were counterstained with DAPI. For Cyto-TelC the cells were hybridized in 20% Dextran sulfate, 4xSSC, 20 mM Tris, pH7.4, overnight at 37 °C. For TERRA FISH, cells were hybridized with a (TTAGGG)$_7$-Cy5.5 probe in 50% formamide, 2xSSC, 2 mg/ml BSA, 10% Dextran sulfate, 0.3 mg/ml yeast t-RNA, 0.5 mg/ml salmon-sperm DNA, 1 µg/µl mouse Cot-1, 2 mM

ribonucleoside vanadyl complexes, under non-denaturing conditions. For the negative controls, fixed cells were either incubated with RNAse A (1 mg/ml) at 37 °C for 30 min or denatured at 80 °C for 5 min. Immunostaining for TRF1 was performed as described above, cells were then washed with 1xPBS and fixed again in 4%FA / 1xPBS for 10 min, RT before hybridization. For 8-oxoG-TelC experiments, cells were stained as described above and hybridized without further denaturation.

## Metaphase chromosome spreads

Metaphase chromosomes from wt and $Tcea1^{-/-}$ MEFs were prepared according to the protocol by van Steensel et al.[64]. Briefly, cells were arrested in colcemid (0.1 μg/ml) for 20 h, harvested by trypsinization, incubated for 15 min at 37 °C in 75 mM KCl, and fixed in freshly made methanol/acetic acid (3:1). Cells were dropped onto wet slides and air-dried overnight in a chemical hood. Fluorescence in situ hybridization was performed as described above. For NHEJ inhibition, 14 μM SCR130 inhibitor (37779, Cayman) was added for 24 h, as indicated. For γH2Ax immunostaining, metaphases were prepared according to Pedersen et al.[65], stained against γH2Ax for 2 h, fixed with 50 mM ethylene glycol bis (succinimidyl succinate) (EGS, 21565, Thermo Scientific) for 3 min and then labeled with TelC-PNA probe.

## ChIP, Oxi-DIP, Telo-DRIP and Co-immunoprecipitation assays

For co-immunoprecipitation assays, nuclear protein extracts from primary MEFs were prepared as previously described[66], using the high-salt extraction method (10 mM HEPES-KOH pH7.9, 380 mM KCl, 3 mM MgCl2, 0.2 mM EDTA, 20% glycerol and protease inhibitors). Nuclear lysates were diluted three-fold by adding ice-cold HENG buffer (10 mM HEPES-KOH pH7.9, 1.5 mM MgCl2, 0.25 mM EDTA, 20% glycerol) and precipitated with antibodies overnight at 4 °C followed by incubation for 3 h with protein G Sepharose beads (Millipore) or directly with BS3 (Thermo) cross-linked antibodies on Protein A Dynabeads (Thermo) at 4 °C overnight. Normal mouse, rabbit or goat IgG (Santa Cruz) was used as a negative control. Immunoprecipitates were washed five times (10 mM HEPES-KOH pH7.9, 300 mM KCl, 0.3% NP-40, 1.5 mM MgCl2, 0.25 mM EDTA, 20% glycerol and protease inhibitors), eluted and resolved on 8-12% SDS-PAGE.

For ChIP assays, primary cells (MEFs) were crosslinked at RT for 2.5 min with 1% formaldehyde. Chromatin was prepared and sonicated on ice for 5-8 min using Covaris S220 Focused-ultrasonicator. Samples were immunoprecipitated with antibodies (6 μg) overnight at 4 °C followed by incubation for 3 h with protein G sepharose beads and washed sequentially. The complexes were eluted, and the crosslinking was heat reversed. Purified DNA fragments were analysed by qPCR using sets of primers targeting telomeric repeats as previously described[67]. For yeast Rap1 ChIP, cells were grown in exponential phase and cross-linked with 1.2% formaldehyde for 10 min at RT and quenched with 115 mM glycine for 5 min. The samples were centrifuged at 4 °C and washed two times with ice-cold 1XPBS. Pellets were resuspended in lysis buffer (50 mM HEPES-KOH pH7.5, 140 mM NaCl, 1 mM EDTA pH8, 1% Triton X-100, protease inhibitors), transferred in lysing Matrix C tubes (MP Biomedicals) and lysed by using the FastPrep machine for 3 times (30 sec for each run; MP Biomedicals) at 4 °C at 6.5 M/sec, with 1 min pause. Addition of lysis buffer with sodium deoxycholate (SOD) supplemented with protease inhibitors (Roche) recovered the extracts. After centrifugation for 15 min at 4 °C, the pellets were resuspended again in 1.5 ml lysis buffer + SOD + protease inhibitor and then mixed with 20 μl 20% SDS. 750 μl of the resulting mix were combined with 0.4 g beads and sonication was performed for 5 cycles of 30 sec on/off at 4 °C using Bioruptor Pico (Diagenode). The remaining mix of each sample was also combined with 0.4 g beads and sonicated. Protein concentration from each ChIP extract was measured by Bradford and diluted to 1 mg/ml in lysis buffer + SOD + protease inhibitor. 50 μl of the mix were stored at −20 °C representing 5%

of input. The remaining volume was split in two parts. One half was used to pull down the protein in presence of the appropriate antibody and the other half as a negative control (without antibody). After antibody addition, incubation for 30 min at 4 °C on rotating wheel was followed. Next, 50 μl of Dynabeads Protein G beads (Invitrogen), were washed and supplemented with 5% BSA in order to be added to the samples which subsequently were incubated overnight at 4 °C on rotating wheel. Beads were washed with 1 ml lysis buffer + SOD, 1 ml lysis buffer 500 (500 nM NaCl added to lysis buffer), 1 ml cold buffer III (10 mM Tris-HCl pH 8, 0.1 mM EDTA pH8, 250 mM LiCl, 1% NP-40 and 1% SOD) and 1 ml TE pH 8.0. Afterwards, the samples were eluted twice in 100 μl elution buffer B (50 mM Tris-HCl pH7.5, 1% SDS and 10 mM EDTA pH8).

Oxi-DIP experiments were performed as previously described[68]. Briefly, genomic DNA from growing MEFs was extracted by using Dneasy Blood & Tissue kit (69504, QIAGEN). 10 μg of genomic DNA per immuno-precipitation were sonicated using Covaris S220 Focused-ultrasonicator, in TE buffer (10 mM Tris–HCl pH8.0, 1 mM EDTA pH8.0) to generate random fragments ranging in size between 200 and 800 bp. 4 μg of fragmented DNA were denatured for 5 min at 95 °C in IP buffer (110 mM NaH2PO4 pH7.4, 110 mM Na2HPO4 pH7.4, 150 mM NaCl, 0.05% TritonX-100, 10 mM Tris–HCl pH8.0, 0.1 mM EDTA pH8.0) and immunoprecipitated overnight at 4 °C under constant rotation. The immunoprecipitated complex was incubated with protein G Sepharose beads, previously saturated with 0.5% bovine serum albumin diluted in IP buffer for 3 h at 4 °C, under constant rotation, and washed three times with 1 ml washing buffer (110 mM NaH2PO4, 110 mM Na2HPO4 pH7.4, 150 mM NaCl, 0.05% TritonX100). The beads–antibody–DNA complexes were then disrupted by incubation with elution buffer (50 mM Tris-HCl pH8.0, 10 mM EDTA pH8.0, 1% SDS, 0.5 mg/ml proteinase K) overnight at 37 °C, and 1 h at 52 °C. The immunoprecipitated DNA was collected and purified by using MinElute PCR Purification kit (28004, QIAGEN). Purified DNA fragments were analysed by qPCR using sets of primers targeting telomeric repeats as previously described[67]. All the steps of OxiDIP-protocol were carried out in low-light conditions.

Telo-DRIP experiments were performed as previously described[67]. Briefly, MEFs were harvested by scraping and lysed in 1 mL Tris-EDTA, 0.5% sodium dodecyl sulfate (SDS), and 200 μg/mL proteinase K (25530049, Invitrogen) in presence of RNAseOUT (10777019, Invitrogen) overnight at 37 °C at 350 rpm. Nucleic acids were extracted with phenol/chloroform/isoamyl alcohol (25:24:1 saturated with 10 mM Tris-Cl pH 8.0 and 1 mM EDTA). The samples were precipitated with ethanol, and pellets were spooled out and washed in 70% ethanol. Chromatin was resuspended in Tris-EDTA-RNAseOUT and sonicated with Covaris S220 Focused-ultrasonicator. DNA was quantified, and 10 μg was either treated with 20U RNAse H (M0297, NEB) or mock-treated for 5 h at 37 °C at 350 rpm. 10 μg of chromatin was incubated with 5 μg of S9.6 antibody (MABE 1095, Millipore) in binding buffer (10x: 100 mM NaPO4 pH7.0, 1.4 M NaCl, and 0.5% Triton X-100, diluted in Tris-EDTA-RNAseOUT) on a rotating wheel overnight at 4 °C. Before adding the antibody, 1/20 of the volume of the reactions was collected as the input. Immunocomplexes were isolated by incubation with protein G Sepharose beads for 2 h at 4 °C on a rotating wheel. The beads were washed twice in binding buffer and were incubated in elution buffer (50 mM Tris pH8, 10 mM EDTA, 0.5% SDS) containing 80 μg/mL RNase A (19101, Qiagen) for 30 min at 50 °C at 350 rpm. Elution was performed by adding 560 μg/mL proteinase K and incubating for 45 min at 55 °C at 750 rpm. The supernatants were recovered, and the DNA was precipitated and resuspended in Tris-EDTA. Purified RNA-DNA fragments were analysed by qPCR using sets of primers targeting telomeric repeats, as previously described[67]. ChIP, Oxi-DIP and Telo-DRIP signals in the figures are shown as % of input after normalizing with IgG and IgM negative corresponding controls.

## Yeast colony Telo-PCR

From overnight cultures, 0.2 OD units were collected and resuspended in 20 μl 0.02 M NaOH for genomic DNA extraction at 100 °C for 10 min. After spin down, 2 μl of the supernatant were transferred into 3 μl of water, incubated at 96 °C for 10 min and cooled down at 4 °C. Next, addition of 5 μl of C-tailing mix, including 0.2 μl terminal transferase (20U/μl; NEB), 0.1 μl 10x NEBuffer 4, 0.1 μl 10 mM dCTP and 3.7 μl water took place followed by incubation at 37 °C for 30 min, 65 °C for 10 min and 96 °C for 5 min. Afterwards, the samples were kept at 65 °C until 30 μl of PCR-master mix, with 21 μl H$_2$O, 4 μl 10x PCR buffer (670 mM Tris–HCl pH8.8, 160 mM (NH$_4$)$_2$SO$_4$, 50% glycerol and 0.1% Tween-20), 4 μl dNTPs (2 mM stock), 0.3 μl forward subtelomeric oligo (100 μM stock) and 0.3 μl G$_{18}$ reverse oligo oBL359 (100 μl stock), were added to each tube. Then, samples were heated at 65 °C until 0.5 μl of Q5 Hot Start DNA Polymerase (2U/μl; NEB) were added to the mixes and thermocycling was performed as follows: 95 °C for 3 min, 45 cycles of 95 °C for 30 s, 63 °C for 15 s and 68 °C for 20 s. At the end, the samples were held at 68 °C for 5 min. The resulted PCR products were separated on a 1.8% agarose gel for 50 min at 150 V. Visualization of the bands was performed by using ChemiDoc™ Touch Imaging System (BioRad) and telomere length for each sample was determined by applying ImageLab (BioRad) software.

## DNA and RNA dot blot

For genomic DNA (2 μg) or ChIP samples, 0.4 M NaOH and 10 mM EDTA, pH8.2 was added for denaturation, followed by incubation at 95 °C for 10 min. The samples were then loaded on an Amersham HyBond-XL membrane (RPN203S, GE Healthcare) using a dot-blotting manifold and the membrane was UV-crosslinked (125 mJoule/cm$^2$ at 254 nM) before SDS-PAGE or hybridization. For s9.6 blotting, genomic samples were not denatured. For EV DNA, EVs from 20×10$^6$ cells were incubated with Proteinase K (0.1 μg/μl) in 0.5% SDS, 50 mM Tris, pH8, 100 mM EDTA at 56 °C for 3 h, DNA was precipitated, denatured with 0.4 M NaOH and 10 mM EDTA, pH8.2, at 95 °C for 10 min and loaded on the membrane. For RNA samples (5 μg), total RNA was incubated at 65 °C for 20 min, in 50% formamide, 7% formaldehyde, 1xSSC before loading on the membrane. For negative controls, RNA was incubated with RNAse A (100 ng/μl), RNAse T1 (1U/μl) and RNAse H (1U/μl) for 1 h at 37 °C. Hybridization for TelC-DNA (37 °C), TERRA and 18SRNA (65 °C) was performed overnight in TelC hybridization buffer as described above (Supplementary Data 1).

## Southern blot

Genomic DNA (300 ng) was digested with 1 μl restriction enzyme RsaI, 1 μl HinfI in 2.5 μl CutSmart buffer (NEB) for 5 h at 37 °C. After digestion, the samples were mixed with 1x loading dye and loaded on a 0.8% pulse-field agarose (Bio-Rad) gel and run on a CHEF-DRIII pulse field electrophoresis apparatus (Bio-Rad). The following electrophoresis conditions were applied to the system: initial pulse 0.5 s, final pulse 15 s, voltage 6 V/cm, run time 20 h. After migration, the DNA was denatured in 0.4 M NaOH, 0.6 M NaCl for 1 hr and neutralized with 1 M Trizma Base, 1.5 M NaCl (pH7.4) for 1 h. The DNA was transferred onto a positively charged nylon membrane (Sigma) in 10xSSC, the membrane was dried and UV-crosslinked (120 mJ/cm$^2$, AnalytikJena). Hybridization took place at 47.5 °C and 15 rpm in 5xSSC, 1x Denhardt's solution, 5 mM EDTA, 2.5 mg/ml yeast RNA, 0.1% SDS, for 1 h. The telomeric (CCCTAA)$_4$ probe was labeled with digoxigenin-ddUTP according to manufacturer's instructions (DIG Oligonucleotide 3′-End Labeling Kit, 2nd Generation, Roche) and incubated with the membrane overnight at 47.5 °C. The membrane was washed twice for 5 min in 10 ml 2xSSC + 0.1% SDS at 47.5 °C, followed by two washes in 10 ml 0.5xSSC + 0.1% SDS for 20 min at 47.5 °C. Next, it was rinsed in 1xDIG wash buffer (0.1 M maleic acid pH7.5, 150 mM NaCl, 0.3% Tween-20) and blocked for 30 min in 0.1 M maleic acid pH7.5, 150 mM NaCl, 1% (w/v) Blocking Reagent (Roche). 1:5000 Anti-Digoxigenin-AP Fab fragments (Roche) was added for 30 min at RT. The membrane was washed 4 times for 15 min in 1xDIG wash buffer, followed by 5 min in DIG detection buffer (0.1 M Tris–HCl pH9.5, 0.1 M NaCl). 4 ml of CDP-Star® (Roche) were distributed directly onto the membrane and incubated in the dark for 5 min. The blot was visualized in a ChemiDoc touch™ Touch Imaging System (BioRad).

## Yeast RNA extraction

Overnight cell cultures were grown to exponential phase in 15 ml of YPD at 30 °C. After centrifugation at 17000 × g, cells were resuspended in 400 μl AE-buffer and mixed with 20 μl 20% SDS and 500 μl pre-equilibrated phenol. Incubation for 5 min at 65 °C and subsequently on ice for 5 min were followed. Afterwards, centrifugation at 4 °C for 3 min at 17000 × g was performed to collect the supernatant and mix it with 500 μl phenol-chloroform (1:1). The samples were precipitated with 3 M NaAc and 100% ethanol and washed with 80% ethanol. DNase treatment followed for 1 hr at 37 °C in RDD buffer with DNase I (Qiagen). Qiagen RNeasy Min Elute Cleanup Kit (Qiagen) was used, and RNA was eluted in 30 μl of water. To purify TERRA RNA, 50 μg total RNA were cleaned up 3 consecutive times with the RNeasy kit (Qiagen) based on the manufacturer's instructions. Incubation in RDD buffer-DNase I took place between each clean-up step. Finally, RNA was eluted in 30 μl H$_2$O.

## BLESS

To map DNA DSBs genome-wide, we applied an adapted set-up of the Breaks Labeling, Enrichment on Streptavidin and next-generation Sequencing (BLESS) protocol according to Crosetto et al.[69]. The procedure includes the in situ blunting of DSB ends, after mild fixation of the cells, and ligation to specialized biotinylated BLESS adapters, bearing the RA5 Illumina RNA sequence, that allow the selective affinity capture of processed DSBs. Upon ligation of the biotinylated adapter on DSBs, genomic DNA is purified and sonicated. Then, streptavidin beads (Dynabeads MyOne C1 #65001) are used to isolate DSB-bearing DNA fragments, followed by blunting of the other end and ligation to a second BLESS adapter containing the RA3 Illumina RNA adapter sequence. PCR amplification was performed according to Illumina's guidelines, for 10 cycles using the RA5 and RA3 adapters, followed by purification and specific-target qPCR amplification. The adapter sequences were previously reported for BLESS[69]. For the RA3, RA5 adapters, RTP primer, and RP1 and RPIX primers, see the sequence information available for the Illumina small RNA library preparation kit.

## RNA-seq and quantitative PCR studies

Total RNA was isolated from cells using a Total RNA isolation kit (Qiagen) as described by the manufacturer. For RNA-Seq studies, libraries were prepared using the Illumina® TruSeq® mRNA stranded sample preparation kit. Library preparation started with 1 μg total RNA. After poly-A selection (using poly-T oligo-attached magnetic beads), mRNA was purified and fragmented using divalent cations under elevated temperature. The RNA fragments underwent reverse transcription using random primers. This is followed by second strand cDNA synthesis with DNA Polymerase I and RNase H. After end repair and A-tailing, indexing adapters were ligated. The products were then purified and amplified (14 PCR cycles) to create the final cDNA libraries. After library validation and quantification (Agilent 2100 Bioanalyzer), equimolar amounts of library were pooled. The pool was quantified by using the Peqlab KAPA Library Quantification Kit and the Applied Biosystems 7900HT Sequence Detection System. The pool was sequenced by using a S2 flowcell on the Illumina NovaSeq6000 sequencer and the 2x100nt protocol. Reverse transcription and qPCR amplification of mRNAs was performed as previously described[59]. For TERC, Reverse Transcription (RT) was performed as it was previously described by Tang et al.[70]. For mouse

TERRA, RT was performed as it was previously described by Feretzaki & Lingner, 2017[71]. For yeast TERRA RT, 3 µg RNA were mixed with 0.4 µl dNTPs (25 mM stock), 1 µl oBL207 (10 µM stock), 0.4 µl oBL293 (10 µM stock), heated for 1 min at 90 °C and then gradually (ca. 0.8 °C/sec) reached 55 °C. A mix containing 1 µl DTT (0.1 M stock, Invitrogen), 1 µl SuperScript III Reverse Transcriptase (Invitrogen), 1 µl RNaseOUT (Invitrogen) and 4 µl First Strand buffer (5X stock, Invitrogen) was added and samples were incubated for 1 hr at 65 °C and then for 15 min at 70 °C. Then, 2 µl cDNA was mixed with 1 µl each primer (10 µM stock), 1 µl water, 5 µl SYBR-Green. The qPCR settings were the following: 10 min at 95 °C; 40 cycles of 15 sec at 95 °C and 1 min at 60 °C; 5 min at 95 °C; 1 min at 65 °C. The melting curve of the amplicon was determined with a gradient from 65 °C to 97 °C with an increase of 0.5 °C/cycle in cycles of 5 sec. The reaction was performed on the CFX384 Touch Real-Time PCR Detection System (Bio-Rad) and for data analysis CFX ManagerTM software (Bio-Rad) was used. Ct values were determined automatically in a regression mode. Quantitative PCR (Q-PCR) was performed with a Biorad 1000-series thermal cycler according to the instructions of the manufacturer (Biorad). All oligonucleotides used in this study can be found in Supplementary Data 1.

## Cytoseq

Cytoplasmic DNA fragments were phenol-chloroform purified from wt and $Tcea1^{-/-}$ cytoplasmic extracts, isolated with NP-40 lysis buffer, as in the immunoprecipitation protocol. The xGenTM ssDNA & Low-Input DNA Library Preparation Kit (10009859, Integrated DNA Technologies) was used. After library validation and quantification (Agilent 2100 Bioanalyzer), equimolar amounts of library were pooled. The pool was sequenced on an Illumina NovaSeq6000 sequencer and the 2x150nt protocol.

## Data and statistical analysis

All graphs were generated with the GraphPad Prism version 9.0.0 (121). For the RNA-Seq data analysis, the data were downloaded as FASTQ files. Their quality was checked with the use of a FASTQC quality control tool for high throughput sequence data: http://www.bioinformatics.babraham.ac.uk/projects/fastqc. The data were aligned via STAR aligner to the mm10 genome assembly (GENCODE). Count normalization was performed with the edgeR algorithm. DESeq, edgeR and NBPSeq algorithms were used in order to perform the statistical analysis. The differentially expressed genes were identified with the use of metaseqR2: https://bioconductor.org/packages/release/bioc/html/metaseqR2.html. Threshold levels of p < 0.05, and FC ≥ ± 1.3 were used to estimate the significance of differences. Gene ontology (GO) enrichment analysis was determined by Gene Ontology: http://geneontology.org and Panther Classification System was used for the identification of enriched/over-represented GO terms. The analysis was repeated twice and none of the samples were excluded from the final experiment.

Quality control was performed with FastQC. TrimmomaticPE was used as a trimming tool. Alignment was performed with the use of STAR aligner on both the mouse reference genome (gencode release M33) and the human reference genome (gencode release 44). Motif identification was performed with the use of blastn (blast.ncbi.nlm.nih.gov). Visual representation was performed with the use of interactive genomics viewer (IGV 2.4.14).

Error bars in the figures indicate S.E.M. among n ≥ 3 biological replicates. Asterisk indicates the significance set at p-value: *≤0.05, **≤0.01, ***≤0.001 (two-tailed Student's *t* test).

## Reporting summary

Further information on research design is available in the Nature Portfolio Reporting Summary linked to this article.

## Data availability

The RNA-Seq and the Cytoseq data have been deposited in the European Nucleotide Archive (ENA) (https://www.ebi.ac.uk/ena/browser/home) with the accession number PRJEB63556 (RNA-Seq) and PRJEB71474 (Cytoseq). Original western blot images and raw data underlying graphs are provided in the Source data file. Any additional information required to reanalyze the data, resources and reagents including the pcDNA-FLAG-mTFIIS and mTFIIS^AIII plasmids reported in this paper is available from the corresponding author upon request. Source data are provided with this paper.

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

## Acknowledgements

The Horizon 2020 ERC Consolidator grant "DeFiNER" (GA 64663); the ERC PoC "Inflacare" (GA 874456); the Horizon 2020 Marie Curie ITN "aDDRess" (GA 812829), and "HealthAge" (GA 812830), ELIDEK grants 631, 6204 and 1059; the "Research-Create-Innovate" actions (MIA-RTDI) "Panther"–00852 and "Liquid Pancreas"–00940; Uni-Pharma Kleon Tsetis Pharmaceutical laboratories S.A (PAR00838) and Pharmathen S.A. (PAR00863) funds; and Greece 2.0, National Recovery and Resilience Plan Flagship program TAEDR-0535850 supported this work. A.A.-C. was supported by the ELIDEK Fellowship 6204.

## Author contributions
Conceptualization: G.A.G., K.S. and A.S. Methodology and Investigation: A.S., K.S., D.G., G.C., A.A.C and E.G. Visualization: A.S., K.S., D.G. and A.A.C. Supervision: G.A.G. Writing-Original draft: G.A.G., K.S. and A.S. Writing-Review and editing: G.A.G., K.S., A.S., B.L. and B.S.

## Competing interests
The authors declare no competing interest.
