## [Peer Review File · Nature Communications]

Transcription stress at telomeres leads to cytosolic DNA release and paracrine senescenceREVIEWER COMMENTS

Reviewer #1 (Remarks to the Author):

In this study the authors examine MEFs lacking the TFIIS factor (Tcea1^{-/-}), which is critical for the resumption of transcription after RNA Polymerase II stalling during elongation. The authors find the Tcea1 ko MEFs show multiple hallmarks of premature senescence and oxidative stress. The knock cells also show higher amounts of staining for 8-oxoguanine, R-loops, and DDR positive telomeres, and reduced BrU staining indicative of reduced transcription, by immunofluorescence. The authors also provide data showing the colocalization of 8-oxoguanine foci with telomeres, and a decrease in TRF1 at telomeres in the knock-out cells. The Tcea1^{-/-} cells and cells from older animals also exhibit cytoplasmic DNA fragments and extracellular vesicles that contain telomeric DNA, and the EVs from Tcea1 knockout cells can induce senescence in wild type cells. Based on the data the authors propose a model that RNA Polymerase II stalling at 8-oxoguanine in telomeres can cause telomere dysfunction-induced senescence, and the release of telomeric DNA in the cytoplasm, stimulating the senescence associated secretory phenotype that triggers paracrine senescence. The manuscript describes a great deal of phenotypic data from Tcea1^{-/-} MEFs related to transcription and genomic and telomere instability, but a demonstration of causal relationships is less strong. While the focus on telomeres is interesting, it is less clear how R-loops elsewhere or RNA PolII stalling at sites outside the telomeres contribute to senescence and the inflammatory response in Tcea1^{-/-} cells. In Figure Tcea1^{-/-} cells show a strong reduction in BrU staining, and an increase in 8-oxoguanine staining, throughout the entire nucleus. Specific comments are below.

1. Fig 2A. Is the 8-oxoguanine staining outside the nucleus mitochondrial? There appears to be a great deal more 8-oxoguanine staining outside the nucleus in Tcea1 ko cells in Supplementary Figure 3C, compared to the images main Fig 2A. If so, then it appears that the abundance of 8-oxoguanine may be higher in the mitochondria than in the nucleus. More detail on how MFI was obtained would be helpful. How were thresholds set?
2. Supplementary Fig 2C shows that a mitochondrial ROS scavenger decreases the 8-oxoguanine staining in the nucleus. Can this scavenger (MitoTEMPO) also suppress the other phenotypes observed in the Tcea1^{-/-} cells, such as increased R-loops, decreased BrU staining, and increased pS2-PolII? In other words, if the oxidative stress arising from dysfunctional mitochondria causes the RNA PolII stalling at 8-oxoguanine lesions in telomeres, this should be ameliorated with MitoTEMPO.
3. In Fig 2C how was the ChIP for pS2-PolII controlled for the higher levels of this phosphorylated form of RNAPII in Tcea1^{-/-} cells? Could this explain the increase in 8-oxoG staining in the associated DNA?
4. Previous reports indicate that imaging the S9.6 antibody is subject artifacts (Smolka et al., JBC, 2021). However, the JBC publication provided methods to control for these artifacts. The authors should indicate how these controls were used to rule out artifactual staining in Fig. 2F.
5. The gammaH2AX and 53BP1 foci are not obvious in the images in Fig 2G. An enlargement or "zoom in" of a representative nucleus would help show these foci better. Addition of RNase H appears to decrease gammaH2AX staining, but the effect on 53BP1 is less obvious. GammaH2AX also marks sites of stalled replication forks. Also, the authors imply that the DSBs arise from R-loops, which expose long stretches of ssDNA that are susceptible to breaks. Therefore, a marker of ssDNA is warranted here. For example, pulse with BrdU, followed by anti-BrdU staining under non-denaturing conditions.
6. The conclusion that TFIIS loss ultimately leads to DSBs would be strengthened greatly by including an assay that tests directly for DSBs, such as the Comet assay, rather than only 53BP1 foci staining. Are these breaks only at telomeres?

7. It is unclear how 8-oxoguanine might inhibit telomere transcription in MEFs since the C-rich strand is transcribed.

8. The loss of TRF1 (as shown later) is expected to cause telomere fragility based on evidence from the literature (first reported in Sfeir et al, 2009, Cell). Did the authors see any evidence of fragile telomeres on the metaphase spreads from Tcea1 ko MEFs? (Fig 3F).

9. In Fig 3F there is a discrepancy between the legend and the description of the results in the text. It is not clear if the percentage of telomere fusion events per metaphase, or the percent of metaphases with at least one telomere fusion event is shown. In this latter case, a more accurate label for the y-axis might be "% of metaphases with ≥ 1 fusions".

10. The conclusion that "defect in TFIIS results in shorter telomeres that become fused, leading to anaphase bridges and mitotic aberrations", does not appear to be well supported by the data. The small reduction in telomere length in Tcea1 ko MEFs by telomere FISH and S blot is not expected to lead to telomere fusions. The telomeres are still quite long in the Tcea1 ko MEFs (between 38 and 48kb).

11. Is the reduction of TRF1, TRF2 and TIN2 localization to telomere by ChIP and/or by IF due to a reduction in protein levels? Is transcription of TRF1, TRF2 and TIN2 impacted by the loss of Teac1?

12. Was the TFIIS complex with TRF1 and RNAPII shown by coIP in Fig 5D (and 5E for TRF1 and pS2-RNAPII) mediated by interaction on DNA?

13. The results and conclusion in Fig 5F are confusing. The decrease in BrU signal (overall and at telomeres) in Tcea1 ko cells suggests transcription is impaired in the ko cells, and at telomeres. But why then do the authors not detect a decrease in TERRA?

14. For Fig 5G, the authors allowed the cells to recover for 16 h after H₂O₂ treatment, which restores TRF1 localization. However, this restoration is impaired if the authors add a drug that inhibits transcription elongation. They interpret this result as evidence that TRF1 recruitment to the telomeres requires active transcription. However, this seems at odds with evidence from the literature that TRF1 is recruited to telomeres by its Myb telomeric DNA binding domain. Furthermore, not all telomeres are typically transcribed in most cell types. Does DRB addition during the 16 hour recovery affect overall transcription and protein levels of TRF1?

15. In Fig. 6B could the TelC probe be recognizing TERRA in the cytoplasm? The C-rich probe should bind to the G-rich TERRA transcripts.

16. In Fig 6C could the micronuclei (Fig 3A) be driving the inflammatory response? In Fig. 6D is TRF1 expression lower in aged mice (24 months). As telomeres shorten with age the binding sites for TRF1 are also reduced. Fig 6E, it is curious that the cytoplasmic telomeric staining does not overlap with DAPI. Is this because the authors suspect these foci are R-loops that may not stain with DAPI? Do they overlap with S9.6 staining? How do the authors know that the cytoplasmic telomeric staining in cells from the aged mice is causing the inflammatory response?

17. In Fig 7 it is interesting that the EVs from Tcea1 ko MEFs show greater telomeric DNA staining. But it is unclear if the telomeric fragments, or R-loop containing telomeric DNA, are responsible for the inflammatory response and senescence induction in recipient cells.

Reviewer #2 (Remarks to the Author):

In their paper Siametis et al., demonstrate using *Tcea1*^{-/-} mouse embryonic fibroblasts that cells with defect in transcription elongation exhibit senescence phenotype due to accumulation of 8-oxoG lesions throughout the genome, stalling of RNAPII at oxidative DNA damage sites, impaired transcription, accumulation of R-loops throughout the genome, DNA damage response and genomic instability. They also claim that TFIIS deficiency leads to telomere shortening and fusions, due to telomere uncapping and DNA damage activation at telomeres; and that R-loops at telomeres contribute to release of telomeric DNA fragments in the cytoplasm of *Tcea1*^{-/-} cells and primary cells derived from naturally aged animals triggering a viral-like immune response.

Although the findings presented are of interest, I have numerous methodological issues with the data presented in the paper, which require careful evaluation. Also I have doubts in telomere-specific origin of the genomic instability and senescence in case of TFIIS deficiency, given that only a minor fraction of telomeres are fused, authors see increase of R-loops in the control loci as well as at telomeres, and according to their model immune response is triggered by ssDNA released from R-loops.

Methodological issues:

Authors base their story on a variety of microscopy and immunoprecipitation-based techniques to generate the data, and these methods and data analyses required have long been established in the field. Unfortunately, data presented in the paper often don't meet the standards.

Golden standard for analyzing ChIP and DRIP data is showing the percentage of input DNA recovered with IP after subtraction of the background signal from IgG pulldowns. Normalization to background noise (IgG) is misleading first of all because it gives higher numbers, and second, it does not account for differences in telomeric DNA content in different samples (meaning telomere length, which authors claim to be different between WT cells and *Tcea1*^{-/-} cells)

Protein co-IP experiments need to be quantified or at least labeled with fraction of input/fraction of IP loaded.

Lots of imaging experiments show pan-nuclear staining, or background in the cytoplasm. Given that authors rely on co-localization of foci to draw their conclusions, there is a need in a detailed description of how data were analyzed and ideally z-stack analysis of confocal pictures. Lens used should be specified in the materials and methods section, and the standard for for TIFs and other co-localization experiments in the field is x63 or x100. All images lack the scale of the scale bar and information about magnification.

All the 8-oxo-G IF images have extremely high background which makes it very difficult to accept the conclusions from co-localization studies. Cells are already pre-treated with RNaseA, where does this background come from? Could pre-extraction help getting rid of the non-specific signal?

Specific issues

- Figure 2A – most of antibody signal is located outside of nuclei. It is worth showing a positive control of some sort to prove specificity of the signal. Oxidative stress+ NAC treatment for example.
- Figure 2C – “chromatin immunoprecipitation (ChIP) for pS2-PolII, followed by dot blot analysis using an 8-oxoG-specific antibody, revealed that in *Tcea1*^{-/-} MEFs, the majority of elongating RNAPII was associated with 8-oxoG DNA lesions”– This sentence is misleading. The ChIP experiment shows that pS2-PolII associates with more 8-oxoG DNA lesions in *Tcea1*^{-/-} MEFs than in WT cells. To be able to state something about the majority of the protein one should at least show the pull-down efficiency

being near 100%.

- Figure 2D – “This decrease indicates impaired transcription elongation likely due to the absence of TFIIS-dependent transcript cleavage, which exacerbates transcription stress”. - An inhibitor control is required to support this statement

- Figure 3B – If anaphase bridges are coming from telomere fusions as stated, one should be able to detect telomeric sequences on the anaphase bridges with FISH.

- Figure 3F – “Fluorescence in situ hybridization (FISH) experiments, with a PNA telomere (TelC) probe, indicated that approximately 50% of the assessed metaphases displayed at least one telomere fusion event (Figure 3F)”. – This sentence is misleading as on the graph the Y axis is labelled “%of fusions/ metaphase”.

- Figure 3G is strongly over-exposed and therefore should not be used for Q-FISH analysis

- Figure 4A – Normally pattern of gH2AX and 53BP1 staining looks different from one another even though many foci overlap. On the representative image gH2AX and 53BP1 look exactly the same (with adjusted brightness) – check if not a duplicated image. Also, for TIF assay cut-off in the field is % of cells with more than 5 TIFs (sometimes 3), but not 1.

- All the ChIP experiments: authors can't draw conclusions about reduced presence of sheltering on telomeric DNA unless they correct for differences in telomere length between samples (which they show earlier). They need to represent the results as fraction of input.

- Moreover, a phenotype for shelterin-free telomeres should lead to fusions via NHEJ – (Celli and de Lange 2005) which is not observed here, this needs to be discussed.

- Figure 5E – There is a band detected by pS2-PolII antibody in the IgG pulldown. Either this experiment can't be used to draw any conclusions or this band needs to be explained.

- Figure 6A – the positive control behaves exactly the same as telomeric region. Doesn't this mean that the phenotype observed is not telomere-specific? Was the presence of Rpl13a sequences in the cytoplasm tested?

- Figure 6B - intensity of TelC signal is suddenly the same in WT cells and Tcea1-/-, unlike in previous figures

- “Notably, TRF1 is released from telomeres upon oxidative DNA damage and requires active transcription to be recruited on telomeric DNA”. – To make this statement TRF1 protein levels need to be assessed on the WB and mRNA levels in the time course – what is the protein half-life? Can the recovery depend on synthesizing new TRF1, therefore recovery of the transcription?

- Figure 7B – mark cytoplasm to state that what you see are EV

- Sup figure 2A – to be able to judge the change of morphology cells need to be seeded at the same density

- Sup figure 4H – mislabeled. Legend states that this TRF demonstrates telomere length in yeast cells What are the lanes?

- “Firstly, in Tcea1-/- cells, the gradual buildup of 8-oxoGs triggers the release of TRF1 from telomeric DNA, resulting in uncapping of telomeres, activation of DDR, and chromosomal fusions”. – according to Sfeir et al., 2009 in absence of TRF1 there are no more than 2% of telomeric fusions, and if you state

that TRF2 is also decreased – it should be up to 20%

- It is not clear to me how R-loops are generated at telomeres in the absence of TFIIS given that most of TERRA molecules in mice are not transcribed from subtelomeres (Viceconte et al., 2020), this needs to be addressed

Reviewer #3 (Remarks to the Author):

In the manuscript by Siametis et al. entitled "Transcription stress at telomeres leads to cytoplasmic DNA release and paracrine senescence" the authors describe that transcription stress caused by TCEA1 deficiency results in accumulation of stalled Pol II and the occurrence of R-loops at telomeres, resulting in telomere uncapping. Interestingly, this results in telomeric DNA fragments in the cytosol that are also exported to other cells in the form of extracellular vesicles contributing to the onset of senescence. This is an exciting observation which provides a mechanistic explanation for the link between transcription stress and aging and is therefore highly interesting and a strong candidate for the broad readership of Nature Communications, if the following comments are addressed:

Major comments:

- Most studies are performed with one single knock-out clone. To rule out clonal effects, a selection of experiments should be executed using either different KO clones or by rescuing the KO cell line with a TCEA1 expression construct.
- The authors suggest that RNAPII is stalled at endogenous oxidative lesions in TCEA1 KO cells. Can the phenotypes be rescued by adding scavengers?
- Can the observed effects of TCEA1 KO on the cell cycle be caused by other -still unknown- functions of TCEA1? The transcription dependency of the observed cell cycle defects should be shown.

Minor comments:

- Figure 1D, and similar figures, this is a strange way of plotting data that also does not provide info on the spread of the individual data points of the wt samples.
- Figure 1E, To properly exclude an arrest in e.g. late S-phase the experiment should be executed with a BrdU or EDU staining to identify s-phase cells.
- Figure 1F, also the CFSE analysis at T=0 and e.g. T=24 hr should be shown
- Figure 2C: the statement "the majority of RNAPII was associated with 8-oxo-G" is an overstatement, it can only be concluded that more RNAPII is associated with oxidative damage
- TCEA1 KO cells are bigger, also the nuclei, how is this taken into account in the analysis of e.g. MFI plotted?
- Figure 4E, how does this OxiDIP signal look at other -non telomeric- regions. Is this effect specific for telomeres or is it observed throughout the genome.
- Figure 5D should be compared in one single IP experiment with supplemental fig 6A as more RNAPII is expected upon H₂O₂ treatment. This should be compared directly in one single experiment. Furthermore this experiment should be executed in TCEA1 KO cells.
- Suppl. Fig. 5F is not convincing, no clear differences in TRF1 localization are observed. If less TRF1 is bound to the chromatin in TCEA1 KO cells, where does this go, no increase in other compartments is observed. This should be repeated with IF experiments or by other more convincing experiments?
- Suppl. Fig. 5G, what is the proof the proteins reacting with UB antibody are TRF1, it might also be TRF1 interacting proteins that are ubiquitinated. If the authors want to make this claim, they should isolate ubiquitinated proteins and stain for TRF1 to detect higher migrating TRF1 species.
- Figure 5G, the essential DRB only control sample is missing in this experiment
- Suppl. Fig 6b-D, I guess the authors mean that TPL and actinomycin D are "NON"-reversible inhibitors?

Authors' reply to Reviewers' remarks

We thank the Reviewers for their support and constructive comments on our work. Below, we present a point-by-point response addressing each of the Reviewers' remarks. In the revised manuscript, all the relevant changes have been highlighted in yellow to ensure easy identification.

REVIEWER COMMENTS

Reviewer #1:

In this study the authors examine MEFs lacking the TFIIS factor (Tcea1^{-/-}), which is critical for the resumption of transcription after RNA Polymerase II stalling during elongation. The authors find the Tcea1 ko MEFs show multiple hallmarks of premature senescence and oxidative stress. The knock cells also show higher amounts of staining for 8-oxoguanine, R-loops, and DDR positive telomeres, and reduced BrU staining indicative of reduced transcription, by immunofluorescence. The authors also provide data showing the colocalization of 8-oxoguanine foci with telomeres, and a decrease in TRF1 at telomeres in the knock-out cells. The Tcea1^{-/-} cells and cells from older animals also exhibit cytoplasmic DNA fragments and extracellular vesicles that contain telomeric DNA, and the EVs from Tcea1 knockout cells can induce senescence in wild type cells. Based on the data the authors propose a model that RNA Polymerase II stalling at 8-oxoguanine in telomeres can cause telomere dysfunction-induced senescence, and the release of telomeric DNA in the cytoplasm, stimulating the senescence associated secretory phenotype that triggers paracrine senescence. The manuscript describes a great deal of phenotypic data from Tcea1^{-/-} MEFs related to transcription and genomic and telomere instability, but a demonstration of causal relationships is less strong.

Authors' reply: We thank the reviewer for her/his supportive remarks and constructive comments.

While the focus on telomeres is interesting, it is less clear how R-loops elsewhere or RNA PolII stalling at sites outside the telomeres contribute to senescence and the inflammatory response in Tcea1^{-/-} cells. In Figure Tcea1^{-/-} cells show a strong reduction in BrU staining, and an increase in 8-oxoguanine staining, throughout the entire nucleus. Specific comments are below.

Authors' reply: We now show that R-loops or RNAPII stalling at sites outside telomeres have a smaller contribution to the Tcea1^{-/-} senescent phenotype, which is mainly telomere-specific. Specifically, we provide the following experimental evidence:

- TERT overexpression (**Supplementary Figure 11C**) increased telomere length (**Supplementary Figure 11D**) and markedly decreased *p16* mRNA levels and SA- β -gal staining in TFIIS-deficient cells (**Supplementary Figure 11E-F**).
- The decrease of 8-oxoG lesions in cells treated with the antioxidants MitoTEMPO or NAC (**Figure 2A** and **Supplementary Figure 3A**), leads to a decrease in pS2-PolII levels (**Supplementary Figure 3E**), the consequent reduction of R-loop accumulation (**Supplementary Figure 5C**) and the mitigation of the transcription impairment observed through BrU incorporation (**Supplementary Figure 5A**) in Tcea1^{-/-} MEFs.

- Breaks Labeling, Enrichment on Streptavidin, and Sequencing (BLESS) for the quantification of telomere-specific DSB levels in TFIIS-defective cells treated with RNase H, revealed that DNA breaks on telomeres are R-loop dependent (**Supplementary Figure 9B-C**). Importantly, the accumulation of DSBs on telomeres is alleviated in NAC-treated cells (**Supplementary Figure 9C**). This reduction in genome instability ultimately leads to an increase in average telomere length of *Tcea1*^{-/-} cells (**Figure 3H**).
- Sequencing of purified cytoplasmic DNA fractions revealed an enrichment of TTAGGG repeats in cytosolic DNA fragments from *Tcea1*^{-/-} compared to wt cells (**Supplementary Figure 10** and **11A-B**).
- We employed a vesicle delivery system to treat wt and *Tcea1*^{-/-} MEFs as well as primary hepatocytes and pancreatic cells derived from 2- and 24-month-old mice with the ssDNA-specific S1 nuclease. Treatment of cells with S1 nuclease led to the significant reduction of cytoplasmic TelC fragments (**Supplementary Figure 12A**) and type I-related immune gene expression (**Figure 6C, 6F, Supplementary Figure 12B** and **12H**).
- Overexpression of TFIIS rescued the transcription impairment and senescent phenotype of *Tcea1*^{-/-} MEFs (**Supplementary Figure 2H-J, 5B** and **11D**).

1. Fig 2A. Is the 8-oxoguanine staining outside the nucleus mitochondrial? There appears to be a great deal more 8-oxoguanine staining outside the nucleus in *Tcea1* ko cells in Supplementary Figure 3C, compared to the images main Fig 2A. If so, then it appears that the abundance of 8-oxoguanine may be higher in the mitochondria than in the nucleus. More detail on how MFI was obtained would be helpful. How were thresholds set?

Authors' reply: Indeed, it cannot be ruled out that mitochondria contribute to a fraction of the cytoplasmic 8-oxoG signal. We now provide representative images of 8-oxoG staining together with MitoTracker, that show that some cytoplasmic oxoG foci are colocalized with mitochondria, comparably in wt and *Tcea1*^{-/-} MEFs (**Supplementary Figure 4C**).

The observed differences in the cytoplasmic oxoG signals between experiments are explained due to the different protocols used. Specifically, the 8-oxoguanine immunofluorescence staining was performed as described in Kumar et al., 2022, with the use of HCl and NaOH (detailed in the **Methods section**). This protocol, although it produces oxoG nuclear signals without any background noise, it does not allow co-staining with other protein-targeting antibodies. For pS2-RNAPII/oxoG colocalization experiments, the immunofluorescence protocol was used (**Methods section**).

MFI for 8-oxoG was measured only in the nucleus (DAPI-stained areas). The images were acquired on a Leica SP8 confocal laser scanning microscope equipped with a 63X oil objective and captured at 1024 × 1024 pixels. Colocalization events were analyzed with Fiji's pixel intensity-based Coloc2 plugin and verified manually after thresholding in grayscale.

2. Supplementary Fig 2C shows that a mitochondrial ROS scavenger decreases the 8-oxoguanine staining in the nucleus. Can this scavenger (MitoTEMPO) also suppress the other phenotypes observed in the *Tcea1*^{-/-} cells, such as increased R-loops, decreased BrU staining, and increased pS2-PolIII? In other words, if the oxidative stress arising from dysfunctional mitochondrial causes the RNA PolIII stalling at 8-oxoguanine lesions in telomeres, this should be ameliorated with MitoTEMPO.

Authors' reply: Indeed, MitoTEMPO suppresses other phenotypes observed in *Tceal*^{-/-} cells. Specifically, MitoTEMPO ameliorates the impaired transcription elongation observed in TFIIIS-deficient cells, as evidenced by an increase in BrU incorporation into nascent transcripts (**Supplementary Figure 5A**). In line, the scavenger-induced oxoG reduction (**Supplementary Figure 3D**), significantly decreases the stalled elongating (pS2) RNA polymerase (**Supplementary Figure 3E**) and further results in a reduction of s9.6-stained R-loops in *Tceal*^{-/-} cells (**Supplementary Figure 5C**).

Additionally, we treated wt and *Tceal*^{-/-} MEFs with the antioxidant N-acetyl cysteine (NAC), which reduced the oxoG levels in NAC-treated *Tceal*^{-/-} cells (**Figure 2A** and **Supplementary Figure 3A**) and further resulted in a significant decrease in DSB accumulation on telomeres, as quantified by Breaks Labeling, Enrichment on Streptavidin, and Sequencing (BLESS) (**Supplementary Figure 9D**). Importantly, qFISH experiments showed that NAC treatment led to an increase of *Tceal*^{-/-} average telomere length (**Figure 3H**).

3. In Fig 2C how was the ChIP for pS2-PolIII controlled for the higher levels of this phosphorylated form of RNAPII in *Tcea1*^{-/-} cells? Could this explain the increase in 8-oxoG staining in the associated DNA?

Authors' reply: In **Figure 2C**, the 8-oxog signal after pS2-PolIII ChIP was normalized over input. In the revised manuscript, we also provide a new series of pS2-PolIII IP experiments, which show that the pulldown efficiency is at about 80%, for both wt and *Tceal*^{-/-} cells (**Supplementary Figure 4A**), indicating that we immunoprecipitate the same amount of pS2-PolIII in wt and *Tceal*^{-/-} cells, independently of the total pS2 protein levels.

4. Previous reports indicate that imaging the S9.6 antibody is subject artifacts (Smolka et al., JBC, 2021). However, the JBC publication provided methods to control for these artifacts. The authors should indicate how these controls were used to rule out artifactual staining in Fig. 2F.

Authors' reply: All s9.6 experiments were performed under the exact same conditions using RNase T1- and RNase III-treated cells, with or without RNase H, as now detailed in the **Methods section**.

5. The gammaH2AX and 53BP1 foci are not obvious in the images in Fig 2G. An enlargement or "zoom in" of a representative nucleus would help show these foci better. Addition of RNase H appears to decrease gammaH2AX staining, but the effect on 53BP1 is less obvious. GammaH2AX also marks sites of stalled replication forks. Also, the authors imply that the DSBs arise from R-loops, which expose long stretches of ssDNA that are susceptible to breaks. Therefore, a marker of ssDNA is warranted here. For example, pulse with BrdU, followed by anti-BrdU staining under non-denaturing conditions.

Authors' reply: To meet the Reviewer's request, we replaced the images with representative "zoom in" versions showing γ H2AX and 53BP1 foci (**Figure 2F**). Treatment with RNase H reduces the γ H2AX signal and decreases the damage-induced 53BP1 foci; 53BP1 staining in untreated wt cells is expected to have a pan-nuclear pattern.

Pulse with BrdU, followed by anti-BrdU staining under non-denaturing conditions in *Tceal*^{-/-} MEFs showed increased ssDNA signal, compared to wt controls (**Figure 2G**).

We also performed Breaks Labeling, Enrichment on Streptavidin, and Sequencing (BLESS) to directly label and isolate DSBs in *Tceal*^{-/-} and wt cells. In line with the decrease in R-loop levels seen in *Tceal*^{-/-} cells cultured from passage 2 (P2) to P5 (**Figure 6A**), we observed an

increase in DSB levels in P5 compared to P2 *Tcea1*^{-/-} cells. The *Rpl13a* gene, used as an R-loop-prone positive control, showed similar kinetics in DSB quantification, albeit in a much smaller range, while a non-transcribed intergenic region, used as a negative control, presented no DSB accumulation (**Supplementary Figure 9B**). Importantly, BLESS experiments, upon transfection with recombinant RNase H, showed that the resolution of R-loops results in fewer DSBs on telomeres, providing additional evidence for the telomeric R-loop-induced genomic instability (**Supplementary Figure 9C**).

6. *The conclusion that TFIIIS loss ultimately leads to DSBs would be strengthened greatly by including an assay that tests directly for DSBs, such as the Comet assay, rather than only 53BP1 foci staining. Are these breaks only at telomeres?*

Authors' reply: We performed Breaks Labeling, Enrichment on Streptavidin, and Sequencing (BLESS) in order to directly label, isolate and quantify DSBs in *Tcea1*^{-/-} and wt cells. We observed a significant increase in DSB levels in *Tcea1*^{-/-} compared to wt telomeres. The *Rpl13a* gene, used as an R-loop-prone positive control, showed a similar DSB accumulation from passage 2 (P2) to passage 5 (P5), albeit in a much smaller range, while a non-transcribed intergenic region, used as a negative control, presented no DSB accumulation (**Supplementary Figure 9B**).

7. *It is unclear how 8-oxoguanine might inhibit telomere transcription in MEFs since the C-rich strand is transcribed.*

Authors' reply: There are instances where 8-oxoGs obstruct ongoing transcription. For instance, during BER, after 8-oxo-Gs are recognized and removed by OGG1, they are processed into nicks by APE1. The presence of a nick in the non-template strand of a transcribed region hampers DNA reannealing, and may serve as an initiation site for R-loops (Roy et al., 2009; Roy et al., 2010). The propensity of telomeric sequences to form G4 structures may, additionally, decrease 8-oxoG repair by inhibiting OGG1 and APE1 (Broxson et al., 2014). In combination with the accumulation of R-loops, this would further stall the elongating RNA polymerase complexes.

8. *The loss of TRF1 (as shown later) is expected to cause telomere fragility based on evidence from the literature (first reported in Sfeir et al, 2009, Cell). Did the authors see any evidence of fragile telomeres on the metaphase spreads from Tcea1 ko MEFs? (Fig 3F).*

Authors' reply: Indeed, we provide evidence that fragile telomeres are present in *Tcea1*^{-/-} MEFs. Specifically, we now include new representative metaphases from *Tcea1*^{-/-} and wt cells, cultured in the absence or presence of the NHEJ DNA ligase IV inhibitor SCR130 (**Figure 3I**). The abrogation of telomeric end-repair, allows for more observable instances of short and broken telomeres (fragile or missing).

9. *In Fig 3F there is a discrepancy between the legend and the description of the results in the text. It is not clear if the percentage of telomere fusion events per metaphase, or the percent of metaphases with at least one telomere fusion event is shown. In this latter case, a more accurate label for the y-axis might be "% of metaphases with ≥1 fusions".*

Authors' reply: This has been corrected. The axis now reads "% of metaphases with ≥1 fusion".

10. The conclusion that “defect in TFIIIS results in shorter telomeres that become fused, leading to anaphase bridges and mitotic aberrations”, does not appear to be well supported by the data. The small reduction in telomere length in *Tcea1* ko MEFs by telomere FISH and S blot is not expected to lead to telomere fusions. The telomeres are still quite long in the *Tcea1* ko MEFs (between 38 and 48kb).

Authors’ reply: Despite the small reduction in average telomere length, *Tcea1*^{-/-} MEFs present critically short telomeres, enough to induce senescence. We have now used the NHEJ DNA ligase IV inhibitor SCR130, to impair NHEJ-mediated DSB repair at unprotected *Tcea1*^{-/-} telomeres. SCR130-mediated NHEJ inhibition led to *Tcea1*^{-/-} cells with no telomere fusions, which allowed us to quantify the frequency of metaphases with at least one critically short telomere (broken/missing). **Figure 3I** shows representative images of fused, fragile, shorter or missing telomeres in *Tcea1*^{-/-} chromosomes.

11. Is the reduction of TRF1, TRF2 and TIN2 localization to telomere by CHIP and/or by IF due to a reduction in protein levels? Is transcription of TRF1, TRF2 and TIN2 impacted by the loss of *Teac1*?

Authors’ reply: The reduction of TRF1, TRF2 and TIN2 localization on telomeres is not due to a reduction in their protein levels. Quantitative PCR and western blotting revealed no difference in the *Trf1*, *Trf2* and *Tinf2* mRNA levels and TRF1, TRF2 and TIN2 protein levels, respectively between *Tcea1*^{-/-} cells and corresponding wt controls. (**Figure 4C-D** and **Supplementary Figure 7C-F**).

12. Was the TFIIIS complex with TRF1 and RNAPII shown by colIP in Fig 5D (and 5E for TRF1 and pS2-RNAPII) mediated by interaction on DNA?

Authors’ reply: These protein interactions do not depend on DNA, as the immunoprecipitations were performed in the presence of benzonase and RNase A.

13. The results and conclusion in Fig 5F are confusing. The decrease in BrU signal (overall and at telomeres) in *Tcea1* ko cells suggests transcription is impaired in the ko cells, and at telomeres. But why then do the authors not detect a decrease in TERRA?

Authors’ reply: BrU incorporation indicates ongoing transcription of nascent RNA, at the time of the experiment (now included in the manuscript, for **Figure 5F**). On the other hand, the steady state levels of RNA transcripts, seen by qPCR, e.g. for TERRA or *Trf1* levels (**Figure 4D**), are not always affected in *Tcea1*^{-/-} cells, also confirmed by the RNAseq analyses (**Figure 1G**).

We have now included additional quantifications of TERRA levels, using TERRA-specific or random hexamers for reverse transcription, and quantitative PCR primers for total TERRA cDNA, or specific for Chr.2, Chr.18 and the PAR locus (**Supplementary Figure 6D-H**). We find a significant reduction only in the PAR-TERRA levels, consistent with a recent study assigning the main contribution of TERRA levels to this locus (Viceconte et al., 2021). In support, TRF1 depletion has been shown to increase TERRA levels (Arora et al., 2014), which in a transcriptionally impaired set up, such as the *Tcea1*^{-/-} cells, could be misrepresented, while genotoxic stress and short telomeres have been reported to stabilize the TERRA RNA levels (Liu et al, 2023; Chen et al., 2022).

14. For Fig 5G, the authors allowed the cells to recover for 16 h after H₂O₂ treatment, which restores TRF1 localization. However, this restoration is impaired if the authors add a drug that inhibits transcription elongation. They interpret this result as evidence that TRF1 recruitment to the telomeres requires active transcription. However, this seems at odds with evidence from the literature that TRF1 is recruited to telomeres by its Myb telomeric DNA binding domain. Furthermore, not all telomeres are typically transcribed in most cell types. Does DRB addition during the 16 hour recovery affect overall transcription and protein levels of TRF1?

Authors' reply: The results shown in **Figure 5G** and **Supplementary Figure 8B-G** support the involvement of the transcription process for the re-loading of TRF1, which is not incompatible with the requirement of two interacting TRF1 Myb domains (TRF1 homodimers) to bind telomeric DNA.

We now include a series of Western blots and qPCR quantifications for TRF1 protein and mRNA levels (**Supplementary Figure 8E-F**) which show that although DRB leads to a decrease in TRF1 mRNA, total protein levels remain unaffected in all conditions of the experiment. We also replaced the sentence, which now reads “involves active transcription”.

Overall, transcription verification at telomeres, besides the reported TERRA sources (Chr18, Chr2, PAR-X/Y) is challenging, since, in contrast to the human Telomere-to-Telomere reference genome, there is still no complete genome assembly to include the massive murine telomere length. Any transcription activity in the telomeres with still unknown sequence or the repetitive part of chromosome ends, would be difficult to sequence/align/map and thus remain undetected. Nevertheless, apart from gene expression, we cannot exclude the transcriptional activity of RNA polymerases during DNA repair (d'Adda di Fagagna, 2014).

15. In Fig. 6B could the TelC probe be recognizing TERRA in the cytoplasm? The C-rich probe should bind to the G-rich TERRA transcripts.

Authors' reply: Technically, the TelC probe can recognize RNA telomeric repeats. However, we find that telomeric DNA fragments are enriched in *Tceal*^{-/-} cytoplasm, compared to TERRA transcripts. We now provide measurements of the cytoplasmic telomeric fragments, from wt and *Tceal*^{-/-} cells, treated with S1 nuclease or RNase A, that show that the *Tceal*^{-/-} cytoplasmic TelC FISH signal is reduced only by S1 treatment (**Supplementary Figure 9E**). We also performed sequencing on the cytoplasmic DNA fraction of wt and *Tceal*^{-/-} cells, which shows the enrichment of *Tceal*^{-/-} cytoplasm in TTAGGG repeats, compared to wt controls (**Supplementary Figure 10** and **11A-B**).

16. In Fig 6C could the micronuclei (Fig 3A) be driving the inflammatory response? In Fig. 6D is TRF1 expression lower in aged mice (24 months). As telomeres shorten with age the binding sites for TRF1 are also reduced. Fig 6E, it is curious that the cytoplasmic telomeric staining does not overlap with DAPI. Is this because the authors suspect these foci are R-loops that may not stain with DAPI? Do they overlap with S9.6 staining? How do the authors know that the cytoplasmic telomeric staining in cells from the aged mice is causing the inflammatory response?

Authors' reply: Indeed, micronuclei have been linked to inflammation induction, through sensing of leaked DNA into the cytoplasm. However, we rarely see TelC signal, if any, overlapping with DAPI stain in the cytoplasm.

TRF1 mRNA levels are not altered with age. We have now included the quantified TRF1 protein levels in primary hepatocytes and pancreatic cells from young (2-month-old) and old (24-month-old) mice, measured by Western blots that show no differences with age (**Supplementary Figure 12D-E**).

We cannot rule out cytoplasmic RNA-DNA hybrids as a source of TelC fragments, but we have new evidence that point to the telomeric fragments being single-stranded DNA. Specifically, we now include sequencing experiments of the *Tceal*^{-/-} cytoplasmic DNA (**Supplementary Figure 10** and **11A-B**) and a series of qPCR experiments of untreated or S1 nuclease-treated wt and *Tceal*^{-/-} MEFs (**Figure 6C** and **Supplementary Figure 12B**), as well as of young and old hepatocytes or pancreatic cells (**Figure 6F** and **Supplementary Figure 12G-H**). We find that the mRNA levels of the inflammatory genes that are upregulated in *Tceal*^{-/-} or 24-month-old mice are reduced upon ssDNA-specific nuclease (S1) treatment. In line, we provide representative images of EVs from wt and *Tceal*^{-/-} MEFs, untreated or treated with RNase H, stained against the s9.6 antibody, together with the TelC FISH signal, which show that the TelC-positive *Tceal*^{-/-} EVs do not contain RNA-DNA hybrids (**Supplementary Figure 13B**).

17. In Fig 7 it is interesting that the EVs from Tcea1 ko MEFs show greater telomeric DNA staining. But it is unclear if the telomeric fragments, or R-loop containing telomeric DNA, are responsible for the inflammatory response and senescence induction in recipient cells.

Authors' reply: The induction of paracrine senescence was found to be DNA-dependent with the cytoplasm of *Tceal*^{-/-} MEFs enriched in telomeric DNA fragments compared to wt MEFs. We have now provided a new series of immunofluorescence assays and representative images of EVs from wt and *Tceal*^{-/-} MEFs. These EVs were either untreated or treated with RNase H and stained against the s9.6 antibody, coupled with TelC FISH. The results clearly demonstrate that EVs from *Tceal*^{-/-} cells do not contain RNA-DNA hybrids (**Supplementary Figure 13B**). At the same time, the cytoplasmic DNA sequencing experiments show that *Tceal*^{-/-} cells are enriched in TTAGGG repetitive DNA fragments (**Supplementary Figure 10** and **11A-B**). When these TelC-positive EVs are treated with DNase I and introduced to recipient wt MEFs, the mRNA levels of type I-related immune genes are reduced, compared to *Tceal*^{-/-} TelC-positive EVs that are left untreated (**Figure 6C** and **Supplementary Figure 12B**). We also employed a vesicle delivery system to treat wt and *Tceal*^{-/-} MEFs as well as primary hepatocytes or pancreatic cells derived from 2-month and 24-month old animals with the ssDNA-specific S1 nuclease, which resulted in a significant reduction of cytoplasmic TelC fragments (**Supplementary Figure 12A**) and type I-related immune gene expression (**Figure 6C, 6F** and **Supplementary Figure 12B** and **12H**).

Reviewer #2:

In their paper Siametis et al., demonstrate using Tcea1^{-/-} mouse embryonic fibroblasts that cells with defect in transcription elongation exhibit senescence phenotype due to accumulation of 8-oxoG lesions throughout the genome, stalling of RNAPII at oxidative DNA damage sites, impaired transcription, accumulation of R-loops throughout the genome, DNA damage response and genomic instability. They also claim that TFIIS deficiency leads to telomere shortening and fusions, due to telomere uncapping and DNA damage activation at telomeres; and that R-loops at telomeres contribute to release of telomeric DNA fragments in the cytoplasm of Tcea1^{-/-} cells and primary cells derived from naturally aged animals triggering a viral-like immune response.

Although the findings presented are of interest, I have numerous methodological issues with the data presented in the paper, which require careful evaluation.

Authors' reply: We thank the reviewer for her/his supportive remarks and constructive comments.

Also I have doubts in telomere-specific origin of the genomic instability and senescence in case of TFIIS deficiency, given that only a minor fraction of telomeres are fused, authors see increase of R-loops in the control loci as well as at telomeres, and according to their model immune response is triggered by ssDNA released from R-loops.

Authors' reply: We provide additional evidence that the genome instability and senescence seen in *Tcea1^{-/-}* MEFs is mainly telomere-specific.

Specifically, we have included the following strategies:

- We overexpressed TERT (**Supplementary Figure 11C**) to increase telomere length in TFIIS-deficient cells (**Supplementary Figure 11D**) and show that the senescent characteristics associated with the TFIIS defect are attenuated (**Supplementary Figure 11E-F**).
- Using the DNA ligase IV-specific inhibitor SCR130 to inhibit NHEJ, we find that the chromosome ends that would be ligated/fused, present critically short or missing telomeres, enough to induce senescence (**Figure 3I**).
- We find that treating cells with the antioxidants MitoTEMPO or NAC to reduce the levels of 8-oxoG lesions (**Figure 2A** and **Supplementary Figure 3A**), leads to a decrease in pS2-PolIII levels (**Supplementary Figure 3E**), the anticipated reduction of R-loop accumulation (**Supplementary Figure 5C**) and the attenuation of the transcription impairment as evidenced by BrU incorporation (**Supplementary Figure 5A**) in *Tcea1^{-/-}* MEFs.
- Using Breaks Labeling, Enrichment on Streptavidin, and Sequencing (BLESS) to quantify the telomere-specific DSB levels, we find that DNA breaks on telomeres are R-loop-dependent (**Supplementary Figure 9B-C**). This DSB accumulation is alleviated in NAC-treated cells (**Supplementary Figure 9C**) reducing genome instability and ultimately resulting in an increase in average telomere length of *Tcea1^{-/-}* cells (**Figure 3H**).

- We find an enrichment of TTAGGG repeats in cytosolic DNA fragments from *Tcea1*^{-/-} compared to wt cells, by sequencing purified cytoplasmic DNA fractions (**Supplementary Figure 10** and **11A-B**).
- We employed a vesicle delivery system to treat wt and *Tcea1*^{-/-} MEFs as well as primary hepatocytes and pancreatic cells derived from 2- and 24-month-old mice with the ssDNA-specific S1 nuclease. Treatment of cells with S1 nuclease led to the significant reduction of cytoplasmic TelC fragments (**Supplementary Figure 12A**) and type I-related immune gene expression (**Figure 6C, 6F, Supplementary Figure 12B** and **12H**).
- We find that overexpression of TFIIS in *Tcea1*^{-/-} MEFs rescues the transcription impairment and senescent phenotype of these cells (**Supplementary Figure 2H-J, 5B** and **11D**).

Methodological issues:

Authors base their story on a variety of microscopy and immunoprecipitation-based techniques to generate the data, and these methods and data analyses required have long been established in the field. Unfortunately, data presented in the paper often don't meet the standards.

*Golden standard for analyzing ChIP and DRIP data is showing the percentage of input DNA recovered with IP after subtraction of the background signal from IgG pulldowns. Normalization to background noise (IgG) is misleading first of all because it gives higher numbers, and second, it does not account for differences in telomeric DNA content in different samples (meaning telomere length, which authors claim to be different between WT cells and *Tcea1*^{-/-} cells)*

Authors' reply: This has now been corrected. All ChIP, DRIP and Oxi-DIP analyses show the percentage of input DNA immunoprecipitated with each antibody, after the subtraction of the background IgG-derived signal.

Protein co-IP experiments need to be quantified or at least labeled with fraction of input/fraction of IP loaded.

Authors' reply: This has now been corrected. IP fractions have been quantified and normalized over input levels.

Lots of imaging experiments show pan-nuclear staining, or background in the cytoplasm. Given that authors rely on co-localization of foci to draw their conclusions, there is a need in a detailed description of how data were analyzed and ideally z-stack analysis of confocal pictures. Lens used should be specified in the materials and methods section, and the standard for for TIFs and other co-localization experiments in the field is x63 or x100. All images lack the scale of the scale bar and information about magnification.

Authors' reply: This information is now detailed in the **Methods** section and/or specific legends. All images were acquired on a Leica SP8 confocal laser scanning microscope equipped with a 63X oil objective and captured at 2,084 × 2,084 pixels and the scale bar was set at 5µm, unless otherwise indicated. For quantification analyses, the same number of 0.35µm sections were used for z-stacks of different conditions within each experiment.

All the 8-oxo-G IF images have extremely high background which makes it very difficult to accept the conclusions from co-localization studies. Cells are already pre-treated with RNaseA, where does this background come from? Could pre-extraction help getting rid of the non-specific signal?

Authors' reply: The residual cytoplasmic background noise is explained due to the different protocols used. The 8-oxoguanine immunofluorescence staining was performed as described in Kumar et al., 2022, with the use of HCl and NaOH (detailed in the **Methods section**). This protocol, although it produces oxoG nuclear signals without any background noise, it does not allow co-staining with other protein-targeting antibodies, thus, experiments such as the pS2-PolIII/oxoG or the MitoTracker/oxoG IF were performed without prior denaturation (**Methods section**). Pre-extraction results in detergent-mediated removal of all soluble proteins, making it unsuitable for some measurements.

Specific issues

- Figure 2A – most of antibody signal is located outside of nuclei. It is worth showing a positive control of some sort to prove specificity of the signal. Oxidative stress+ NAC treatment for example.

Authors' reply: We have now added a positive (H₂O₂) and a negative (NAC) control for the 8-oxoG experiment (**Figure 2A** and **Supplementary Figure 3A**).

- Figure 2C – “chromatin immunoprecipitation (ChIP) for pS2-PolIII, followed by dot blot analysis using an 8-oxoG-specific antibody, revealed that in Tcea1-/- MEFs, the majority of elongating RNAPII was associated with 8-oxoG DNA lesions”– This sentence is misleading. The ChIP experiment shows that pS2-PolIII associates with more 8-oxoG DNA lesions in Tcea1-/- MEFs than in WT cells. To be able to state something about the majority of the protein one should at least show the pull-down efficiency being near 100%.

Authors' reply: The text now reads “more elongating RNAPII was associated with 8-oxoG DNA lesions in TFIIS-deficient cells compared to wt controls”. We now also provide a new series of pS2-PolIII IP experiments which show that the pulldown efficiency is at about 80%, for both wt and *Tcea1*^{-/-} cells (**Supplementary Figure 4A**).

- Figure 2D – “This decrease indicates impaired transcription elongation likely due to the absence of TFIIS-dependent transcript cleavage, which exacerbates transcription stress”. - An inhibitor control is required to support this statement

Authors' reply: **Figure 2D** and **Supplementary Figure 4D** now include a control treatment with the transcription elongation inhibitor DRB.

- Figure 3B – If anaphase bridges are coming from telomere fusions as stated, one should be able to detect telomeric sequences on the anaphase bridges with FISH.

Authors' reply: **Figure 3G** now shows the TelC-FISH signal spanning a chromatin bridge between two *Tcea1*^{-/-} MEFs.

- Figure 3F – “Fluorescence in situ hybridization (FISH) experiments, with a PNA telomere (TelC) probe, indicated that approximately 50% of the assessed metaphases displayed at least one telomere fusion event (Figure 3F)”. – This sentence is misleading as on the graph the Y axis is labelled “%of fusions/ metaphase”.

Authors’ reply: This has now been corrected.

- Figure 3 G is strongly over-exposed and therefore should not be used for Q-FISH analysis

Authors’ reply: This has now been corrected (now **Figure 3H**).

- Figure 4A – Normally pattern of gH2AX and 53BP1 staining looks different from one another even though many foci overlap. On the representative image gH2AX and 53BP1 look exactly the same (with adjusted brightness) – check if not a duplicated image. Also, for TIF assay cut-off in the field is % of cells with more than 5 TIFs (sometimes 3), but not 1.

Authors’ reply: This has now been corrected. The representative images of **Figure 4A** have been replaced and the graph depicts the % of cells with ≥ 5 TIFs.

- All the ChIP experiments: authors can’t draw conclusions about reduced presence of sheltering on telomeric DNA unless they correct for differences in telomere length between samples (which they show earlier). They need to represent the results as fraction of input.

Authors’ reply: This has been corrected. All ChIP analyses show the percentage of input DNA immunoprecipitated with each antibody, after the subtraction of the background IgG-derived signal.

- Moreover, a phenotype for shelterin-free telomeres should lead to fusions via NHEJ – (Celli and de Lange 2005) which is not observed here, this needs to be discussed.

Authors’ reply: This has been corrected. We now provide a series of metaphase spreads from cells treated with the DNA ligase IV-specific inhibitor SCR130 (**Figure 3I**). They show that, upon NHEJ inhibition, the chromosome ends that would be ligated/fused, present short or missing telomeres. Of note, according to the TRF1, TRF2, TIN2 ChIP and Immuno-FISH experiments, the deprotected telomere phenotype is not as severe as what would be expected in a complete TRF2 abrogation set up.

- Figure 5E – There is a band detected by pS2-PollII antibody in the IgG pulldown. Either this experiment can’t be used to draw any conclusions or this band needs to be explained.

Authors’ reply: This has been corrected. The experiments have now been repeated with the use of BS3 crosslinker and magnetic beads, which significantly decrease the background noise of the protocol (**Figure 5E**).

- Figure 6A – the positive control behaves exactly the same as telomeric region. Doesn’t this mean that the phenotype observed is not telomere-specific? Was the presence of Rpl13a sequences in the cytoplasm tested?

Authors' reply: The *Rpl13a* gene was selected as a positive control as it is prone to R-loop accumulation. Yet, **Supplementary Figure 9A** (*Rpl13a* R-loop levels) presents small differences compared to **Figure 6A** (telomere R-loop levels). The *Rpl13a* gene locus shows the tendency to form R-loops both in wt and *Tcea1*^{-/-} backgrounds, which remain present after 3 passages (P2 to P5). Instead, telomeres only show R-loop accumulation in *Tcea1*^{-/-} cells, which are then reduced in P5. In addition, in the new series of BLESS experiments, the increased accumulation of DSBs in P5 *Tcea1*^{-/-} telomeres (**Supplementary Figure 9B**), which is R-loop dependent (**Supplementary Figure 9C**), is significantly lower in *Rpl13a* and is not affected by RNase H transfection.

The presence of *Rpl13a* sequences in the cytoplasmic DNA fragments of *Tcea1*^{-/-} MEFs has been now quantified by the enrichment of sequenced reads/50bp bins over the *Rpl13a* locus (**Supplementary Figure 11A**) and have been found to be decreased compared to telomeric repeats (**Supplementary Figure 10A**).

- *Figure 6B - intensity of TelC signal is suddenly the same in WT cells and Tcea1-/-, unlike in previous figures*

Authors' reply: This has been corrected.

- *“Notably, TRF1 is released from telomeres upon oxidative DNA damage and requires active transcription to be recruited on telomeric DNA”. – To make this statement TRF1 protein levels need to be assessed on the WB and mRNA levels in the time course – what is the protein half-life? Can the recovery depend on synthesizing new TRF1, therefore recovery of the transcription?*

Authors' reply: This has been corrected. We now provide a series of Western blots and qPCR quantifications for TRF1 protein and mRNA levels (**Supplementary Figure 8E-F**) which show that although DRB leads to a decrease in TRF1 mRNA, total protein levels remain unaffected in all conditions of the experiment. At the same time, we have replaced the sentence which now reads “involves active transcription”.

- *Figure 7B – mark cytoplasm to state that what you see are EV*

Authors' reply: This has now been included.

- *Sup figure 2A – to be able to judge the change of morphology cells need to be seeded at the same density*

Authors' reply: This has been corrected.

- *Sup figure 4H – mislabeled. Legend states that this TRF demonstrates telomere length in yeast cells What are the lanes?*

Authors' reply: This has been corrected (now **Supplementary Figure 6L**). The different numbers refer to separate biological samples (now included in the legend).

- *“Firstly, in Tcea1-/- cells, the gradual buildup of 8-oxoGs triggers the release of TRF1 from*

telomeric DNA, resulting in uncapping of telomeres, activation of DDR, and chromosomal fusions". – according to Sfeir et al., 2009 in absence of TRF1 there are no more than 2% of telomeric fusions, and if you state that TRF2 is also decreased – it should be up to 20%

Authors' reply: This is correct. Indeed, we see 1-2 fusions per metaphase. The graphs in **Figures 3F** and **3I** measure the percentage of metaphases with at least one fusion.

- It is not clear to me how R-loops are generated at telomeres in the absence of TFIIIS given that most of TERRA molecules in mice are not transcribed from subtelomeres (Viceconte et al., 2020), this needs to be addressed

Authors' reply: Overall, transcription verification at telomeres, besides the reported TERRA sources (Chr18, Chr2, PAR-X/Y) is challenging, since, in contrast to the human Telomere-to-Telomere reference genome, there is still no complete genome assembly to include the massive murine telomere length. Any transcription activity in the telomeres of chromosomes with still unknown sequence or the repetitive part of chromosome ends, would be difficult to sequence/align/map and thus remain undetected. Nevertheless, apart from gene expression, we cannot exclude the transcriptional activity of RNA polymerases during DNA repair (d'Adda di Fagagna, 2014).

Reviewer #3:

In the manuscript by Siametis et al. entitled "Transcription stress at telomeres leads to cytoplasmic DNA release and paracrine senescence" the authors describe that transcription stress caused by TCEA1 deficiency results in accumulation of stalled Pol II and the occurrence of R-loops at telomeres, resulting in telomere uncapping. Interestingly, this results in telomeric DNA fragments in the cytosol that are also exported to other cells in the form of extracellular vesicles contributing to the onset of senescence. This is an exciting observation which provides a mechanistic explanation for the link between transcription stress and aging and is therefore highly interesting and a strong candidate for the broad readership of Nature Communications, if the following comments are addressed:

Authors' reply: We thank the reviewer for her/his enthusiasm, supportive remarks and constructive comments.

Major comments:

-Most studies are performed with one single knock-out clone. To rule out clonal effects, a selection of experiments should be executed using either different KO clones or by rescuing the KO cell line with a TCEA expression construct.

Authors' reply: We have not used one clone, as *Tcea1*^{-/-} MEFs are isolated from single embryos that are derived from the cross between *Tcea1*^{fl/fl} mice and CMV-Cre mice (detailed breeding scheme available in the Methods section). Single embryo-MEFs are then cultured for 4-5 passages and used for experiments.

We now additionally provide a series of experiments where we overexpress TFIIS (**Supplementary Figure 2H**). Specifically, **Supplementary Figure 2H-J** shows the effect of TFIIS overexpression on the *Tceal*^{-/-} senescent phenotype, by means of p16 mRNA levels quantification and measurements of the percentage of SA-β-gal⁺ cells.

-The authors suggest that RNAPII is stalled at endogenous oxidative lesions in TCEA1 KO cells. Can the phenotypes be rescued by adding scavengers?

Authors' reply: Indeed, we find that scavengers rescue the *Tceal*^{-/-} phenotype. Specifically, treatment of cells with the superoxide scavenger MitoTEMPO ameliorates the impaired transcription elongation observed in TFIIS-deficient cells, as evidenced by an increase in BrU incorporation into nascent transcripts (**Supplementary Figure 5A**). In line, the scavenger-induced oxoG reduction (**Supplementary Figure 3D**), significantly decreases the stalled elongating (pS2) RNAPII (**Supplementary Figure 3E**) and further results in a reduction of s9.6-stained R-loops in *Tceal*^{-/-} cells (**Supplementary Figure 3C**).

Additionally, treatment of wt and *Tceal*^{-/-} MEFs with the antioxidant N-acetyl cysteine (NAC), reduced the oxoG levels in NAC-treated *Tceal*^{-/-} cells (**Supplementary Figure 3A**) and further resulted in a significant decrease in DSB accumulation on telomeres, as quantified by Breaks Labeling, Enrichment on Streptavidin, and Sequencing (BLESS) (**Supplementary Figure 9D**). Importantly, qFISH experiments showed that NAC treatment leads to an increase of *Tceal*^{-/-} average telomere length (**Figure 3H**).

-Can the observed effects of TCEA1 KO on the cell cycle be caused by other -still unknown- functions of TCEA1? The transcription dependency of the observed cell cycle defects should be shown.

Authors' reply: While we cannot exclude the possibility of unknown functions of TFIIS affecting the cell cycle, our findings now demonstrate that G2/M accumulation may result from the activation of RNAPII by TFIIS. In the revised manuscript, we present a new cell cycle assessment of wild-type MEFs overexpressing a mutant form of TFIIS (TFIIS^{ΔIII}), which lacks the C-terminal domain III responsible for activating the RNAPII RNA cleavage function. BrdU-Propidium Iodide staining shows that the mutant TFIIS^{ΔIII} overexpression results in a higher percentage of cells in G2/M (**Supplementary Figure 2C**). This suggests that the RNA cleavage function of RNAPII is implicated in the cell cycle phenotype observed in *Tceal*^{-/-} MEFs.

Minor comments:

-Figure 1D, and similar figures, this is a strange way of plotting data that also does not provide info on the spread of the individual data points of the wt samples.

Authors' reply: This has been corrected.

-Figure 1E, To properly exclude an arrest in e.g. late S-phase the experiment should be executed with a BrdU or EDU staining to identify s-phase cells.

Authors' reply: This has been included.

-Figure 1F, also the CFSE analysis at T=0 and e.g. T=24 hr should be shown

Authors' reply: This has been included (**Supplementary Figure 3C**).

-Figure 2C: the statement "the majority of RNAPII was associated with 8-oxo-G" is an overstatement, it can only be concluded that more RNAPII is associated with oxidative damage

Authors' reply: This has been corrected. We now provide a new series of pS2-PolIII IP experiments which show that the pulldown efficiency is at about 80%, for both wt and *Tcea1*^{-/-} cells (**Supplementary Figure 4A**). The text now reads "more elongating RNAPII was associated with 8-oxoG DNA lesions in TFIIS-deficient cells compared to wt controls."

-TCEA1 KO cells are bigger, also the nuclei, how is this taken into account in the analysis of e.g. MFI plotted?

Authors' reply: All MFI measurements are normalized per cell volume.

-Figure 4E, how does this OxidIP signal look at other -non telomeric- regions. Is this effect specific for telomers are is it observed throughout the genome.

Authors' reply: **Figure 4G** now includes a GC-rich region (*Hras* locus) as a positive control, showing a greater accumulation of 8-oxoG lesions in the *Tcea1*^{-/-} and H₂O₂-treated wt MEFs, while an AT-rich region (centromeric major satellite repeat), with no such induction, served as a negative control.

-Figure 5D should be compared in one single IP experiment with supplemental fig 6A as more RNAPII is expected upon H2O2 treatment. This should be compared directly in one single experiment. Furthermore this experiment should be executed in TCEA1 KO cells.

Authors' reply: This has been corrected (**Figure 5D**). The IP in the *Tcea1*^{-/-} cells has been included (**Supplementary Figure 8A**).

-Suppl. Fig. 5F is not convincing, no clear differences in TRF1 localization are observed. If less TRF1 is bound to the chromatin in TCEA1 KO cells, were does this go, no increase in other compartments is observed. This should be repeated with IF experiments or by other more convincing experimetns?

Authors' reply: This has been corrected. We have now included TRF1 MFI measurements from IF experiments with and without pre-extraction (detergent-mediated removal of soluble proteins), to quantify the total, cytoplasmic, nuclear and chromatin-bound TRF1 fractions (**Supplementary Figure 7K**).

-Suppl. Fig. 5G, what is the proof the proteins reacting with UB antibody are TRF1, it might also be TRF1 interacting proteins that are ubiquitylated. If the authors want to make this claim, they should isolate ubiquitylated proteins and stain for TRF1 to detect higher migrating TRF1 species.

Authors' reply: This has been corrected. We have now included a new series of IP experiments using an anti-Ubiquitin antibody, and blotting for TRF1 (**Supplementary Figure 7M**).

-Figure 5G, the essential DRB only control sample is missing in this experiment

Authors' reply: This has been included (**Supplementary Figure 8B**).

-Suppl. Fig 6b-D, I guess the authors mean that TPL and actinomycin D are "NON"-reversible inhibitors?

Authors' reply: This has been corrected.

REVIEWER COMMENTS

Reviewer #1 (Remarks to the Author):

The authors have added a great deal of more data. The BLESS assay to show DBSs at telomeres strengthens the conclusions. I still think that the author cannot rule out contribution the contribution of R-loops or RNA PolII stalling outside telomeres to senescence and inflammatory responses in Tcea1^{-/-}. This is difficult to conclude based on TERT overexpression given roles for TERT in replicative senescence and proposed non-canonical roles. But their data does support a major role for R-loops and RNA PolII stalling at telomeres in determining the cellular outcomes. Therefore, this represents an important advance in my opinion. I still have an unresolved comment; see below.

Although the rebuttal states that Figure 3I shows a representative image of a fused telomere in Tcea1^{-/-} chromosomes, I could not find this in the figure. I'm still not convinced that the fusions arose from missing or critically short telomeres. How was a telomere classified as "short", what was the criteria? How short is "critically" short such that the telomere is not unprotected. Do the authors see telomeric sequence at the site of fusions; this should be apparent if the telomeres was "critically" short but not lost? Also, quantification of fragile telomeres is missing.

Reviewer #2 (Remarks to the Author):

The authors have performed an impressive number of additional experiments which partly strengthen the manuscript, however many of my methodological concerns were not addressed, therefore I still can't recommend this manuscript for publication.

1. There is no information provided regarding how MFI was calculated for the microscopy images.
2. MFIs are extremely inconsistent between experiments (for example in figure 3H the MFI of TelC for WT cells is the same as in Tcea1^{-/-} cells, and in Supplementary figure 11D the MFI of WT cells is 3 times higher than in Tcea1^{-/-} cells). None of which can be correlated with TRF from Figure 6H.
3. MFIs for TRF1 staining also don't correspond well to the example figures
4. If 53BP1 staining gives such a high background in this lab, positive controls should be demonstrated (treatment with DSB inducing agent), to show the signal specificity
5. The same applies to many experiments using BrdU staining (for example, Figure 2C and G)
6. FISH staining has a lot of background, which makes statements made based on figure 3G and 7B not strong enough.

My concern regarding telomere-specific origin of the genomic instability and senescence in case of TFIIS deficiency, did not completely disappear despite the additional experiments, here are major comments:

We overexpressed TERT (Supplementary Figure 11C) to increase telomere length in TFIIS-deficient cells (Supplementary Figure 11D) and show that the senescent characteristics associated with the TFIIS defect are attenuated (Supplementary Figure 11E-F).

What is the time frame of this experiment? Also, from quantification of Supplementary figure 11D it seems like overexpression of TFIIS and TERT could have synergistic effect, suggesting their independent contribution to the phenotype you observe. Have you overexpressed both of them at the same time?

- Using the DNA ligase IV-specific inhibitor SCR130 to inhibit NHEJ, we find that the chromosome ends that would be ligated/fused, present critically short or missing telomeres, enough to induce senescence (Figure 3I).

In order to make this statement, one needs to show accumulation of gammaH2AX on short telomeres in metaphase spreads

- We find that treating cells with the antioxidants MitoTEMPO or NAC to reduce the levels of 8-oxoG lesions (Figure 2A and Supplementary Figure 3A), leads to a decrease in pS2-PolII levels (Supplementary Figure 3E), the anticipated reduction of R-loop accumulation (Supplementary Figure 5C) and the attenuation of the transcription impairment as evidenced by BrU incorporation (Supplementary Figure 5A) in *Tcea1*^{-/-} MEFs.

This shows that indeed oxidative lesions in *Tcea1*^{-/-} MEFs lead to the described phenotypes, but not that these phenotypes are telomere specific.

- Using Breaks Labeling, Enrichment on Streptavidin, and Sequencing (BLESS) to quantify the telomere-specific DSB levels, we find that DNA breaks on telomeres are R-loop-dependent (Supplementary Figure 9B-C). This DSB accumulation is alleviated in NAC-treated cells (Supplementary Figure 9C) reducing genome instability and ultimately resulting in an increase in average telomere length of *Tcea1*^{-/-} cells (Figure 3H).

BLESS signal for Rpl13a in WT and *Tcea1*^{-/-} untreated cells in Supplementary figure 9B and 9C does not match which imply quite high variability of the method and therefore low sensitivity, at least for the given region. Could this be the reason RH treatment does not get significant result?

- We find an enrichment of TTAGGG repeats in cytosolic DNA fragments from *Tcea1*^{-/-} compared to wt cells, by sequencing purified cytoplasmic DNA fractions (Supplementary Figure 10 and 11A-B).

There is an increase of the signal from Rpl13a as well, and my guess is if you look at other R-loop prone sequences you are going to see enrichment as well. Telomeric R-loops represent a minuscule fraction of total R-loops in the cell.

- We employed a vesicle delivery system to treat wt and *Tcea1*^{-/-} MEFs as well as primary hepatocytes and pancreatic cells derived from 2- and 24-month-old mice with the ssDNA-specific S1 nuclease. Treatment of cells with S1 nuclease led to the significant reduction of cytoplasmic TelC fragments (Supplementary Figure 12A) and type I related immune gene expression (Figure 6C, 6F, Supplementary Figure 12B and 12H).

Could you probe or sequence for other R-loop prone regions to see if phenotype is the same?

My other concerns have either been addressed or are of less importance.

Reviewer #3 (Remarks to the Author):

The authors have provided a significantly improved manuscript with additional data to support their conclusions and overcome my previous comments/suggestion. I am happy to support the publication of the manuscript in its current form in Nature Communications.

REVIEWER COMMENTS

Reviewer #1:

*The authors have added a great deal of more data. The BLESS assay to show DBSs at telomeres strengthens the conclusions. I still think that the author cannot rule out contribution the contribution of R-loops or RNA PolII stalling outside telomeres to senescence and inflammatory responses in *Tcea1*^{-/-}. This is difficult to conclude based on TERT overexpression given roles for TERT in replicative senescence and proposed non-canonical roles. But their data does support a major role for R-loops and RNA PolII stalling at telomeres in determining the cellular outcomes. Therefore, this represents an important advance in my opinion. I still have an unresolved comment; see below.*

Authors' reply: We thank the Reviewer for her/his supporting comments. Indeed, we cannot rule out the contribution of R-loops or RNA PolII stalling outside telomeres to senescence and inflammatory responses in *Tcea1*^{-/-} cells. As TFIIIS is a general transcription elongation factor, its depletion would lead to genome-wide transcription-blocking lesions, impacting all transcribed loci. In the present work, we emphasize the previously unexplored role of transcription-associated DNA damage at telomeres in senescence. Certain sections of the manuscript have been revised to clarify this aspect.

Regarding the non-canonical functions of TERT in gene expression and senescence, we now provide an alternative telomere elongation approach described by Mukherjee *et al.*, 2018 and Wright *et al.*, 1996, in which *Tcea1*^{-/-} and wt cells are repetitively treated with G-rich terminal oligonucleotides leading to an increase in telomere length (**Supplementary Figure 12B**), a reduction in p16 mRNA levels (**Supplementary Figure 12C**) and to fewer SA-β-Gal⁺ cells (**Supplementary Figure 12D**).

*Although the rebuttal states that Figure 3I shows a representative image of a fused telomere in *Tcea1*^{-/-} chromosomes, I could not find this in the figure. I'm still not convinced that the fusions arose from missing or critically short telomeres. How was a telomere classified as "short", what was the criteria? How short is "critically" short such that the telomere is not unprotected. Do the authors see telomeric sequence at the site of fusions; this should be apparent if the telomeres was "critically" short but not lost ? Also, quantification of fragile telomeres is missing.*

Authors' reply: This has now been corrected. **Figure 3J** shows a representative image of fused chromosomes, with one telomeric FISH signal. All observed fusion events have a telomeric signal. We hypothesize that these events arise either between a chromosome end lacking a telomere and a chromosome end with a short telomere, or between two chromosome ends with short or broken telomeres. While we cannot precisely define the telomere length considered 'critically' short for inducing senescence, we acknowledge that unprotected telomeres of any length may lead to fusions. We have included MFI measurements of TelC-FISH signals on telomeres from both *Tcea1*^{-/-} and wt metaphases, which further support the observed telomere attrition in *Tcea1*^{-/-} MEFs. The graph in **Figure 3K** illustrates that the threshold we set to classify telomere signals as “short” is below the smallest wt MFI value (dashed line). **Figure 3J** now shows the percentage of metaphases with at least one fragile telomere.

Reviewer #2:

The authors have performed an impressive number of additional experiments which partly strengthen the manuscript, however many of my methodological concerns were not addressed, therefore I still can't recommend this manuscript for publication.

1. *There is no information provided regarding how MFI was calculated for the microscopy images.*

Authors' reply: This has now been included in the Methods section. For MFI measurements we followed the instructions for the ImageJ software. Specifically, z-stacks were prepared by using all sections (0.35 μ m intervals) of a scan, from slide to coverslip. Then, for total protein fluorescence intensity measurements, we split the channels and a ROI was designed (around the whole cell or nucleus). The Integrated Density (IntDen = mean x area) was calculated by subtracting the IntDen of a “background” ROI (non-fluorescent area) from the IntDen of the fluorescent signal. For cytoplasmic measurements, the nuclear IntDen was subtracted from the whole cell. For TRF1-TelC FISH, a ROI of the size of the telomere was drawn, and TRF1 IntDen was calculated on each telomere; an average IntDen per cell was plotted. Cells with cytoplasms or telomeres overlapping on the z-axis were excluded from quantifications.

2. MFIs are extremely inconsistent between experiments (for example in figure 3H the MFI of TelC for WT cells is the same as in *Tcea1*^{-/-} cells, and in Supplementary figure 11D the MFI of WT cells is 3 times higher than in *Tcea1*^{-/-} cells). None of which can be correlated with TRF from Figure 6H.

Authors' reply: There is considerable variation in experiments involving separate animals (embryos in our case). In order to minimize differences in measurements, every experiment was performed simultaneously in all embryos under identical conditions ensuring uniformity. Across all TelC experiments performed for this study, we consistently observed significantly decreased signals for the *Tcea1*^{-/-} cells compared to the wt cells (as shown in **Figure 3H**, **Supplementary Figure 11H**, and the newly included **Supplementary Figure 12B**). However, it's worth noting that the scale may not always be identical. Given that MFI measurements do not correspond directly to exact length (in kilobases), it's essential to note that the 2.1-fold difference in **Supplementary Figure 11H** (or 1.2-fold in **Figure 3H**) does not equate to a twofold difference in length. Consequently, these values cannot be directly compared to the TRF analysis presented in **Supplementary Figure 6H**.

3. MFIs for TRF1 staining also don't correspond well to the example figures

Authors' reply: We have now included an additional set of zoomed-in images (**Supplementary Figure 8C**). Of note, TRF1 MFI was measured only on TelC-positive foci, which could account for the discrepancies compared to the representative images (the additional signal seen outside telomeres).

4. If 53BP1 staining gives such a high background in this lab, positive controls should be demonstrated (treatment with DSB inducing agent), to show the signal specificity

Authors' reply: 53BP1 staining in untreated cells typically gives a pan-nuclear staining pattern, while upon DNA damage, a fraction of the protein is redistributed/translocated to the sites of damage (Schultz *et al.*, 2000; Rappold *et al.*, 2001; Xia *et al.*, 2001). Thus, the remaining fraction of 53BP1 protein is not considered background. As a positive control, we now include a representative image of etoposide-treated wt cells, stained against 53BP1 (**Supplementary Figure 4E**).

5. *The same applies to many experiments using BrdU staining (for example, Figure 2C and G)*

Authors' reply: BrU experiments generate cytoplasmic foci, considered true signals originating from RNA synthesized during BrU treatment. It's important to note that while these foci are acknowledged, they are not included in our nuclear measurements. This is more evident in **Supplementary Figure 5A-B** which show that the cytoplasmic foci are reduced in *Tcea1*^{-/-}, in line with the impaired transcription in these cells. This reduction is also consistent with the DRB-treated control shown in **Supplementary Figure 4D**. Similarly, the non-denaturing BrdU staining protocol in **Figure 2G** (ssDNA) shows some signal in the cytoplasm of *Tcea1*^{-/-} cells, not in wt cells. This finding fully aligns with the accumulation of cytoplasmic DNA fragments.

6. *FISH staining has a lot of background, which makes statements made based on figure 3G and 7B not strong enough.*

Authors' reply: In **Figure 7B** the EVs were incubated with cells on the coverslip, which was subsequently used during confocal microscopy without re-seeding the cells elsewhere. TelC-FISH was conducted after EV incubation, resulting in the labeling of telomeres within the nuclei. Any ExoFlow-TelC signal observed outside the cells originates from EVs that were not uptaken by cells. Our measurements only considered the cytoplasmic ExoFlow-TelC double-stained foci.

My concern regarding telomere-specific origin of the genomic instability and senescence in case of TFIIIS deficiency, did not completely disappear despite the additional experiments, here are major comments:

Authors' reply: We agree with the Reviewer's observation that we cannot disregard the potential contribution of stalled RNAPII and R-loops throughout the genome to the phenotype of *Tcea1*^{-/-} cells. Indeed, as TFIIIS is a general transcription elongation factor, its depletion can result in genome-wide transcription-blocking lesions, impacting all transcribed loci and leading to overall genome instability. In accordance, we confirm that genome-wide oxidative damage also affects telomeres by impeding RNAPII progression, resulting in R-loop accumulation. In the present work, we

emphasize the previously unexplored role of transcription-associated DNA damage at telomeres in senescence. Relevant modifications have been made to certain sections of the manuscript to enhance clarity on this aspect.

We overexpressed TERT (Supplementary Figure 11C) to increase telomere length in TFIIIS-deficient cells (Supplementary Figure 11D) and show that the senescent characteristics associated with the TFIIIS defect are attenuated (Supplementary Figure 11E-F).

What is the time frame of this experiment? Also, from quantification of Supplementary figure 11D it seems like overexpression of TFIIIS and TERT could have synergistic effect, suggesting their independent contribution to the phenotype you observe. Have you overexpressed both of them at the same time?

Authors' reply: Wt and *Tcea1*^{-/-} MEFs were cultured for one passage (P0 to P1, covering the isolation to approximately one division cycle, lasting 2 days). Subsequently, three consecutive electroporations were performed at P2, P3, and P4, involving TFIIIS- and/or TERT-expressing plasmids (every 2 days, as outlined in the Methods section). Two days after P4, the cells were harvested for various analyses, including BrU, FISH, SA-β-Gal assay, or RNA extraction. We now provide a new set of experiments where both TFIIIS and TERT were overexpressed simultaneously (**Supplementary Figure 11F-I and 12A**). Notably, the TFIIIS-TERT double transfection resulted in increased telomere length and decreased p16 mRNA levels and SA-β-Gal⁺ cells in *Tcea1*^{-/-} cells, aligning with the effects observed with each single transfection, and bringing the measurements to wt levels.

- Using the DNA ligase IV-specific inhibitor SCR130 to inhibit NHEJ, we find that the chromosome ends that would be ligated/fused, present critically short or missing telomeres, enough to induce senescence (Figure 3I).

In order to make this statement, one needs to show accumulation of gammaH2AX on short telomeres in metaphase spreads

Authors' reply: We now provide representative images of an SCR130-treated metaphase (**Figure 3I**) and an interphase *Tcea1*^{-/-} cell (**Supplementary Figure 6M**) stained against γH2Ax and labelled with the TelC-PNA probe, that show γH2Ax foci on telomeres.

- We find that treating cells with the antioxidants MitoTEMPO or NAC to reduce the levels of 8-oxoG lesions (Figure 2A and Supplementary Figure 3A), leads to a decrease in pS2-PolIII levels (Supplementary Figure 3E), the anticipated reduction of R-loop accumulation (Supplementary Figure 5C) and the attenuation of the transcription impairment as evidenced by BrU incorporation (Supplementary Figure 5A) in *Tcea1*^{-/-} MEFs.

This shows that indeed oxidative lesions in Tcea1^{-/-} MEFs lead to the described phenotypes, but not that these phenotypes are telomere specific.

Authors' reply: This is correct. We used these antioxidants to show the causal contribution of oxidative damage to transcription-induced R-loops and genomic instability. We hypothesize that this sequence of events (oxoG-stalled RNAPII/R-loops) takes place at telomeres as well, but telomere-associated DNA damage is not the sole contributor to senescence in *Tcea1*^{-/-} cells. Nevertheless, telomeres are among the genomic regions that are affected the most from 8-oxoG accumulation, due to their G-rich tandem sequence. In line with our results, it has been previously shown that oxidative stress on telomeres alone can lead to telomeres crisis and can drive rapid premature senescence even in the absence of telomere shortening (Fouquerel et al., 2019; Barnes et al., 2022). We now show that these defects can occur in DNA repair proficient cells, driven by transcription stress-induced 8-oxoG accumulation.

- *Using Breaks Labeling, Enrichment on Streptavidin, and Sequencing (BLESS) to quantify the telomere-specific DSB levels, we find that DNA breaks on telomeres are R-loop-dependent (Supplementary Figure 9B-C). This DSB accumulation is alleviated in NAC-treated cells (Supplementary Figure 9C) reducing genome instability and ultimately resulting in an increase in average telomere length of Tcea1^{-/-} cells (Figure 3H).*

BLESS signal for Rpl13a in WT and Tcea1^{-/-} untreated cells in Supplementary figure 9B and 9C does not match which imply quite high variability of the method and therefore low sensitivity, at least for the given region. Could this be the reason RH treatment does not get significant result?

Authors' reply: **Supplementary Figure 9B** (at passage 5) and **9C** (at passage 4) show the levels of DSBs (relative accumulation, not actual numbers) for *Rpl13a* to be at 0.05

for wt and approximately 0.05 to 0.09 for *Tcea1*^{-/-} cells. There was a small variation between the *Tcea1*^{-/-} biological replicates in Supplementary Figure 9C. We replicated the experiments following RNase H transfection, now incorporating results from two additional biological replicates. The accumulation of DSBs on the Rpl13α locus in TFIIIS-deficient cells is significantly higher compared to wt controls.

- We find an enrichment of TTAGGG repeats in cytosolic DNA fragments from Tcea1^{-/-} compared to wt cells, by sequencing purified cytoplasmic DNA fractions (Supplementary Figure 10 and 11A-B).

There is an increase of the signal from Rpl13a as well, and my guess is if you look at other R-loop prone sequences you are going to see enrichment as well. Telomeric R-loops represent a minuscule fraction of total R-loops in the cell.

Authors' reply: Indeed, we would expect to find sequences from other genes that accumulate co-transcriptional R-loops due to oxoG-induced stalled RNAPII. We provide additional examples of such genes in **Supplementary Figure 11C-E**.

- We employed a vesicle delivery system to treat wt and Tcea1^{-/-} MEFs as well as primary hepatocytes and pancreatic cells derived from 2- and 24-month-old mice with the ssDNA-specific S1 nuclease. Treatment of cells with S1 nuclease led to the significant reduction of cytoplasmic TelC fragments (Supplementary Figure 12A) and type I related immune gene expression (Figure 6C, 6F, Supplementary Figure 12B and 12H).

Could you probe or sequence for other R-loop prone regions to see if phenotype is the same?

Authors' reply: **Supplementary Figure 11C-E** shows additional examples of genes prone to R-loops, with sequences found to be enriched in the cytoplasmic fraction of *Tcea1*^{-/-} cells. Furthermore, we used FISH to label a gene that was previously shown to accumulate R-loops upon transcription activation (Chatzinikolaou et al., 2023). Using *trans* retinoic acid (tRA) to induce transcription in wt and *Tcea1*^{-/-} MEFs, we performed FISH with a Cy3-labeled probe targeting the *Rarb* locus. Gene expression is induced in the *Rarb* locus upon tRA treatment and leads to promoter and terminator R-loop formation, yet the resolution provided by the FISH approach did not generate visibly labelled *Rarb* DNA fragments in the cytoplasm of tRA-treated *Tcea1*^{-/-} MEFs.

My other concerns have either been addressed or are of less importance.

Reviewer #3:

The authors have provided a significantly improved manuscript with additional data to support their conclusions and overcome my previous comments/suggestion. I am happy to support the publication of the manuscript in its current form in Nature Communications.

Authors' reply: We thank the Reviewer for her/his supportive remarks.

REVIEWERS' COMMENTS

Reviewer #1 (Remarks to the Author):

The authors have satisfied most of my prior concerns.

However, the following comment from Reviewer 2 was not adequately addressed. These figures show gammaH2AX on short telomeres in metaphase spreads as requested by the reviewer (Figure 3I or Supplementary Figure 6M). These metaphase chromosomes in 3I are not clear (i.e. it is difficult to discern chromosome ends), and the telomere length of the telomeres co-localized with gammaH2AX was not measured. The authors may need to soften the statement from "we find that" to "this suggests that".

Reviewer 2:

"Using the DNA ligase IV-specific inhibitor SCR130 to inhibit NHEJ, we find that the chromosome ends that would be ligated/fused, present critically short or missing telomeres, enough to induce senescence (Figure 3I)."

In order to make this statement, one needs to show accumulation of gammaH2AX on short telomeres in metaphase spreads

Authors' reply: We now provide representative images of an SCR130-treated metaphase (Figure 3I) and an interphase Tcea1^{-/-} cell (Supplementary Figure 6M) stained against γH2Ax and labelled with the TelC-PNA probe, that show γH2Ax foci on telomeres.

Response to Reviewers

Reviewer #1 (Remarks to the Author):

The authors have satisfied most of my prior concerns.

Authors' reply: We thank the Reviewer for her/his constructive and supportive remarks and for taking the time to comment on the remaining issues.

However, the following comment from Reviewer 2 was not adequately addressed. These figures show gammaH2AX on short telomeres in metaphase spreads as requested by the reviewer (Figure 3I or Supplementary Figure 6M). These metaphase chromosomes in 3I are not clear (i.e. it is difficult to discern chromosome ends), and the telomere length of the telomeres co-localized with gammaH2AX was not measured. The authors may need to soften the statement from "we find that" to "this suggests that".

Reviewer 2:

"Using the DNA ligase IV-specific inhibitor SCR130 to inhibit NHEJ, we find that the chromosome ends that would be ligated/fused, present critically short or missing telomeres, enough to induce senescence (Figure 3I)."

In order to make this statement, one needs to show accumulation of gammaH2AX on short telomeres in metaphase spreads

Authors' reply: We now provide representative images of an SCR130-treated metaphase (Figure 3I) and an interphase Tcea1^{-/-} cell (Supplementary Figure 6M) stained against gammaH2Ax and labelled with the TelC-PNA probe, that show gammaH2Ax foci on telomeres.

Authors' reply: The sentence was already changed in the manuscript and now reads "Indeed, specific inhibition of DNA Ligase IV, with the use of the selective inhibitor SCR130, revealed gammaH2Ax-stained telomeres (Figure 3I and Supplementary Figure 6M) and led to Tcea1^{-/-} metaphase chromosomes without fusions, but with discernible chromosome ends with short or completely missing telomeres (Figure 3J)."